# KIN10 promotes stomatal development through stabilization of the SPEECHLESS transcription factor

Chao Han [1], Yue Liu [1], Wen Shi [1], Yan Qiao [1], Lingyan Wang [1], Yanchen Tian[1], Min Fan [1], Zhiping Deng [2], On Sun Lau [3], Geert De Jaeger[4,5] & Ming-Yi Bai [1]✉

Stomata are epidermal structures that modulate gas exchanges between plants and the atmosphere. The formation of stomata is regulated by multiple developmental and environmental signals, but how these signals are coordinated to control this process remains unclear. Here, we showed that the conserved energy sensor kinase SnRK1 promotes stomatal development under short-day photoperiod or in liquid culture conditions. Mutation of *KIN10*, the catalytic α-subunit of SnRK1, results in the decreased stomatal index; while over-expression of *KIN10* significantly induces stomatal development. KIN10 displays the cell-type-specific subcellular location pattern. The nuclear-localized KIN10 proteins are highly enriched in the stomatal lineage cells to phosphorylate and stabilize SPEECHLESS, a master regulator of stomatal formation, thereby promoting stomatal development. Our work identifies a module links connecting the energy signaling and stomatal development and reveals that multiple regulatory mechanisms are in place for SnRK1 to modulate stomatal development in response to changing environments.

[1] The Key Laboratory of Plant Development and Environmental Adaptation Biology, Ministry of Education, School of Life Sciences, Shandong University, Qingdao 266237, China. [2] State Key Laboratory for Quality and Safety of Agro-products, Institute of Virology and Biotechnology, Zhejiang Academy of Agricultural Sciences, Hangzhou 310021, China. [3] Department of Biological Sciences, National University of Singapore, 14 Science Drive 4, Singapore 117543, Singapore. [4] Department of Plant Biotechnology and Bioinformatics, Ghent University, Ghent, Belgium. [5] VIB Center for Plant Systems Biology, Ghent, Belgium. ✉email: baimingyi@sdu.edu.cn

Stomata, the pores on the surface of leaves, regulate gas exchange between plants and the atmosphere and play critical roles for maintaining photosynthetic and water-use efficiency in plants. Stomatal development is highly plastic, and is modulated by multiple intrinsic developmental and environmental signals[1,2]. The stomatal lineage in *Arabidopsis thaliana* is initiated by asymmetric divisions of undifferentiated meristemoid mother cells (MMC) to produce a small triangular cell meristemoid and a larger sister cell, stomatal-lineage ground cell (SLGC). The meristemoid either differentiates into a guard mother cell, which undergoes a single symmetric division to generate a pair of guard cells or goes through several additional rounds of asymmetric divisions to generate more SLGCs that then undergo spacing divisions to create satellite meristemoids or alternatively differentiate into pavement cells[1–3]. The prime regulator linked to the initiation and proliferation of stomatal precursors is the basic helix-loop-helix transcription factor SPEECHLESS (SPCH), which controls hundreds of downstream genes to promote cell division and fate transitions[4,5]. SPCH also integrates a wide range of environmental and hormonal signals to regulate stomatal development[6–9].

Maintenance of cellular and organismal energy homeostasis is a major challenge for all living organisms. Sucrose non-fermenting-1 (SNF1)-related kinase 1 (SnRK1) is a central energy sensor kinase in plants that is functionally and evolutionarily conserved with SNF1 in yeast and AMP-activated kinase (AMPK) in animals[10–13]. Similar to its orthologs SNF1 and AMPK, SnRK1 activation by energy-depleting stress conditions induces catabolic reactions and represses energy-consuming anabolic processes, which redirect energy resources to support stress tolerance and survival[12,13]. The eukaryotic AMPK/SNF1/SnRK1 protein kinases typically function as heterotrimeric complexes composed of one α-catalytic subunit (KIN10, KIN11, and KIN12 in *Arabidopsis*) and two regulatory subunits, β and γ[11,12]. SnRK1 acts within complex signaling networks to maintain energy metabolism. Phosphorylation of a conserved threonine in the activation T-loop of the catalytic subunit (such as KIN10$^{T175}$ and KIN11$^{T176}$) is important for SnRK1 activity[14–16]. Two upstream kinases SnRK1 Activating Kinase 1 and 2 (SnAK1 and SnAK2 are Geminivirus Rep-interacting Kinase 2 and 1, respectively) were identified to interact with KIN10 and phosphorylate it at the conserved Thr175[16–18]. The phosphatases ABI1 and PP2CA, two important components of ABA signaling pathway, were found to interact with and dephosphorylate KIN10, causing its inactivation[15]. In yeast and animal cells, SNF1 and AMPK are regulated by adenine nucleotide charge, but plant SnRK1 has been reported to be insensitive to adenosine mono- and diphosphate (AMP and ADP)[10,19]. Further, plant SnRK1 is known to be inhibited by diverse sugar phosphates and is highly sensitive to trehalose-6-phosphate (Tre6P)[20–22]. Tre6P, a signal of cellular sucrose status, binds directly to KIN10 to inhibit its interaction with the upstream kinases SnAK1/2[23]. SnRK1 is also regulated by sumoylation- and ubiquitination-mediated degradation, as well as intracellular redox-mediated oxidation[24–26]. Two recent studies reported that KIN10 was dynamically localized in endoplasmic reticulum and nucleus, and translocated to the nucleus to reprogram gene expression under metabolic stress conditions[27,28].

Sugar has been reported to promote stomatal formation in liquid culture condition, but the molecular mechanism remains unclear[29]. Here, we showed that KIN10 is involved in the sugar-promoted stomatal development. When plants were grown in liquid media, sucrose treatment significantly induced the KIN10 protein accumulation by increasing its translation. Overexpression of *KIN10* resulted in the increased stomatal index, while the loss of function of *KIN10* led to the reduced stomatal index. KIN10 displayed the cell-type-specific subcellular localization pattern in the epidermal cells of leaves, mainly localized in the nucleus of the stomatal-lineage cells and guard cells, but localized in the cytoplasm of the pavement cells. In stomatal-lineage cells, the nuclear-localized KIN10 phosphorylated and stabilized SPCH to promote stomatal development. These results demonstrated that fine-tuning of KIN10 activity by environmental and developmental signals optimizes stomatal development in Arabidopsis.

## Results

**Sugar promotes stomatal development under certain conditions.** Sugar has been shown to play crucial roles in the proliferation and differentiation of plant meristems[30–32], but the molecular mechanism is still fragmentary. Stomata, the pores on plant epidermis, that facilitate gas exchange with the atmosphere, comprise an excellent system for investigating how environmental and developmental signals regulate cell fate determination in plants[1,2]. To assess the effects of sugar on stomatal development, we analyzed the stomatal production under different light conditions and in the presence or absence of exogenous sucrose. Consistent with previous results[33,34], the stomatal index, which is the number of stomata relative to total epidermal cells, gradually increased with the increasing light photon irradiance (Fig. 1a). Similarly, the stomatal index of plants grown under the 16 h light/8 h dark photoperiod condition was much higher than that of plants grown under the 4 h light/20 h dark photoperiod condition (Fig. 1b). However, the decreased stomatal index of plants resulting from the low light quantity and short-time light irradiance was partially recovered by the exogenous sucrose supply (Fig. 1a, b). Prolonged darkness, which induces the carbon starvation phenotype in seedlings due to the lack of photosynthesis and the depletion of the reserved starch, inhibited stomatal development (Supplementary Fig. 1a). Under such starvation conditions, exogenous application of sucrose partly increased stomatal development, and the stomatal index increased from 11 to 15% approximately (Supplementary Fig. 1b). To evaluate the role of sugar in the development of stomata, different types of sugar were used to treat the wild-type seedlings that were grown under the 16 h light/8 h dark condition for 10 days in liquid half-strength MS medium, which might cause the mild hypoxia. Sucrose exerted the most obvious positive effects on the development of stomata (Fig. 1c). Glucose, fructose, and fructose bisphosphate (FBP) also could induce stomatal development, but glucose-6-phosphate (G6P) and mannitol (Man) had no significant effects on stomatal development (Fig. 1c). Treatment with sucrose not only promoted cell expansion but also induced cell division, and led to the increased amounts of stomata in the whole cotyledon and high ratio of clustered stomata (Supplementary Fig. 2a–h). These results indicated that sugar promotes stomatal development under certain conditions, such as low light quantity, short-day photoperiod, or in hypoxic liquid growth media.

To further verify the effects of sucrose on the stomatal-lineage cell fate transition, we analyzed the sucrose-treated stomatal development phenotype in different stomatal cell-type-specific marker lines, including: *pSPCH::nucGFP* that was mostly expressed in the cells of the stomatal lineage[4], *pSPCH::SPCH-GFP* that was only expressed in the MMCs and meristemoids[4], *pBASL::GFP-BASL* that was expressed in the asymmetrically dividing MMCs, meristemoids and the larger SLGCs[35]; and *pMUTE::MUTE-GFP* that was expressed in the guard mother cells[36]. Sucrose treatment dramatically increased the number of cells marked by *pSPCH::nucGFP, pSPCH::SPCH-GFP, pMUTE:: MUTE-GFP* and *pBASL::BASL-GFP* relative to total cells

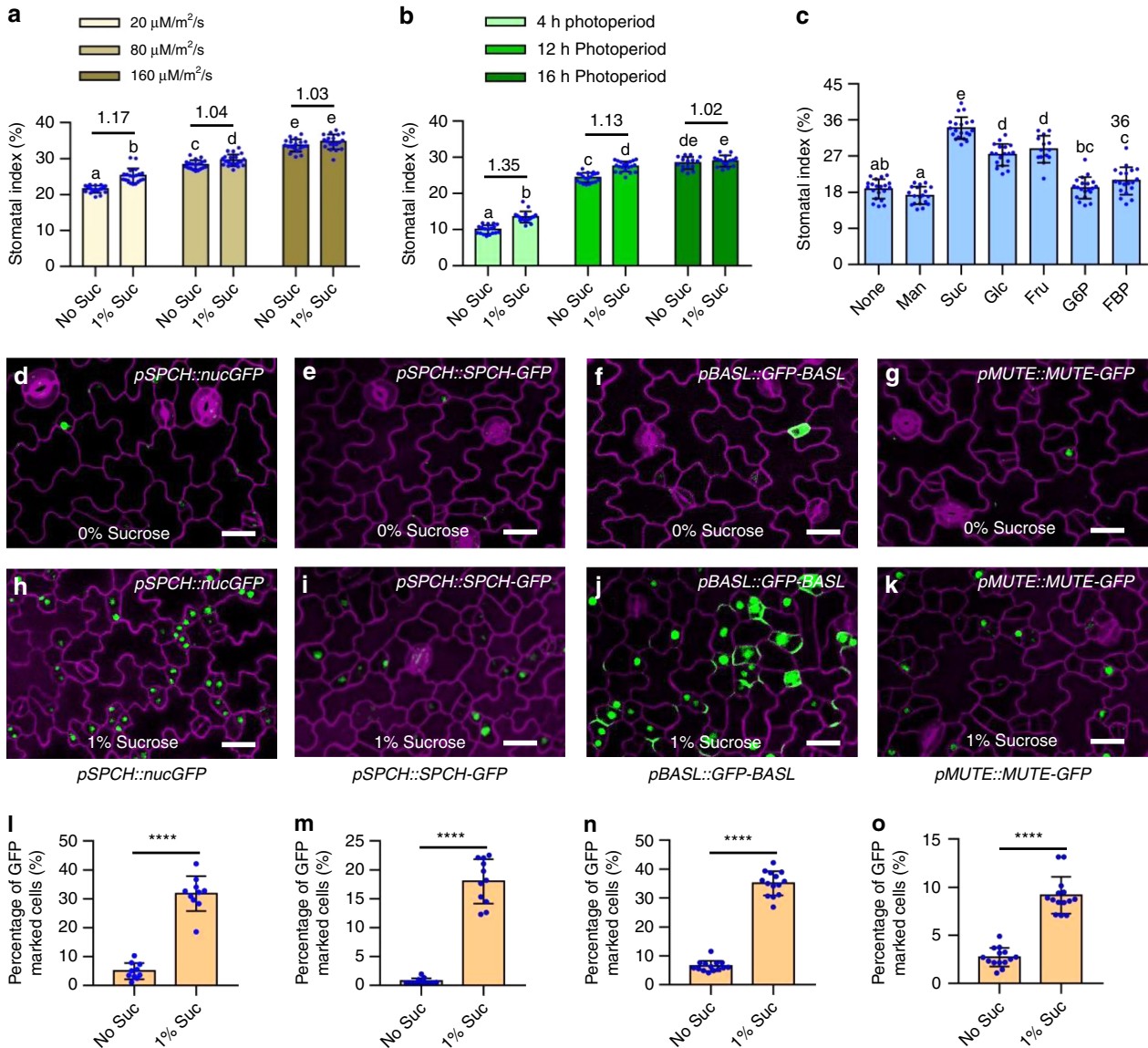

**Fig. 1 Sucrose promotes stomatal development and alters cell fate in Arabidopsis epidermis. a** The stomatal index changes in response to light quantity. Seedlings of wild-type plants were grown on ½ MS solid medium with or without 1% sucrose under 16 h light/8 h dark photoperiod with different light intensity for 10 days. **b** Quantification of the effects of photoperiod and exogenous sucrose on stomatal development. Seedlings of wild-type plants were grown on ½ MS solid medium with or without 1% sucrose under different photoperiod with 100 µMol m$^{-2}$ s$^{-1}$ for 10 days. Numbers between bars indicated the relative fold changes of average means in the indicated conditions. **c** Quantification of the effects of different sugars on stomatal index. Seedlings of wild type Col-0 were grown in ½ MS liquid medium containing 30 mM mannitol (Man, 0.55%), sucrose (Suc, 1%), glucose (Glu, 0.54%), fructose (Fru, 0.54%), glucose-6-phosphate (G6P, 0.78%), and fructose-1,6-bisphosphate (FBP, 1%) under 16 h light/8 h dark photoperiod with 100 µMol m$^{-2}$ s$^{-1}$ continuously for 10 days. Error bars indicate standard deviation (S.D.) ($n = 15$–$20$). Different letters above the bars indicated statistically significant differences between the samples (ANOVA analysis followed by Uncorrected Fisher's LSD multiple comparisons test, $p < 0.05$). **d–o** Sucrose alters cell fate in Arabidopsis epidermis. Seedlings of *pSPCH::nucGFP*, *pSPCH::SPCH-GFP*, *pBASL::GFP-BASL* and *pMUTE::MUTE-GFP* were grown in ½ MS liquid medium with or without 1% sucrose for 3 days under long-day condition. Quantification of the percentage of GFP-expressing cells over total epidermal cells of different transgenic plants. The fluorescent signals of GFP or GFP fused proteins are in green, PI-marked cell outlines are in purple. Scale bars in confocal images represent 20 µm. Error bars indicate S.D. ($n = 15$–$20$). Asterisk between bars indicated statistically significant differences between the samples (Student $t$ test, ****$p < 0.0001$).

(Fig. 1d–o). These results indicated that sucrose alters the epidermal cells fate in the Arabidopsis leaves.

**KIN10 positively regulates stomatal development**. Previous study showed that the prolonged darkness and hypoxia could activate the energy sensor SnRK1 to regulate the plant growth and stress response[13]. Our study showed that sucrose promoted stomatal development only when plants were grown under the short-

day photoperiod or in hypoxic liquid culture conditions, indicating that SnRK1 might be involved in this process. To test this hypothesis, we analyzed the stomatal phenotype of SnRK1-related materials under different growth conditions. First, when plants were grown in the liquid half-strength MS medium with 1% sucrose, we observed that the transgenic plants overexpressing *KIN10*, a catalytic subunit of SnRK1, displayed the increased stomatal index phenotype comparing to wild-type plants (Fig. 2a).

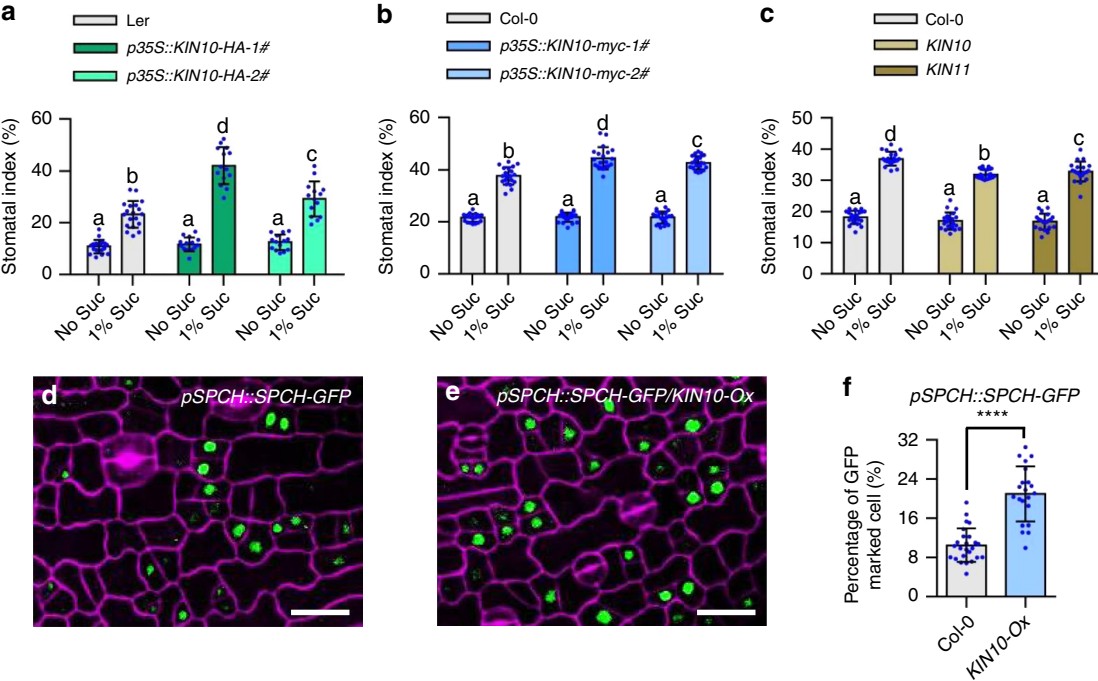

**Fig. 2 KIN10 plays crucial roles in sugar-induced stomatal development. a–c** Quantification of the stomatal index of wild-type plants, *KIN10-Ox* and *kin10* mutants. Seedlings of wild-type plants (Ler and Col-0), *p35S::KIN10-HA* (Ler background), *p35S::KIN10-myc* (Col-0 background), *kin10* mutant and *kin11* were grown in ½ MS liquid medium with or without 1% sucrose for 10 days under long-day condition. Error bars indicate standard deviation (S.D.) (*n* = 15–20). Different letters above the bars indicated statistically significant differences between the samples (ANOVA analysis followed by Uncorrected Fisher's LSD multiple comparisons test, *p* < 0.05). **d–f** Overexpression of *KIN10* increased the percentage of SPCH-YFP labeled cells to total epidermal cells. Seedlings of *pSPCH::SPCH-GFP* and *pSPCH::SPCH-GFP/KIN10-Ox* (*p35S::KIN10-myc-2#*) were grown in ½ MS liquid medium with 1% sucrose concentration conditions for 3 days under 16 h light/8 h dark photoperiod. SPCH-GFP is in green, PI-marked cell outlines are in purple. Scale bars in confocal images represent 20 μm. Error bars indicate S.D. (*n* = 20). Asterisk between bars indicated statistically significant differences between the samples (Student *t* test, ****\*\*\*\*p* < 0.0001).

Because the *KIN10-Ox* genetic material that we obtained from Arabidopsis Biological Resource Center is in the *Landsberg* (Ler) background, in which ERECTA (ER), a key component of stomatal development, is mutated, it showed the increased number of stomata[37]. To rule out the effects of *ER* mutation, we generated *KIN10*-overexpressing transgenic plants in the *Columbia-0* (Col-0) background. The results showed that, in the presence of sucrose, overexpression of *KIN10* in the Col-0 background also led to the increased stomatal index, elevated stomata number in whole cotyledon, and higher ratio of clustered stomata compared to Col-0 plants. (Fig. 2b and Supplementary Fig. 3a–i). In addition, both *kin10* and *kin11* single mutants exhibited the lower stomatal index in liquid medium containing 1% sucrose, indicating that SnRK1 kinase is required for sucrose-induced stomatal development (Fig. 2c and Supplementary Fig. 4a–c). Second, we analyzed the stomatal phenotypes of wild type Col-0, *KIN10-Ox*, and *kin10* seedlings that were grown on half-strength MS solid medium with or without 1% sucrose under different photoperiod conditions. The results showed that overexpression of *KIN10* significantly induced stomatal development in the presence of sucrose, particularly under the 4 h light/20 h dark photoperiod condition (Supplementary Fig. 5a–c). Third, to determine whether the positive effect of KIN10 on stomatal development is similar in the cotyledons and true leaves, the fifth rosette leaves of wild type, *KIN10-Ox* and *kin10* plants grown in soil under the 4 h light/20 h dark photoperiod for 5 weeks were used for stomatal index analysis. The stomatal index of rosette leaves of *KIN10-Ox* was higher, and that of *kin10* mutant was lower than the stomatal index of wild type rosette leaves (Supplementary Fig. 6a–d). These results indicated that KIN10 positively regulates stomatal development under certain conditions.

To further verify the effects of KIN10 on stomatal development, we analyzed the stomatal phenotype of wild-type plants grown in liquid medium with Tre6P, which is a known inhibitor of KIN10 through weakening the interaction between KIN10 and its upstream kinases SnAK1 and SnAK2[23]. The results showed that cotreatment with Tre6P strongly counteracted the sucrose effects and reduced the stomatal index (Supplementary Fig. 7a). Trehalose-6-Phosphate Synthase1 (TPS1) is the key enzyme for the biosynthesis of Tre6P in plants[38]. Mutation of *TPS1* resulted in a significantly increased stomatal index comparing to that of wild-type plants in the presence of sucrose (Supplementary Fig. 7b). Furthermore, we found that the transgenic plants of *pKIN10::KIN10^{T175D}–myc*, *pKIN10::KIN10^{T175E}–myc* and *p35S: SnAK2-GFP* showed the increased stomatal index comparing to wild-type plants (Supplementary Figs. 8a–c and 9a, b). These results further confirmed that KIN10 is a positive regulator for stomatal development.

Next, we analyzed whether KIN10 regulates the cell fate transition of stomatal-lineage cells. The expression levels of stomatal cell-type-specific markers were monitored in *KIN10-Ox* and wild-type plants. The results showed that a higher fraction of cells labeled by *pSPCH::SPCH-GFP* in *KIN10-Ox* than in wild-type plants (Fig. 2d–f). The difference between *KIN10-Ox* and wild-type plants for SPCH-GFP accumulation was much higher than the difference for stomatal index. This might be due to that the accumulated SPCH protein induces cell division to form many small cells, but only a few small cells will eventually develop into stomata[4]. These results indicated that KIN10 promotes the differentiation of epidermal cells into stomatal-lineage cells, leading to enhanced stomatal formation.

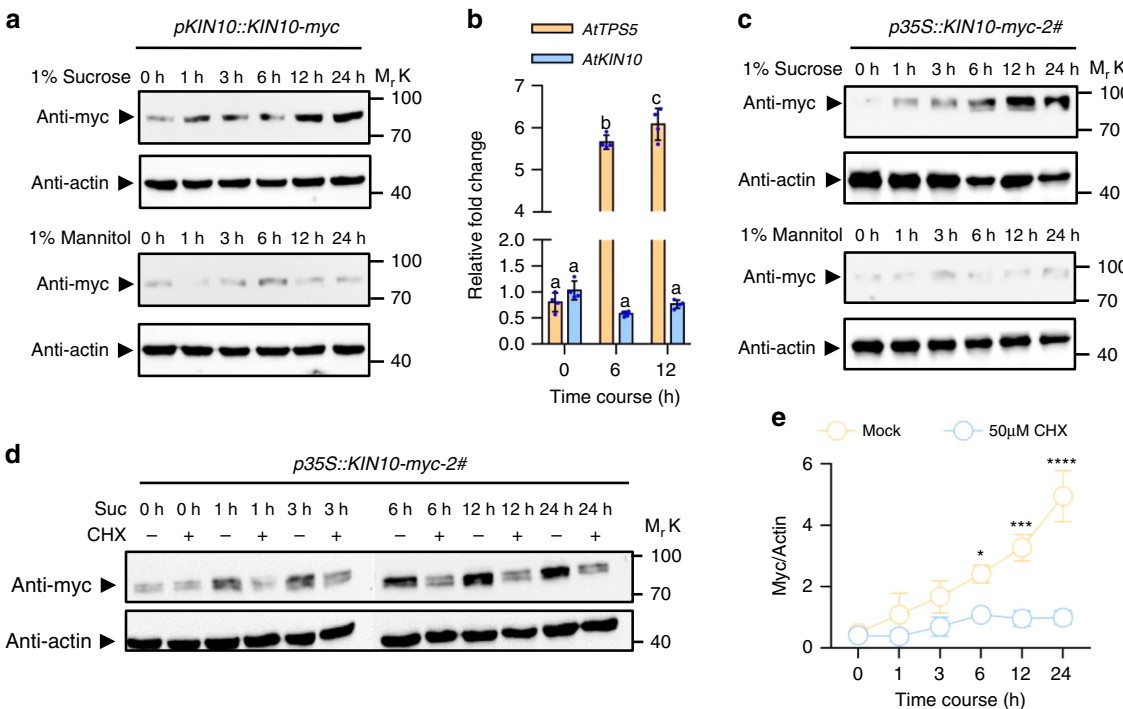

**Fig. 3 Sucrose induces the KIN10 protein accumulation. a** Immunoblot analysis of the protein levels of KIN10-myc in *pKIN10::KIN10-myc* transgenic plants using anti-myc antibody. Seedlings of *pKIN10::KIN10-myc* transgenic plants were grown in sugar free ½ MS liquid medium for 3 days and then treated with 1% sucrose or 1% mannitol for different time periods. **b** Quantitative RT-PCR analysis the expression of *KIN10* in response to sucrose. Wild-type seedlings were grown in sugar free 1/2 MS liquid medium for 3 days then treated with 1% sucrose for different times. *PP2A* gene was analyzed as an internal control. Error bars represent standard deviation of four independent experiments. Different letters above the bars indicated statistically significant differences between the samples (ANOVA analysis followed by Uncorrected Fisher's LSD multiple comparisons test, $p < 0.05$). **c** Sucrose induces KIN10 protein accumulation in *p35S::KIN10-myc* transgenic plants. Seedlings of *p35S::KIN10-myc* transgenic plants were grown in sugar free ½ MS liquid medium for 3 days and then treated with 1% sucrose or 1% mannitol for different time periods. **d** Immunoblot analysis the effects of cycloheximide (CHX) on the Sucrose-induced the KIN10 protein accumulation. Seedlings of *p35S::KIN10-myc* transgenic plants were grown in sugar free ½ MS liquid medium for 3 days and then treated with or without 1% sucrose and 50 µM CHX for different time periods. **e** Quantification analysis of CHX effect on sucrose-induced KIN10 protein accumulation. The ratio of immunoprecipitated KIN10-myc to actin was quantified by ImageJ software. Values for different time points after sucrose addition with or without CHX treatment are given as mean ± S.E. ($n = 3$). Asterisk above dots indicates significant inhibition of KIN10 protein accumulation by CHX treatment at different time points (ANOVA analysis followed by Uncorrected Fisher's LSD multiple comparisons test, *$p < 0.05$; **$p < 0.01$; ****$p < 0.0001$).

**Sucrose induces KIN10 protein accumulation**. To investigate how sugar optimizes stomatal development through KIN10, we first analyzed the effects of sucrose on the protein levels of KIN10 in the *pKIN10::KIN10-myc* transgenic plants that were grown in liquid half-strength MS medium without any sugar for 3 days and then treated with sucrose or mannitol for different time spans. Sucrose treatment increased the protein levels of KIN10-myc, which became more obvious after 6 h of treatment, while mannitol showed no significant effects on the protein levels of KIN10-myc (Fig. 3a). To examine whether the accumulation of KIN10 protein was due to the increased transcriptional levels of *KIN10*, we performed quantitative reverse transcription-PCR (qRT-PCR) assays and these revealed that sucrose significantly induced the expression of *AtTPS5*, a known sucrose-induced gene, but had no effect on the transcription of *KIN10* (Fig. 3b), indicating that sucrose induces the accumulation of KIN10 via posttranscriptional regulation. To test this hypothesis, we analyzed the KIN10 protein levels in the *p35S::KIN10-myc* transgenic plants, and also found that sucrose treatment significantly induced the accumulation of KIN10 protein (Fig. 3c). Next, we analyzed the sucrose-induced accumulation of KIN10 in the presence or absence of cycloheximide (CHX), a protein translation inhibitor. Without new protein synthesis, the ability of sucrose to induce KIN10 protein accumulation diminished (Fig. 3d, e), suggesting that

sucrose induces the accumulation of KIN10 protein through the activation of KIN10 protein translation.

**KIN10 interacts with SPCH in vivo and in vitro**. SPCH is a master regulator of stomatal development downstream of a wide range of environmental and hormonal signals[6–9]. We found that sucrose treatment and overexpression of *KIN10* both resulted in the increased levels of SPCH (Figs. 1m and 2f), indicating that sucrose and KIN10 might regulate stomatal development through SPCH. To test this hypothesis, we first analyzed the interaction between KIN10 and SPCH. Yeast two-hybrid analysis showed that KIN10 interacts with SPCH, but not MUTE and FAMA (Fig. 4a and Supplementary Fig. 10). In vitro protein–protein pull-down assays showed that glutathione S-transferase (GST)-SPCH interacts with maltose-binding protein (MBP)-KIN10, but not MBP alone (Fig. 4b). The ratiometric bimolecular fluorescence complementation (rBiFC) assays showed that the strong YFP fluorescence signal was observed in the nucleus when KIN10-nYFP was co-transformed with SPCH-cYFP in the same rBiFC vector (Fig. 4c). Consistent with the rBiFC and in vitro pull-down assays, coimmunoprecipitation assays using transgenic plants expressing *KIN10-YFP* and *SPCH-myc* from *35S* promoters confirmed their interaction in plants (Fig. 4d and Supplementary Fig. 11). We further analyzed the tissue expression patterns of

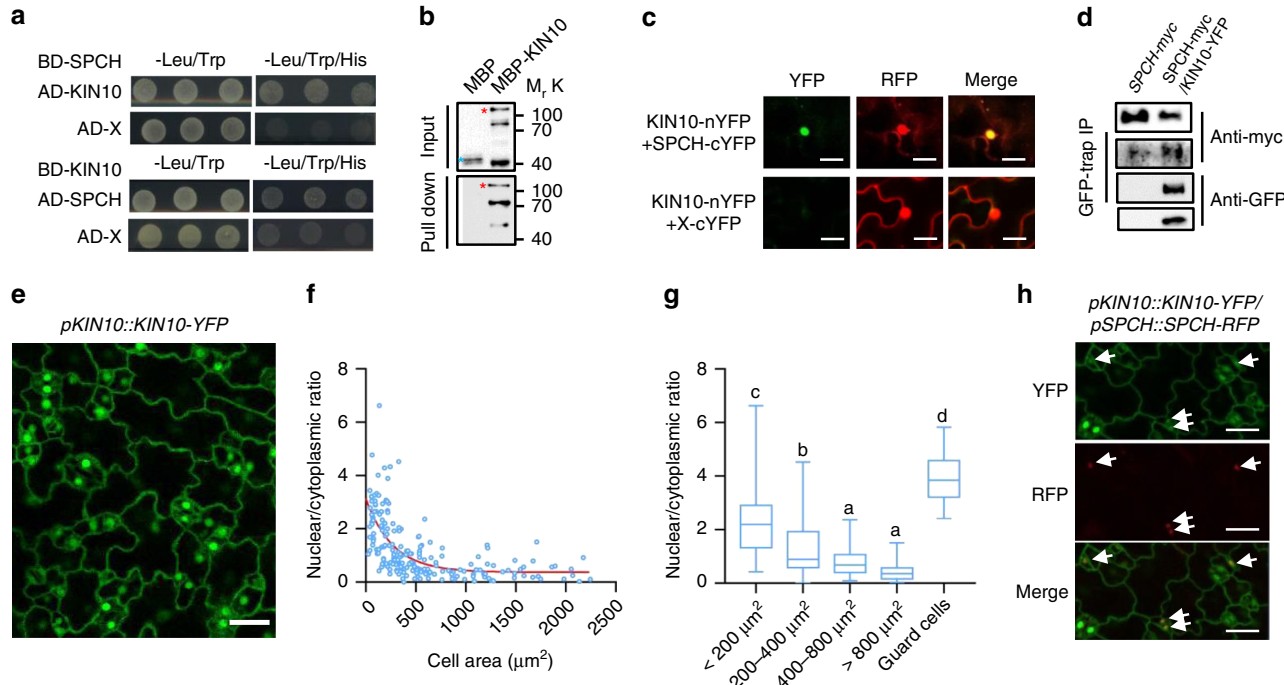

**Fig. 4 KIN10 interacts with SPCH in vitro and in vivo. a** KIN10 interacts with SPCH in yeast. **b** KIN10 directly interacts with SPCH in vitro. MBP and MBP-KIN10 were incubated with GST-SPCH bound to glutathione agarose beads and then eluted and analyzed by immunoblotting using anti-MBP antibody. Red asterisk indicated KIN10-MBP, while blue asterisk indicated MBP. **c** rBiFC showed that KIN10 interacts SPCH in tobacco leaves. **d** Coimmunoprecipitation assay of KIN10 and SPCH with seedlings expressing KIN10-YFP and SPCH-myc under *35S* promoters or seedlings expressing only SPCH-myc. CoIP was performed using GFP-trap and immunoblotted using anti-GFP and anti-Myc antibodies. **e** The subcellular location of *pKIN10::KIN10-YFP* in Arabidopsis cotyledon epidermal cells. Seedlings of *pKIN10::KIN10-YFP* were grown in ½ MS liquid medium containing 1% Sucrose for 3 days under long-day condition. Scale bar, 20 μm. **f, g** Quantification of nuclear localization of KIN10 in different scale epidermal cells of (**e**). Nuclear and cytoplasmic KIN10-YFP signal from more than 200 epidermal cells in 10 cotyledons were analyzed by ImageJ software. Scatter plot (**f**) and box plot (**g**) showed negative relationship of KIN10-YFP nuclear/cytoplasmic ratio and the size of epidermal cells. Error bars indicate standard deviation (S.D.). Different letters above the bars indicated statistically significant differences between the samples (ANOVA analysis followed by Uncorrected Fisher's LSD multiple comparisons test, $p < 0.05$). **h** Co-localization of *pKIN10::KIN10-YFP* and *pSPCH::SPCH-RFP* in cotyledon epidermal cells. The seedling was grown in ½ MS liquid culture containing 1% Sucrose. Scale bars in confocal images represent 20 μm. Arrow indicates the colocalized KIN10-YFP and SPCH-RFP proteins.

KIN10 using the *pKIN10::KIN10-YFP* transgenic plants. KIN10 displayed the ubiquitous expression pattern in all epidermal cells, and it was mainly localized in the nucleus of guard cells and the smaller cells that were smaller then 200 μm², while mainly distributed in the cytoplasm of larger cells (Fig. 4e–g). There is significantly increased nuclear localization of KIN10 in smaller cells in liquid growth media comapred to that in solid growth media (Supplementary Fig. 12a–e). Whereas, the Thr175 phosphorylation of KIN10 had no significant effect on its subcellular localization in small scale dividing cells and pavement cells (Supplementary Fig. 13a–f). Co-localization analysis with *pKIN10::KIN10-YFP/pSPCH::SPCH-RFP* showed that KIN10 and SPCH were simultaneously distributed in the nucleus of the stomatal-lineage cells (Fig. 4h). These results indicated that KIN10 physically interacted with SPCH in stomatal-lineage cells.

**KIN10 stabilizes SPCH to promote stomatal development.** Next, we investigated whether KIN10 could control SPCH activity by phosphorylation. The in vitro kinase assays showed that MBP-KIN10 had no ability to phosphorylate MBP-SPCH by itself (Fig. 5a). However, in the presence of its upstream kinase GST-SnAK2, MBP-KIN10 was activated to phosphorylate MBP-SPCH (Fig. 5a). Mass spectrometry analysis identified in vitro phosphorylation residues in SPCH by KIN10, including Thr49, Thr50, Ser 51, and Ser52 (Supplementary Fig. 14a, b). To verify that these residues of SPCH are the target sites of KIN10, we replaced them

with alanine or aspartic acid to generate S/T49-52A or S/T49-52D, which we named SPCH-4A or SPCH-4D, respectively. Transformation of the *pSPCH::SPCH-RFP* and *pSPCH::SPCH-4D-RFP* constructs into *spch-4* mutants both rescued stomatal development and defective growth phenotypes of *spch-4* mutants, while SPCH-4A only partially rescued the growth defect phenotypes of *spch-4* mutants, suggesting that these four KIN10-dependent phosphorylated residues are essential for SPCH to control stomatal development (Fig. 5b and Supplementary Figs. 15a–g and 16a, b). Quantification of the fluorescent intensity of *pSPCH::nucGFP* and *pSPCH::SPCH-GFP* revealed that the short-time sucrose treatment had marginal effects on the transcription of *SPCH*, but significantly induced the protein accumulation of SPCH protein (Fig. 5c). Furthermore, we showed that sugar starvation caused by washing away sucrose in liquid medium had no effects on the expression of *SPCH* but significantly reduced the protein levels of SPCH (Fig. 5d). Immunoblot analysis showed that sucrose treatment significantly induced SPCH protein accumulation in *p35S::SPCH-myc* transgenic plants (Fig. 5e). More importantly, SPCH-4A performed weaker protein stability compared to wild type SPCH and SPCH-4D in developing cotyledon epidermis (Fig. 5f, g and Supplementary Fig. 17a, b). Combined with overexpression of *KIN10* leading to the accumulation of SPCH (Fig. 2d–f), these results suggested that KIN10 phosphorylates SPCH, thereby increasing SPCH protein stability and subsequently promoting stomatal development.

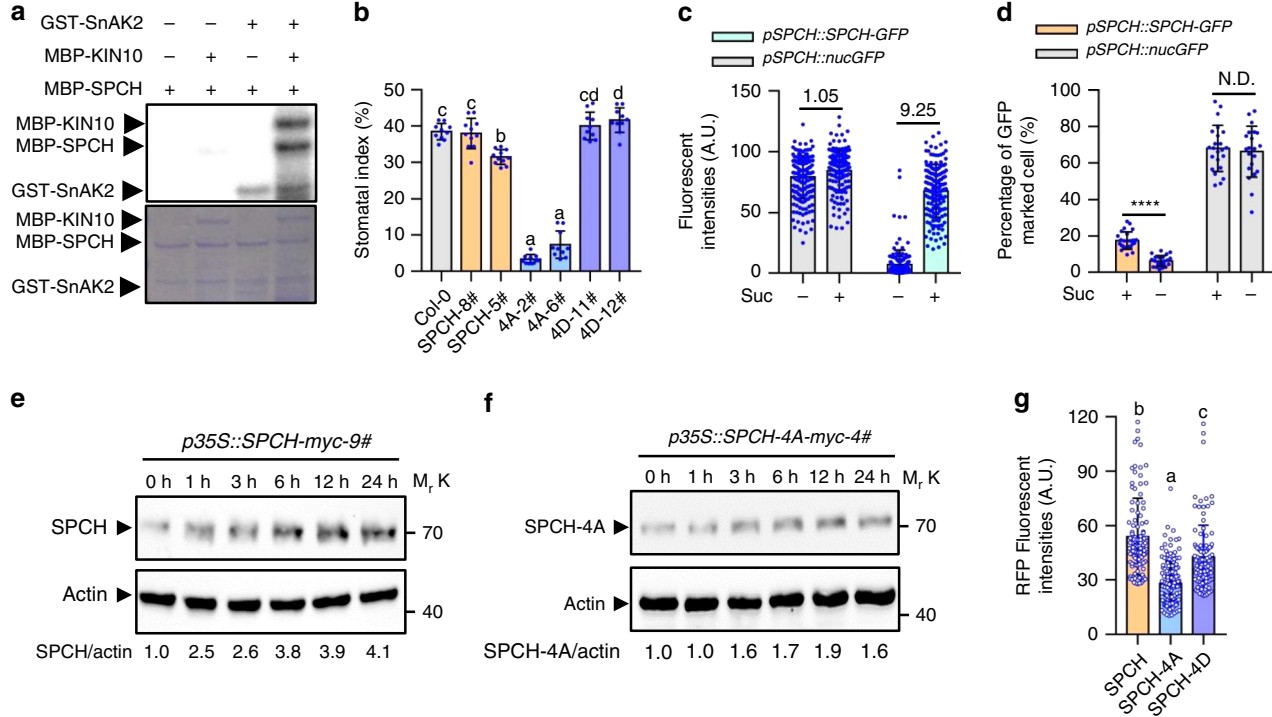

**Fig. 5 KIN10 phosphorylates and stabilizes SPCH to promote stomatal development. a** Phosphorylation of SPCH by KIN10-MBP in vitro. Upper lines showed the gel image of ATP-γ-p32 labeled protein and bottom lines showed the gel image of Coomassie brilliant blue staining. **b** Quantification of stomatal index in wild-type plants, and different SPCH complemented transgenic plants. Seedlings of wild-type plants, *pSPCH::SPCH-RFP/spch-4*, *pSPCH::SPCH-4A-RFP/spch-4* and *pSPCH::SPCH-4D-RFP/spch-4* transgenic plants were grown in ½ MS liquid medium with 1% sucrose for 10 days under long-day condition. Error bars indicate standard deviation (S.D.) (n = 10). **c** Sucrose increased the protein stability of SPCH. Seedlins of *pSPCH::SPCH-GFP* and *pSPCH::nucGFP* were grown in sugar-free liquid medium for 2.5 days, and then treated with or without 1% sucrose for 8 h. The GFP fluorescent intensity was analyzed with more then 150 cells from 10 leaves. Numbers between bars indicated the relative fold changes of average means in the indicated materials. **d** SPCH proteins become unstable when sucrose was removed from liquid medium. Seedlings of *pSPCH::SPCH-GFP* and *pSPCH::nucGFP* were grown in ½ MS liquid medium containing 1% sucrose for 2.5 days then tranferred to sugar-free medium for 6 h. Asterisk between bars indicated statistically significant differences between the samples (Student's *t* test, ****$p < 0.0001$). Error bar indicates S.D. **e, f** Immunoblot analysis of the effects of sucrose on the protein levels of SPCH-myc and SPCH-4A-myc using anti-myc antibody. Seedlings of *p35S::SPCH-myc* and *p35S:SPCH-4A-myc* were grown in sugar-free medium for 3 days, and then treated with 1% sucrose for different times. Actin bands were used as loading control. **g** Analysis the SPCH, SPCH-4A and SPCH-4D protein intensities on abaxial cotyledons of 3-day-old *pSPCH::SPCH-RFP/pKIN10::KIN10-YFP*, *pSPCH::SPCH-4A-RFP/pKIN10::KIN10-YFP* and *pSPCH::SPCH-4D-RFP/pKIN10::KIN10-YFP* transgenic plant by ImageJ software. The red fluorescent signals of SPCH-RFP (n = 114), SPCH-4A-RFP (n = 114) or SPCH-4D-RFP (n = 144) were quantified from more then 100 cells of 5 cotyledons. Error bars indicate S.D. Different letters above the bars indicated statistically significant differences between the samples (ANOVA analysis followed by Uncorrected Fisher's LSD multiple comparisons test, $p < 0.05$).

## Discussion

SnRK1 is a central metabolic regulator of energy homeostasis in plants that is functionally and evolutionarily related to SNF1 in yeast and AMPK in mammals[10,13]. Here, our genetic and biochemical analyses revealed an important role of SnRK1 in plant stomatal development under conditions that are likely associated with mild energy starvation of plants, such as short-day photoperiod or liquid cultures. Sucrose supply induces the accumulation of KIN10 by increasing its translation in the liquid culture condition. KIN10 is expressed in all epidermal cells, but displays the cell-type-specific subcellular location. The nuclear-localized KIN10 is highly enriched in the stomatal-lineage cells. Under certain stress conditions, activated KIN10 phosphorylates SPCH to increase its stability, thereby promoting stomatal development. Thus, our research demonstrates the highly conserved SnRK1 kinase as a positive regulator of SPCH and stomatal development that influences many plant responses to changing environmental conditions (Fig. 6).

SnRK1 has been reported to play central roles in the regulation of plant growth and development in response to energy signals[10,39]. Mutation in the α-catalytic subunit of SnRK1 resulted in defective embryo development as well as in delayed germination and seedling development[13]. In the present study, we demonstrated, through several lines of evidence, that SnRK1 is a positive regulator for stomatal development under certain conditions. First, overexpression of *KIN10* displayed the increased stomatal index, partially under the short-day photoperiod or in the hypoxic liquid growth condition. The loss of function of *KIN10* or *KIN11* resulted in the decreased stomatal index in the liquid medium with 1% sucrose. Whereas, when plants were grown on solid medium without sucrose under 12 h light/12 h dark photoperiod or 4 h light/20 h dark photoperiod, *kin10* mutants showed the higher stomatal index than wild-type plants (Supplementary Fig. 5), which maybe due to the increased the phosphorylated KIN11 proteins in *kin10* mutant comparing to wild-type plants (Supplementary Fig. 18). In addition, the stomatal index of rosette leaves of *KIN10-Ox* was higher, and that of *kin10* mutant was lower than the stomatal index of wild-type rosette leaves. Second, Tre6P has been showed to inhibit the activity of KIN10 by reducing the interaction between KIN10 and SnAK1/SnAK2[23]. Our results showed that Tre6P treatment significantly reduced the stomatal index, while mutation of *TPS1* that contained low levels of Tre6P in plants resulted in the increased stomatal index comparing to wild-type plants. Consistent with this, the transgenic plants of *pKIN10::*

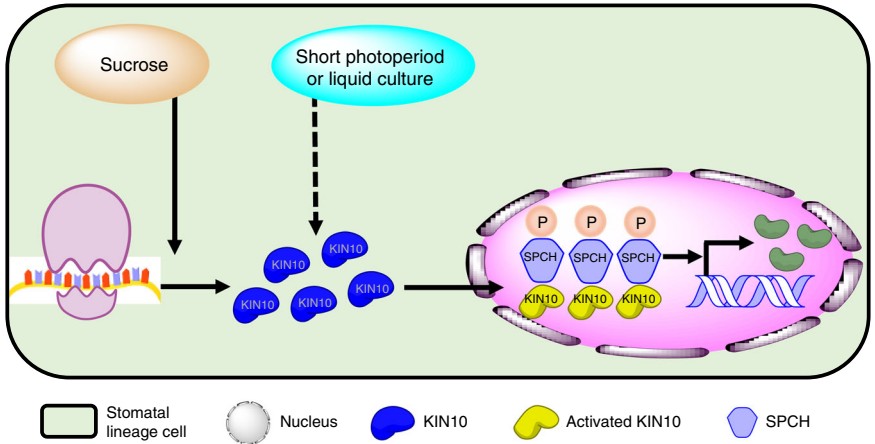

**Fig. 6 A model for KIN10-promoted stomatal development through phosphorylation and stabilization of SPCH.** When plants were grown under conditions that are likely associated with mild energy starvation of plants, such as short photoperiod or liquid cultures, sucrose supply induced the KIN10 protein accumulation by increasing its translation. KIN10 displayed the cell-type-specific subcellular location pattern in the epidermal cells of leaves. The nuclear-localized KIN10 in stomatal-lineage cells phosphorylated and stabilized SPCH to promote stomatal development. Thus, the combination of multiple regulatory mechanisms controls the activity of KIN10 and thereby optimizes stomatal development.

$KIN10^{T175D}$-*myc*, $pKIN10::KIN10^{T175E}$-*myc* and *p35S:SnAK2-GFP* showed the increased stomatal index. Third, the nuclear-localized KIN10 was highly enriched in the cells smaller then 200 μm² that may belong to the stomatal-lineage cells, which was further verified by the colocalization of KIN10 and SPCH in nucleus of these cells. Finally, KIN10 physically interacted with SPCH to phosphorylate and stabilize SPCH. Mutation of the KIN10 phosphorylation sites on SPCH greatly reduced the stability of SPCH. Further genetic analysis showed that mutated *SPCH* cannot rescue the growth defect of *spch-4* mutants. Together, these results demonstrated that KIN10 promotes stomatal development through stabilization of the transcription factor SPCH.

In this study, we found that KIN10-YFP fluorescent signals were present in all epidermal leaf cells, but the nuclear localization signals of KIN10-YFP were significantly enhanced in the small cells that might belong to the stomatal-lineage cells, leading to phosphorylation of SPCH by KIN10 in the nucleus to increase SPCH protein stability and promote stomatal development. The nuclear/cytoplasmic ratios of KIN10-YFP in stomatal-lineage cells of plants grown in liquid growth media were significantly higher than that grown on solid media in the presence of 1% sucrose, indicating KIN10 activity in stomatal-lineage cells is more sensitive for use of liquid growth condition. Consistent with this, plants grown in liquid media with 1% sucrose showed the increased stomatal index comparing to the plants grown on solid media with 1% sucrose, which may be due to the increased KIN10 activity by the mild hypoxia of the liquid culture.

Phosphorylation of a highly conserved threonine residue in the T-loop is essential for the catalytic activity of AMPK, SNF1, and SnRK1 in mammalian, yeast and plant cells, respectively[11]. The phosphorylation of KIN10 has been reported to be activated by submergence[40]. However, submergence treatment had no significant effect on the transcriptional level and total protein level of KIN10, although it significantly increased the phosphorylation of KIN10 at the T-loop conserved threonine residue[40]. In this study, a liquid culture system was used to assess stomatal index and KIN10 activity. The significant promotion of stomatal development in plants grown in liquid medium with sucrose might be due to the activation of KIN10 by mild hypoxia of the liquid culture, similar to submergence treatment. Accordingly, exogenous sucrose supply had no obvious promoting effects on stomatal index of plants grown on solid medium under the 16 h light/8 h dark photoperiod

condition, but it significantly increased the stomatal index of plants grown on solid medium under the 4 h light/20 h dark photoperiod condition. This may be due to the activiation of KIN10 under prolonged darkness[13]. These results indicated that sugar induces the accumulation of KIN10 by enhancing its translation, while liquid culture or prolonged darkness might activate KIN10 through inducing its phosphorylation. Subsequently, the activated and nuclear-localized KIN10 in stomatal-lineage cells phosphorylated and stabilized SPCH to promote stomatal development. Overall, the combination of multiple regulatory mechanisms controls the activity of KIN10 and thereby optimizes stomatal development.

The promotion of KIN10 on stomatal development may occur when plants encounter sunny days after consistent cloudy weather for many days, or when plants recover from the flooding stress. The cloudy weather or flooding stresses resulted in the energy starvation of plants. When plants encounter the sunny days again, plants restart photosynthesis, and produce sugar to induce the accumulation of KIN10. The activated KIN10 phosphorylates and stabilizes SPCH to promote stomatal formation and then increase the ability of plant photosynthesis and carbon assimilation, thus forming a positive feedback loop to help plants recover from stress. Taken together, our research not only establishes the highly conserved SnRK1 kinase as a positive regulator for stomatal development, but also provides a tractable system for investigating how environmental stresses integrate with metabolic signals to modulate stomatal development.

## Methods

**Plant materials and growth conditions.** Arabidopsis ecotype Columbia (Col-0) was used as the wild-type except where indicated. Plants were grown in a greenhouse with white light at 100 μmol m⁻² s⁻¹ and relative humidity of 50% under a 16 h light/8 h dark cycle at 22 °C for general growth and seed harvesting. Mutants and transgenic plants used in this study were *spch-4*[36], *p35S::KIN10-HA*[13], *pSPCH::nucGFP*[4], *pSPCH::SPCH-GFP*[4], *pMUTE::MUTE-GFP*[36], *pBASL::BASL-GFP*[35], *p35S::KIN10-Myc*, *pKIN10::KIN10-YFP*, $pKIN10::KIN10^{T175D}$-*YFP*, *p35S::SnAK2-GFP*, *pSPCH::SPCH-RFP*, *pSPCH::SPCH-4A-RFP*, and *pSPCH::SPCH-4D-RFP*. The T-DNA insertion mutants of *GABI_579_E09* (*kin10/snrka1*) and *WiscDsLox384F5* (*kin11/snrka2*) were in Col-0 background and ordered from The European Arabidopsis Stock Centre. The *GABI_579_E09* mutant has been used the knock out mutant of *KIN10* to determine its functions in hypocotyl elongation, root growth and stress response in plants[28,41,42]. The mutant *WiscDsLox384F5* has not been used in previous study. Our RT-PCR analysis showed there are some amount of *KIN10* transcript in *kin10* mutant, but no *KIN11* transcript in *kin11* mutant. Immunoblot further confirmed that no significant amount of KIN10 protein and KIN11 protein could be detected in *kin10* and *kin11* mutants, respectively

(Supplementary Fig. 4a–c). In order to analyze the effect of different types of sugar on stomatal development, seedlings of wild-type and indicated mutants were grown in 1/2 MS liquid medium containing sucrose or different types of sugar for 10 days under 16 h photoperiod. For protein extraction, seedlings of the indicated transgenic plants were grown in sugar-free half-strength MS medium for 3 days and then treated with or without sucrose, CHX for different times.

**Plasmid constructs and transgenic plants.** Full-length cDNA of KIN10, SPCH, SnAK1, and SnAK2 without stop codon were amplified by PCR and cloned into pENTR™/SD/D-TOPO™ vectors (Thermo Fisher), and then recombined with destination vector p1390-MH (p35S::X-Myc-His), pX-YFP (p35S::X-YFP), pGAL4BDGW (GAL4BD-X), pGAL4ADGW (GAL4AD-X), pDEST15 (N-GST), and pMAL2CGW (N-MBP). The promoters and genomic DNA of KIN10 and SPCH were amplified by PCR and cloned into pENTR™/SD/D-TOPO™ vectors (Thermo Fisher), and then recombined with destination vector pEG-TW1 (Native promoter::YFP) and pAL-1296R (Native promoter::RFP) to generate pKIN10::KIN10-YFP and pSPCH::SPCH-RFP, respectively. The constructs of KIN10-T175D, SPCH-4A and SPCH-4D were performed using the quick-change site-directed mutagenesis kit (Stratagene). Oligo primers used for cloning are listed in Supplementary Table 1. All binary vector constructs were introduced into Agrobacterium tumefaciens (strain GV3101), and transformed into Col-0 plants by the floral dipping method.

**Stomatal quantification.** Arabidopsis seedlings for stomatal quantification were cleared by decolor solution (75% ethanol 25% acetic acid) for 6 h or overnight. Cotyledon was captured and dried on paper, then submerge into Hoyer's solution until cotyledon turning transparent completely for microscope observation. In total, 700 µm × 550 µm images were captured per cotyledon from central regions of abaxial leaves. For cotyledon with small size, 350 µm × 250 µm images were captured. Guard cells, dividing small cells and pavement cells were counted for stoma index calculation. Abaxial epidermises of cotyledons pictures were edited in Adobe Illustrator software based on DIC picture. The size of epidermal cells was calculated by ImageJ software.

**Microscopy.** Confocal microscopy was performed using an LSM-700 laser scanning confocal microscope (Zeiss). Stomatal phenotype analysis was carried out with modified Pseudo-Schiff Propidium Iodide (mPS-PI) staining method and imaged by using the following settings: excitation 515 nm and emission 620–670 nm. Seedlings of wild-type and various mutants for analysis of stomatal phenotype were grown in ½ MS liquid medium or on ½ MS solid medium for 10 days under the 16 h light/8 h dark growth conditions as indicated in each figure legends. Quantitative analyses the fluorescent intensities of pSPCH::nucGFP, pSPCH::SPCH-GFP, pMUTE::MUTE-GFP, pBASL::BASL-GFP, pKIN10::KIN10-YFP and pSPCH::SPCH-RFP were performed using ImageJ software as previously described. Images were captured at 488 nm and 515 nm laser excitation and 500–530 nm and 620–670 nm for GFP and RFP, respectively. The rBiFC assays were performed in Nicotiana Benthamiana leaves and imaged by using the following settings: excitation 488 nm and 515 nm, emission 500–530 nm and 620–670 nm, respectively. Laser intensity setting was restricted lower than 30%. Master gain value was not more than 850.

**Ratiometric bimolecular fluorescence complementation assays.** Full-length cDNA of KIN10, SPCH, and SnAK1 were amplified by PCR and cloned into the pDONR221-P1P4, or pDONR221-P3P2 vector, respectively, using the BP recombination reaction (Invitrogen). The cDNA of KIN10, SPCH, and SnAK1 were recombined into destination vector pBiFCt-2in1-NN[43] to generate the expression constructs with p35S::nYFP-KIN10-p35S::RFP-p35S::cYFP-SPCH, or p35S::nYFP-KIN10-p35S::RFP-p35S::cYFP-x. Agrobacterial suspensions containing these constructs were injected into tobacco leaves. The transfected plants were kept in the greenhouse for 36 h at 22 °C. Fluorescent signals were visualized by using the LSM-700 laser scanning confocal microscope (Zeiss) and the signal intensities of YFP and RFP were determined by ImageJ software.

**Pull-down assays.** KIN10 fused to MBP and SPCH fused to GST were purified from bacteria using amylose resin (NEB) or glutathione beads (GE Healthcare), respectively. In total, 1 µg of GST-SPCH was incubated with 1 µg MBP or MBP-KIN10 in pull-down buffer (20 mM Tris-HCl pH 7.5, 100 mM NaCl, 1 mM EDTA) at 4 °C for 1 h, and the beads were washed five times with wash buffer (20 mM Tris-HCl pH 7.5, 300 mM NaCl, 0.1% Nonidet P-40, 1 mM EDTA). The proteins were eluted from beads by boiling in 50 µl 2 × SDS sample buffer and separated on 8% SDS-PAGE gels. Gel blots were analyzed using anti-MBP (NEB, 1:5000 dilution) and anti-GST antibodies (Santa Cruz Biotechnology, 1:3000 dilution).

**Coimmunoprecipitation assays.** Plants expressing both p35S::KIN10-YFP and p35S::SPCH-myc or plants only expressing p35S::SPCH-myc were grown in half-strength MS medium with 1% sucrose for 5 days and then harvested to perform coimmunoprecipitation assays. Harvested tissues were ground in liquid nitrogen, homogenized in immunoprecipitation buffer containing 20 mM HEPES-KOH (pH 7.5), 40 mM KCl, 1 mM EDTA, 0.5% Triton X-100, and 1× protease inhibitor (Sigma

Aldrich). After centrifugation, the supernatant was incubated with GFP-Trap agarose beads (Chromotek) at 4 °C for 1 h, and the beads were washed four times using wash buffer (20 mM HEPES-KOH pH 7.5, 40 mM KCl, 1 mM EDTA, 300 mM NaCl, and 1% Triton X-100). The proteins were eluted from the beads by boiling with 2 × SDS sample buffer, analyzed by SDS-PAGE, and immunoblotted with anti-YFP (1:5000 dilution) and anti-myc (Sigma Aldrich, Cat: M4439, 1:5000 dilution) antibodies.

**In vitro kinase assay and phosphopeptide analysis.** MBP-KIN10, MBP-SPCH and GST-SnAK2 proteins were expressed and purified from Escherichia coli. MBP-SPCH was incubated with MBP-KIN10 and/or GST-SnAK2 as indicated in the kinase buffer (25 mM HEPES, pH7.4, 10 mM MgCl₂, 50 mM KCl, 1 mM DTT, and 30 µM cold ATP) containing [γ³²P] ATP (10 µCi). The reaction was stopped by addition of 10 µl of 5 × LDS buffer. Proteins were resolved by 10% SDS-PAGE. After nonradioactive in vitro kinase assays, proteins were digested with trypsin and endoproteinase Asp-N. The digested peptide mixtures were injected into Q Exactive HF hybrid quadrupole-Orbitrap mass spectrometer (ThermoFisher Scientific) for Mass spectrometer analysis. The phosphorylated residues in MBP-SPCH were identified by Maxquant software.

**Reporting summary.** Further information on research design is available in the Nature Research Reporting Summary linked to this article.

## Data availability

All data in this study are available in the main text or the Supplementary materials. Source data are provided with this paper.

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

## Acknowledgements

We thank Prof. Dominique C. Bergmann and Prof. Jie Le for providing the seeds of *pSPCH::nucGFP, pSPCH::SPCH-GFP, pMUTE::MUTE-GFP, pBASL::BASL-GFP* and *spch-4*. We appreciated Prof. Jen Sheen provided valuable comments to improve paper. And we also thank Mrs Xiaotian Duan for paper revision. We thank Haiyan Yu and Xiaomin Zhao from the Analysis and Testing Center of SKLMT (State Key Laboratory of Microbial Technology, Shandong University) for assistance with the laser scanning confocal microscopy. This work was funded by Shandong Province Agricultural Improved Variety Project (2019LZGC-015), by the National Natural Science Foundation of China (grant nos. 31800211, 31870262, 31670284, and 31600199), by Shandong Province Natural Science Foundation (grant nos. ZR2019ZD16, JQ201708, and ZR2018ZC0334), and by the Fundamental Research Funds of Shandong University (2017HW009).

## Author contributions

C.H. and M.Y.B. together designed the experiments. C.H. performed statistical analysis of stomatal index of wild type and various mutants in our experimental conditions, microscopy analysis, western blot, subcellular location analysis, pull down, rBiFC, CoIP. W.S., Y.L., Y.T., M.F., L.W., and Y.Q. generated KIN10 related mutants and transgenic plants, *pKIN10::KIN10-YFP, p35S::KIN10-YFP/p35S::SPCH-myc* and *pSPCH::SPCH-RFP*. C.H. and Z.D. performed performed the kinase assays and Mass spec analysis. O.L and G.J. provided the critical discussion on the work. C.H. performed all other experiments. C.H. and M. B. wrote the paper.

## Competing interests

The authors declare no competing interests.
