## [Peer Review File · Nature Communications]

Reviewers' comments:

Reviewer #1 (Remarks to the Author):

This manuscript describes how physiological levels of sucrose promote stomatal development, and implicates the SnRK1 kinase in this process through the direct phosphorylation of the SPEECHLESS transcription factor. The subject is of very high interest, adding up to the increasing list of regulatory roles of sugars and of the SnRK1 kinase in plant developmental processes. However, there are several major issues regarding experimental design, controls and interpretations that should be addressed before this manuscript can be considered for publication. One major concern relates to the significance of the differences shown in the stomatal index of the KIN10 overexpressor lines vs WT when the degree of variation in this parameter for WT seedlings is very large across experiments (Figs. 1-2). Another one relates to the experiments where the effect of sucrose on KIN10 accumulation is assessed (Fig. 3). These experiments lack important osmotic controls. A third one relates to the overstatement that phosphorylation of KIN10 plays a major role in this process (Fig. 4). And a fourth one related to missing controls in many experiments.

In detail:

Figure 1

- 1) MAJOR CONCERN: There is large variation in stomatal index for the WT from one graph to the other. For example, compare the Col control in panel b (~ 29%) to the one of 0% sucrose in panel d (~20%). Considering such differences between WTs in different replicate experiments the differences between WT and OE in Fig. 2b seem negligible. Are the compared genotypes always grown in the same plates or in separate plates? Given the differences shown for the WT I believe all compared genotypes should be grown in the same plate, but this is not described in the methods. What can explain this variation?
- 2) Panel (b). What is a ' ?
- 3) In panel (a) there is a difference between no sucrose and sucrose (3rd and 4th columns, c,d), but in panel (b) there are no differences. Why?
- 4) Panel (b). The starch mutants should in principle have differential sucrose accumulation already in the no sucrose plates and one would expect that they have different stomatal indexes in this condition. However, they seem all equal. Also, in the presence of exogenous sucrose (3% sucrose plates) the *sex1* mutant could be expected to be "rescued" and be similar to the WT, but it is not. I think this graph is just confusing and does not provide much information.
- 5) Legend: express the sugar concentrations of panel c also as % (in addition to mM) to be able to compare with all other results in the paper.

Figure 2

- 1) Panel f. The differences between KIN10 OX and WT for SPCH-GFP accumulation are much higher than the differences in stomatal index shown for these backgrounds in panel b. This should be at least mentioned and discussed.
- 2) Panel f. Which KIN10-OX line is used here?

Figure 3

- 1) MAJOR CONCERN: In these experiments the seedlings are probably experiencing an osmotic shock, since they are not grown in sucrose for a few days and all of a sudden they are exposed to 1% sucrose. An osmotic control should be added to show that this is indeed sucrose specific. These controls are provided for the experiments of Fig. 1 where seedlings are grown from the very beginning in these conditions, but not for the ones in this figure where sucrose is provided as a short-term treatment.
- 2) MAJOR CONCERN: There is definitely more KIN10 in the sugar treated seedlings. However:
 - That does not mean that there is more activity (quite the contrary, as often KIN10 accumulation is observed under conditions where it is inactive)
 - This could be due to osmotic stress rather than to the presence of sugar, as pointed out in point 1)

➤ So, in brief, in the absence of a proof that there is indeed enhanced KIN10 activity in these conditions, the authors should limit their comments to their observation that there is more protein accumulation

3) Why is there a duplet in the p35S:KIN10-Myc line in panels (c) and (d)?

4) Sometimes the KIN10 blots are indicated to have been done with IP:ed KIN10 (Figure 4) and sometimes not (Figure 3). Which one is it?

Figure 4 (and associated text)

1) MAJOR CONCERN: The claim that sugar controls stomata development by controlling the levels of phosphorylated KIN10 is a strong overstatement (e.g. lines 245-247). The enhanced T-loop signal is fully explained by the higher KIN10 accumulation. Only in the very high sugar concentrations (3-5%) lower phosphorylation ratios can be observed, but given that the focus are the conditions where higher stomatal index is observed (1% sucrose), the effect of 3-5% sucrose on Kin10 phosphorylation is irrelevant.

2) The quality of the P-AMPK westerns is quite low, and if T-loop phosphorylation is to be made a strong point, they should be improved.

3) Panel (e). BiFC controls with empty vectors are missing. Most importantly, without showing by Western that both SnAK and KIN10 accumulate similarly in 1% and 5% sucrose one cannot conclude that the decreased BiFC signal in 5% is due to decreased interaction (as opposed to decreased accumulation of either one or both proteins).

4) Panel (h). Is there indeed higher T-loop phosphorylation in the SnAK OX lines? That would support the interpretation of the authors that the effect of sucrose on stomatal development is mediated by an increase in KIN10 phosphorylation.

Figure 5

1) Panel (a). Missing the control of at least BD-SPCH with empty AD vector

2) Panel (d). It would be more correct to see whether SPCH interacts with GFP alone, as for the MBP in vitro pull down experiments. What are the plants used for these experiments? They are double overexpressors? They are not described anywhere.

3) Panel (g) Wouldn't you expect that the D mutation causes higher stomata index, just as what happens in the KIN10-OE lines? At the very least this should be mentioned and discussed.

4) Given all the different crosses and transgenics the authors have, it would be good to see some in vivo evidence that SPCH phosphorylation is indeed altered in the KIN10 OE background. They could do a Phostag for this TF.

Material and methods

1) They refer to the kin10 mutant as kin10-1, which it is not. I believe this is kin10-3; also the kin11 allele is not kin11-1 and to my knowledge this has not been previously characterized, so a basic characterization should be performed if this is to be included.

2) MAJOR CONCERN: There is no explanation on the metabolomics analyses, so it is hard to assess whether the T6P measurements are reliable or not. Often LC-MS is used and this cannot separate T6P from other disaccharide-monophosphate isomers.

3) What are the light conditions? The plants in Suppl Fig. 1 look quite atrophied and with so elongated petioles? Is this effect on stomatal index only observed under very low light?

4) There is no information of any of the lines generated in this study. A basic molecular characterization should be provided in the supplementary material and a precise description of the constructs employed for their generation. Also the lines should always be referred to in the same manner throughout the text and in the figures. Indicate which variant of KIN10 is expressed in the KIN10 lines (which gene model).

Other minor issues:

- Line 100. I would not call dramatic an increase from 11% to 15%.
- Line 288. "Wild type plants grown in medium with different concentrations of sucrose". Is it grown or treated for 4h, as said in the legend?
- Lines 473-474. "Overexpression of the upstream kinase SnAK2 and KIN10T175D..." Is the latter

indeed an overexpressor? In the figure it is marked as ProKIN10...

- What is the KIN10-OX used in Suppl Figure 3?
- Sugars have been mostly shown to inactivate KIN10, so the issue of sucrose activating KIN10 should be discussed in this context
- The manuscript would benefit from language editing

Reviewer #2 (Remarks to the Author):

The authors report the novel and important finding that the SnRK1 kinase mediates sugar-regulated stomatal development in Arabidopsis cotyledons. The study provides strong genetic and molecular evidence that KIN10 (a SnRK1 catalytic subunit) is a positive regulator of stomatal development by directly interacting with and phosphorylating and stabilizing the SPCH transcription factor. The science is of high quality and the results are carefully presented. SnRK1 is generally established as a metabolic stress sensor, which is activated by carbon starvation and repressed by increased sugar levels. The latter is also observed in this study with higher sucrose levels (and associated increased T6P levels) that inhibit stomatal development. The observation that lower levels of sucrose (1%) increase SnRK1 activity by increasing KIN10 translation to promote stomatal development is surprising. The manuscript therefore requires more discussion of these apparent discrepancies with previous findings on SnRK1 regulation.

More specific remarks and suggestions:

- Line 7-11: The abstract could be clearer on what was demonstrated before by other groups (T6P inhibition of KIN10 interaction with its upstream kinases) and what is found in the present study.
- These studies are done with cotyledons. Why? Would similar results be obtained with true (sink) leave? Wouldn't this be physiologically more relevant?
- Line 96: A prolonged darkness of 7 days seems extreme. Are similar effects seen after a few days of darkness, which should already cause a significant starvation condition?
- Figure 1: In (a) the effect of 3% sucrose appears much smaller than in (d). What could be the reason for that? Is there a lot of variation between experiments? In (b) the stomatal index does not appear to be affected in the absence of exogenous sucrose, as would be expected with these mutations. Why have G6P and (especially) FBP been tested specifically? Are such sugar-phosphates taken up?
- Figure 2: Why is a different representation of the statistical differences used here (similar multiple comparisons test)? In (c) the stomatal index of the kin10 mutants appears higher than WT in the absence of sucrose. Is this (statistically) relevant?
- Line 196: Not sure the 'activity' of SnRK1 is tested here.
- Figure 3: SnRK1 activity is not really quantified. TPS5 is a known sugar upregulated gene, but some of the (many) established SnRK1 targets should be tested in these conditions. I think this is very important to confirm that SnRK1 activity is indeed up in these conditions! T-loop phosphorylation does not always correlate well with activity. In (b) expression is tested only after 4h, while (c) shows that protein levels are not upregulated yet at that timepoint.
- Figure 4: The authors correctly conclude that there is 'an accumulation of phosphorylated KIN10' with 6h of low levels of sucrose. However, it is not clear whether the T-loop phosphorylation status (i.e. the ratio of phosphorylated versus non-phosphorylated KIN10) changes. Apparently not, indicating that protein levels go up without changing T-loop phosphorylation status, still resulting

in more total active SnRK1. Upon addition of higher inhibitory sucrose levels, however, this ratio clearly decreases as expected. This could maybe be discussed better. The figure 4 legend title suggests a causal link between T6P and T-loop phosphorylation, which was indeed shown before, but cannot be concluded from these data (correlation). I would change the title. T6P levels also already go up significantly after just 4h of 1% sucrose. Analysis of stomatal development in a *tps1* mutant (containing a rescue construct to bypass embryo-lethality) could be interesting.

- No methods information is included for sugar and sugar-P level quantification; especially for T6P, this is critical.

- Figure 5 is very convincing (complementary, independent ways to show physical interaction and phosphorylation). Could SPCH stabilization (h) be confirmed using immunoblotting?

- Discussion/Line 404: As mentioned above, I am not sure the prolonged darkness is reflecting a physiologically relevant starvation condition. Just like high sugar supply feedback inhibits stomatal development, I would expect mild starvation stress to stimulate (through SnRK1) stomatal development and C uptake.

- Line 468: I would be careful making statements about T-loop phosphorylation status correlating with stomatal development. An 'accumulation of phosphorylated KIN10' with low sucrose concentrations (Line 470) is correct.

Minor remarks:

- Line 36: SnAK1 and SnAK2 are GRIK2 and GRIK1
- Line 43: Delete 'in plants'
- Line 45: How is this a feedback loop?
- Line 172: 'containing'
- Some other typo's

Reviewer #3 (Remarks to the Author):

Review - Han et al. 2019 submitted to Nat Comm

The author present evidence that the sugar-sensitive SnRK1 kinase regulates SPCH stability in a sugar concentration dependent manner to regulate stomatal development. They provide evidence that low excess sugar levels increase stomatal index (SI) while high excess sugar level does not. Overexpressing and mutating the SnRK1 subunit KIN10 enhances and attenuates the sugar effect on SI, respectively. The authors show that sugar accumulates KIN10 through regulating translation and not transcription. Interestingly, the KIN10 protein level increases linearly with sugar concentration but the active phosphorylated form of KIN10 is sensitive to too high sugar levels likely due to an excess of T6P, which scales linearly with sucrose. Finally, the authors present evidence that KIN10 indeed interacts with and phosphorylates SPCH at specific residues leading to stabilization of SPCH and, consequently, to a higher SI.

While much of the data presented is sound, the story built in a logical and easy-to-read manner, I would like to see the following issues addressed:

Major concerns

1. solid MS plates vs. liquid MS cultures

For their initial findings of how sucrose induces a higher SI, the authors use plate grown plates (eg. 1A, B) but most of their other quantifications stem from liquid MS culture grown plants. While I

understand that these are much easier to deal with when performing protein work, I am a bit worried about how submergence of plants affects stomatal development and the sensitivity to sugar b/c gas exchange is obviously highly limited. On plate in Fig. 1A the change of SI from 0% to 3% sugar is from 28% to 32%, whereas in Fig. 2b and c in liquid culture SI changes from less than 25% to more than 40% in 1% sucrose. Therefore, I would like to see how solid vs. liquid culture affects stomatal development (ie. SI) with 1% sucrose and w/o sucrose for wildtype, KIN10-OE and kin10 mutant plants.

2. sugar-dependency

Throughout the manuscript the authors are a bit inaccurate regarding the effects they see, which are mostly sugar dependent (eg. line 168-170, 107-108, 177-179) whereas in other places it is clearly specified (eg. line 172). I would like to ask the authors to carefully go through the text and figure legends again and clarify where appropriate.

Along similar lines it is sometimes unclear if the data derives from sugar-treated plants or not. Therefore, please specify this for each subpanel in the figure legends and in addition add the sugar concentration above the graphs that do not compare different sucrose levels throughout the manuscript like in Fig. 4G and H.

In addition, the authors keep using the expression "sugar starvation" (eg. in Fig. 6 and the discussion). Except when grown in the dark, this expression is not true. 0% sugar is rather "no sucrose excess. Fig. 6 and throughout the text should rather say no sucrose, low sucrose and high sucrose. For example line 378 claims that washing out sucrose leads to sugar starvation. But the plants still experience light and can make their own sucrose.

Along these lines, I also don't really understand how the prolonged darkness experiment fits in. This is a confounding effect of light regulation on stomatal development together with a potential sugar starvation signal. Please elaborate or remove.

3. T6P

The authors show correlative evidence that T6P might inhibit the interaction of SnAK and KIN10 in high excess sugar. Accordingly, growing plants with T6P and 1% sucrose should decrease the SI much like the kin10 mutant. I would like to see this experiment to support their claim.

4. nuclear localization of KIN10-YFP

One of the key points of this paper is that KIN10-YFP is present in the nucleus in stomatal cells. Since the authors have the KIN10-T175D cloned it would be nice to see if the phosphomimic KIN10 is preferentially targeted to the nucleus compared to KIN10-YFP.

5. original blots

Since many of the findings relate on quantifications of Western blots, I would like to see all the original blots as supplementary figure(s).

Medium concerns

6. Activation of SnRKs

In animal and yeast SnRKs get activated through energy depletion but this seems not to be the case here. I would like an extended discussion on this and also how the authors think that this finding translates in real nature. If there is excess sucrose in the plant, why would the plant want to build more stomata? More stomata means more water loss so why making more stomata if there is already photoassimilates? In the manuscript the authors simply use generic terms like "finetuning of sugar and stomata to efficiently regulate photosynthesis" which in my opinion is not enough. In line 435 for example the authors talk about water-use efficiency, which I think is not appropriate. More stomata, more water loss, so sucrose induces stomatal pores to provide more carbon for photosynthesis thereby reducing their water-use efficiency under suboptimal light conditions (i.e. dawn and dusk, clouds etc.)

7. Discussion

The last sentence in the discussion is confusing. How would the repression of SnRK1Beta1 (what is this by the way?) reduce the nuclear translocation of KIN10? Please elaborate and clarify

8. Figures:

Fig. 2a-c. I would like to see statistical analysis of the differences in 0% Sucrose conditions. There seems to be a difference between Col-0 and kin10 and kin11 also sucrose independently. If this is the case then this observation should be discussed.

Fig. 3d. Why are there two myc bands in the upper row?

Fig. 4c. Why did the authors perform this experiment with the KIN10-HA line from Ler rather than with the myc line in Col-0? Please explain or repeat the experiment with KIN10-myc in Col-0.

9. Methods:

Since some of the protein level quantifications are done by confocal microscopy I would appreciate if the authors include a "microscopy" section including type of microscope, settings (laser intensity, gain etc.). This paragraph should also contain details regarding the BIFC in Figure 4e.

10. Figure legends:

Throughout the figure legends the authors should indicate the number of replicates (both individuals and fields-of-views that were quantified).

Minor concerns

114: F6P has a significant effect (Fig. 1C), please correct.

273ff: This is not a complete sentence, please change.

295ff and 319ff. BiFC is introduced in line 319 rather than in line 295. I would suggest to introduce it in line 295ff

325ff: in the text it says KIN10-myc and SPCH-YFP, in Figure 5D it is vice versa. Please correct.

375ff. The authors mention qRT-PCR, but the figure shows quantifications with a transcriptional reporter. Please change.

391ff. The authors claim here that they show dynamic regulation of SnRK1 activity to different environmental conditions. It is, however, only sucrose excess. Please correct.

Figure 6 Figure legend: This legend in particular has quite some grammar mistakes and typos. Please correct.

475; stabilisation rather than activation of SPCH.

508: please indicate the ecotypes for kin10 and kin11

Reviewer #4 (Remarks to the Author):

This manuscript investigates for the first time what are the molecular mechanisms underlying stomatal development promoting effect of sucrose. Authors combine genetic and thorough biochemical analyses to tackle this question. Authors show that sucrose promotes KIN10 translation, which in turn phosphorylates SPEECHLESS (SPCH), master regulator of stomatal development. This leads to stabilization of SPCH protein and promotion of stomatal developmental pathway. Authors identify set of KIN10 phosphorylation sites in SPCH and show that phosphorylation of these sites is sufficient to enhance SPCH stability. Authors show that KIN10 and SPCH physically interact, are colocalized in the stomatal lineage, and both stability of SPCH and amount of phosphorylated KIN10 are inhibited in high sucrose conditions. Authors show that sucrose promotes accumulation of trehalose-6-phosphate (tre6P) in dose dependent fashion, which is known to prevent interaction between KIN10 and kinases acting upstream of KIN10 thereby inhibiting KIN10 activity. KIN10 has been shown to localize to nucleus as a response to metabolic stress. Surprisingly, KIN10 is nuclear-localized in stomatal lineage cells. Authors suggest that

SPCH might directly repress SnRK1 β 1 in stomatal lineage cells, which inhibit nuclear localization of KIN10, hypothesis supported by previous studies. Authors suggest a model where upon low and intermediate sucrose conditions KIN10 translation is enhanced by sucrose. Kinases upstream of KIN10 phosphorylate KIN10, which in turn phosphorylates SPCH thereby stabilizing it. In high sucrose condition tre6P prevents interaction between KIN10 and its upstream kinases and thus, further increase in SPCH levels and stomatal numbers.

Proposed mechanism would allow fine-tuning of the SPCH stability and lead to optimization of stomatal numbers depending on sucrose availability during leaf development. Such a mechanism would allow coordination of gas-exchange ability with photosynthesis status, two crucial processes for plant growth and development and for biomass production. The work presented and model constructed are novel and most of the data presented is convincing and impressive. However, there are some concerns related to stomatal developmental phenotypes and KIN10/SPCH coexpression data. These concerns are detailed below.

1. Authors use two sucrose metabolism mutants; *sex1* (high starch/low free sugar content) and *pgm* (low starch content/high free sugar content) to show that sucrose-induced changes in stomatal development are modified in the conditions with altered sucrose/starch levels. Stomatal indexes in these mutants are similar to wild-type if grown without sucrose supplementation. Stomatal development in *sex1* mutant is less sensitive for external sucrose whereas *pgm* mutant shows increased sensitivity to sucrose treatment. It is puzzling that dramatically altered cellular sucrose levels in these mutants do not lead to changes in the stomatal development but external sucrose treatment does. Please clarify this aspect.

2. Authors examine how different sugars modify stomatal index by growing seedlings for 10 days in liquid growth media supplemented with various sugars. However, condition without any added sugars is not included into this experiment. Authors state "Sucrose exerts the most obvious positive effects on the development of stomata". Data on wild-type grown in liquid growth media without any sugars for 10d should be added to this experiment in order to be able to evaluate effect of different sugars on stomatal index.

3. Authors show that wild-type plants show dramatic reduction in total stomatal numbers when grown 10 days in liquid growth conditions without sucrose compared to sucrose supplemented plants (Figure S1a-f). Seedlings shown in Figure S1a and S1d show smaller cotyledon size and tiny true leaves suggesting that growth has been severely compromised in these growth conditions. Since presence or absence of sucrose alone does not cause such a differences on plant growth (see for example Hanson et al (2001) *Plant Molecular Biology* 45: 247–262; figure 3) it seems likely that the observed differences are caused by combination of liquid growth condition related stress and presence/absence of sucrose. Growing plants in liquid growth media may expose them to mild hypoxia and it clearly causes stress by severely reducing seedling growth (Figure S1a-f). Use of liquid growth media might be justified when performing large scale experiments such as biochemical assays, however, analysis of developmental phenotypes should be done as little stress causing conditions as possible. Cho et al. (2016) (*J Ex Bot* 67, 2745–2760) showed that submergence is sufficient to activate KIN10 and this causes broad changes in the phosphorylation status of the target proteins of KIN10. Authors should discuss previously mentioned findings on submergence related stress and KIN10 activation and evaluate what implications it may have on their data on stomatal development.

4. Authors state (rows 157-161): "To examine how sugar regulates stomatal development, we analyzed the stomatal phenotypes of key components of different sugar signal transduction pathways. We found that several sugar signalling pathways are involved in sugar-mediated stomatal developmental process. Here, in this study, we will focus on the function of SnRK1 in stomatal development." Please consider, whether it is necessary to refer to the data not included to this manuscript. If this is necessary, please indicate that data is not shown in this article.

5. Authors claim that MG132 treatment does not further decrease levels of phosphorylated KIN10 in the high sucrose conditions, however, the data presented in Figure 4c does not support this statement. Please quantify the data (as is done in Fig 3e) and revise the text according to results of the quantification.

6. Authors say that "KIN10 displayed the ubiquitous expression pattern in all epidermal cells, and mainly localized in the nucleus in guard cells and the smaller cells less than 200 μm^2 , while distributed in the cytoplasm in the pavement cells (Supplementary Fig. 5a-c)". It is great that quantification has been performed. However, these images do not show convincingly the cytosolic localization or absence of nuclear localization in the pavement cells. Figure S5a shows YFP signal in DAPI stained elongated pavement cell nucleus. Please revise the text or replace with more representative images.

7. As authors state, it has been shown that hypoxia causes nuclear localization of KIN10 and authors see nuclear localized KIN10 in stomatal cells. If seedlings have been grown in liquid growth conditions, it seems possible that plants had experienced certain extent of hypoxia related stress and this would explain the nuclear localization. It is not indicated how plants have been grown in this particular experiment (Figure 5e). Please describe in the figure legend or methods systematically used growth conditions. Subcellular localization of KIN10 should be studied also in seedlings grown on solid growth media to rule out that nuclear localization is due to stress caused by liquid growth conditions.

Minor comments:

8. Please indicate number of samples used for each experiment in the figure legend as a numbers ($n=x$). Currently sample numbers are indicated for most experiments as a dots in the graph, however, counting these dots is tedious for the reader. In addition, for some experiments this information is completely lacking, for example Figure 1d.

9. Supplementary files include data on cell-size distribution (data Fig S1g and Fig S3g). Currently nothing is mentioned on this data in the results or methods sections. Please remove or refer to this data in the text and explain how it has been produced in the methods.

10. Please consider indicating significant differences between samples using consistent style throughout the figures. Currently, two styles (letters and asterisks) are used (see for example Fig 1b vs Fig 2 a-c or Fig 1d vs Fig 3e). Also, please consider adding control line to consistent position in the panels. It is misleading that in some panels it is shown in the middle of the samples (Figure 2a-c) and sometimes on the left side (Figure 4g-h).

11. Most quantifications of stomatal marker gene expression are done with plants grown in liquid growth conditions (Figure S2, Figure 2e-f). For subset of them, it is not indicated whether solid or liquid conditions has been used (For example Figure 1e-l). Please indicate systematically how plants have been grown.

12. Figure 3e is mislabeled in legend to be Figure 3f. Please revise the lettering to match the figure.

13. Supplementary Figure 4 shows changes in metabolic pathways as a response to sucrose treatment. Currently the title is "Metabolism pathway of Arabidopsis seedling treated with different concentrations of sucrose". Please revise the title of the figure legend and legend text to indicate what concentration of sucrose has been used. Please also explain how to interpret the data shown in this figure.

14. "Co-localization analysis with pKIN10::KIN10-YFP/pSPCH::SPCH-RFP showed that KIN10 and SPCH were simultaneously distributed in the nucleus in the stomatal lineage cells, indicating that KIN10 physically interacted with SPCH in stomatal lineage cells (Fig 5e)." Co-localization data does not show that two proteins physically interact. Please revise the sentence accordingly or refer to

another figure panel.

15. Figure 5e, legend: Legend states that anti-YFP and anti-myc antibodies were used. The Figure panel is labeled with α -myc and α -GFP. Please revise this image or legend.

16. Figure S8a: Please indicate what sucrose concentration has been used for the treatment. Currently only length of the sucrose treatment is indicated in the Figure legend.

17. "RT-qPCR analysis showed that the short time sucrose treatment had marginal effects on the transcription of SPCH, but significantly induced the protein accumulation of SPCH protein (Supplementary Fig 8b)." Fig S8b does not show qRT-PCR data, but Fig S7b does so please refer here also to Fig S7b.

18. "Consistent with these results, the pBASL::BASL-GFP-labeled cells representing the dividing MMCs gradually increase as the increased concentrations of sucrose, while the pMUTE::MUTE-GFP-labeled cells representing the differentiated cells are most abundant at 1% sucrose concentration and are relatively lower at other high or low concentrations of sucrose." Please correct the description of expression domains. MUTE-GFP labels guard mother cells (cell committed to form stomata) whereas BASL-GFP labels MMCs, meristemoids and the larger SLGCs. Please also revise latter part of the sentence, which is currently hard to interpret.

Responses to comments by reviewers

We wish to express our deep appreciation for the constructive comments on our manuscript by the reviewers. In response to these comments, we have conducted additional experiments and modified the text extensively to improve our manuscript. Specifically, we have added the following results:

1. We have analyzed the stomatal phenotype of plants grown under different light quantity and different light irradiance time to confirm the promoting effects of sugar on stomatal development (Fig. 1a-b and Supplementary Fig. 1a-b).
2. To verify the functions of KIN10 on stomatal development, we have analyzed the stomatal phenotype of wild type, *kin10* and *KIN10-Ox* under different growth condition (Fig. 2a-c, Supplementary Fig. 5a-i, Supplementary Fig. 7a-c, and Supplementary Fig. 8a-d).
3. To rule out the regulation of KIN10 protein by osmotic stress, we have performed immunoblot assays and found that sucrose, but not mannitol, significantly induces the accumulation of KIN10 protein (Fig. 3a,c).
4. We have repeated the immunoblot assays to make the high quality of the P-AMPK western blot (Fig. 4a-c).
5. We have analyzed the stomatal index of *tps1-11* mutant and plants treated with Tre6P to show that Tre6P negatively regulates stomatal development (Supplementary Fig. 10a,b).
6. We have analyzed the subcellular location of KIN10 and KIN10^{T175D} to show that the phosphorylation of KIN10 at T175 has no significant effects on the subcellular location of KIN10 in stomatal lineage cells (Supplementary Fig. 15a-f).
7. We have analyzed the stomatal phenotypes of *pSPCH:SPCH-RFP* and *pSPCH:SPCH4D-RFP* transgenic plants under different growth conditions, and found that the stomatal index of *pSPCH:SPCH4D-RFP* was higher than that of *pSPCH:SPCH-RFP* when plants were grown in solid medium (Supplementary Fig. 18a,b).
8. We have performed immunoblot assays to confirm that sucrose increases the stability of SPCH (Supplementary Fig. 19a-e).
9. We have provided all original blots of western blot images used in this study (Supplementary Fig. 20).
10. We have extensively edited the manuscript to correct grammatical errors and to improve clarity.

Please find below a detailed response to the points raised.

Sincerely,

Mingyi

Reviewers' comments:

Reviewer #1 (Remarks to the Author):

This manuscript describes how physiological levels of sucrose promote stomatal development, and implicates the SnRK1 kinase in this process through the direct phosphorylation of the SPEECHLESS transcription factor. The subject is of very high interest, adding up to the increasing list of regulatory roles of sugars and of the SnRK1 kinase in plant developmental processes. However, there are several major issues regarding experimental design, controls and interpretations that should be addressed before this manuscript can be considered for publication. One major concern relates to the significance of the differences shown in the stomatal index of the KIN10 overexpressor lines vs WT when the degree of variation in this parameter for WT seedlings is very large across experiments (Figs. 1-2). Another one relates to the experiments where the effect of sucrose on KIN10 accumulation is assessed (Fig. 3). These experiments lack important osmotic controls. A third one relates to the overstatement that phosphorylation of KIN10 plays a major role in this process (Fig. 4). And a fourth one related to missing controls in many experiments.

In detail:

Figure 1

1) MAJOR CONCERN: There is large variation in stomatal index for the WT from one graph to the other. For example, compare the Col control in panel b (~ 29%) to the one of 0% sucrose in panel d (~20%).

Response: Thank you for pointing this out. The different stomatal index of wild type control plants in different figures is due to the different growth conditions. The seedlings of Col-0 were grown on ½ MS solid medium or in ½ MS liquid medium for 10 days under long-day condition in figure 1b and figure 1d, respectively. The plants grown in the liquid medium without sugar displayed more severe sugar starvation than that in solid medium, then subsequently resulting in the much significant inhibition of stomatal development. In the revised manuscript, we have added the plant growth conditions to the figure legend.

Considering such differences between WT in different replicate experiments the differences between WT and OE in Fig. 2b seem negligible.

Response: To investigate whether KIN10 is involved in stomatal development, we analyzed the stomatal development of wild type, *kin10* and *KIN10-Ox* plants under different growth conditions. When plants were grown in ½ MS liquid medium containing 1% sucrose, *KIN10-Ox* transgenic plants showed the significantly increased stomatal index, but the *kin10* mutants had the decreased stomatal index comparing to wild type plants (Fig. 2a-c). To rule out the effects of liquid growth condition on the function of KIN10, seedlings of wild type, *kin10* and *KIN10-Ox* were grown on the ½ MS solid medium under different photoperiod conditions. The results showed that overexpression of *KIN10* significantly induced the stomatal development, especially under the 4h light/20h dark photoperiod condition (Supplementary Fig. 7a-c). To further determine whether the positive effects of KIN10 on the stomatal development is similar in the cotyledon and true leaf, the fifth rosette leaves of wild type, *KIN10-Ox* and *kin10* that were grown in soil under the 4h light/20h dark photoperiod condition for 5 weeks were used to perform the stomatal index analysis. The results showed that the stomatal index of rosette leaves of *KIN10-Ox* was higher, and that of *kin10* mutant was lower than that of wild type (Supplementary Fig. 8a-d). Together, these genetic data strong supported that KIN10 is a positive regulator for stomatal development in Arabidopsis.

Figure 2: KIN10 plays crucial roles in sugar-induced stomatal development.

a-c, Quantification of the stomatal index of wild type plants, *KIN10-Ox* and *kin10* mutants. Seedlings of wild type plants (Ler and Col-0), *p35S::KIN10-HA* (Ler background), *p35S::KIN10-myc* (Col-0 background), *kin10* and *kin11* were grown in ½ MS liquid medium with or without 1% sucrose for 10 days under long-day condition. Error bars indicate standard deviation (s.d.) (n=15-20). Different letters above the bars indicated statistically significant differences between the

samples (ANOVA analysis followed by Uncorrected Fisher's LSD multiple comparisons test, $p < 0.05$).

Supplementary Fig. 7. KIN10 is required for sugar-induced stomatal development

a-c, Quantification of the effects of KIN10 on the stomatal development under different photoperiod conditions. Seedlings of Col-0, *p35S::KIN10-myc* and *kin10* were grown on ½ MS solid medium with or without 1% sucrose for 10 days under 16h light/8h dark photoperiod (a), 12h light/12h dark photoperiod (c), and 4h light/20h dark photoperiod conditions (b). Error bars indicate standard deviation (s.d.) (n=20). Different letters above the bars indicated statistically significant differences between the samples (ANOVA analysis followed by Uncorrected Fisher's LSD multiple comparisons test, $p < 0.05$).

Supplementary Fig. 8. KIN10 promotes the stomatal development in rosette leaves.

a, Representative plants of wild type Col-0, *p35S::KIN10-myc* and *kin10* grown in soil under 4h light/20h dark photoperiod for 5 weeks. Scale bar represent 1 cm. b-d, Quantification of stomatal index (b), stomatal density (c) and cell density (d) in the 5th rosette leaves of wild type Col-0, *p35S::KIN10-myc* and *kin10* that were grown in soil under 4h light/20h dark photoperiod for 5 weeks. Error bars indicate standard deviation (s.d.) (n=20). Different letters above the bars

indicated statistically significant differences between the samples (ANOVA analysis followed by Uncorrected Fisher's LSD multiple comparisons test, $p < 0.05$).

Are the compared genotypes always grown in the same plates or in separate plates? Given the differences shown for the WT I believe all compared genotypes should be grown in the same plate, but this is not described in the methods. What can explain this variation?

Response: Thank you for pointing this out. In all experiments, the seedlings of wild type Col-0 or Ler and the compared genotypes were always grown in the same plate. The stomatal index in all experiments was quantified from at least 20 plants and experiments were repeated at least three times.

2) Panel (b). What is a'?

Response: a` represents the statistical analysis of stomatal index of Col-0, *pgm* and *sex1* that were only grown in the medium without sucrose.

3) In panel (a) there is a difference between no sucrose and sucrose (3rd and 4th columns, c,d), but in panel (b) there are no differences. Why?

Response: The statistical difference analysis showed in this ms is the statistical analysis of the stomatal index of Col-0, *pgm* and *sex1* that were only grown in the medium without sucrose. Therefore, it cannot be shown whether sucrose promotes stomatal development. We have re-statistically analyzed the difference of stomatal index among Col-0, *pgm* and *sex1* in the presence and absence of sucrose, and found that sucrose induces the stomatal development in Col, *pgm* and *sex1*. As reviewer suggested, because the results of *pgm* and *sex1* are confusing, we have deleted these data in this revised manuscript.

4) Panel (b). The starch mutants should in principle have differential sucrose accumulation already in the no sucrose plates and one would expect that they have different stomatal indexes in this condition. However, they seem all equal. Also, in the presence of exogenous sucrose (3% sucrose plates) the *sex1* mutant could be

expected to be “rescued” and be similar to the WT, but it is not. I think this graph is just confusing and does not provide much information.

Response: We agree and therefore we have removed these data in the revised manuscript.

5) Legend: express the sugar concentrations of panel c also as % (in addition to mM) to be able to compare with all other results in the paper.

Response: Thank you for pointing this out, we have illustrated it in the figure legend.

Figure 2

1) Panel f. The differences between KIN10 OX and WT for SPCH-GFP accumulation are much higher than the differences in stomatal index shown for these backgrounds in panel b. This should be at least mentioned and discussed.

Response: To verify the effects of KIN10 on stomatal-lineage cell fate transition, we monitored the expression of stomatal cell-type-specific markers in *KIN10-Ox* and wild type plants and found that a higher fraction of cells labeled by *pSPCH::SPCH-GFP* in *KIN10-Ox* than that in wild type plants. The differences between *KIN10-Ox* and wild type for SPCH-GFP accumulation are much higher than the differences in stomatal index, which maybe because the accumulated SPCH induced cell division to form many small cells, and only a few fraction of them will eventually develop into stomata (Macalister et al., 2007)."

References:

MacAlister CA, Ohashi-Ito K, Bergmann DC. Transcription factor control of asymmetric cell divisions that establish the stomatal lineage. *Nature* 2007, **445**(7127): 537-540.

2) Panel f. Which KIN10-OX line is used here?

Response: Thank you for pointing this out. In figure 2f, the *p35S::KIN10-myc-2#* transgenic plants in Col-0 background was used to cross with the *pSPCH::SPCH-YFP* to analyzed the effects of KIN10 on the SPCH protein stability and stomatal development.

Figure 3

1) MAJOR CONCERN: In these experiments the seedlings are probably experiencing an osmotic shock, since they are not grown in sucrose for a few days and all of a sudden they are exposed to 1% sucrose. An osmotic control should be added to show that this is indeed sucrose specific. These controls are provided for the experiments of Fig. 1 where seedlings are grown from the very beginning in these conditions, but not for the ones in this figure where sucrose is provided as a short-term treatment.

Response: Thank you for this excellent suggestion. To rule out the regulation of KIN10 protein by osmotic stress, we have performed immunoblot assays to analyze the KIN10 protein level in the presence of 1% sucrose or 1% mannitol. The results showed that sucrose, but not mannitol, significantly induces the accumulation of KIN10 protein.

Fig. 3. Sucrose induces the KIN10 protein accumulation.

a, Immunoblot analysis of the protein levels of KIN10-myc in *pKIN10::KIN10-myc* transgenic plants using anti-myc antibody. Seedlings of *pKIN10::KIN10-myc* transgenic plants were grown in sugar free ½ MS liquid medium for 3 days and then treated with 1% sucrose or 1% mannitol for different time periods. **b**, Quantitative RT-PCR analysis the expression of *KIN10* in response to sucrose. Wild-type seedlings were grown in sugar free ½ MS liquid medium for 3 days then treated with 1% sucrose for different times. The *PP2A* gene was analyzed as an internal control. Error bars represent standard errors of four independent experiments. Different letters above the bars indicated statistically significant differences between the samples (ANOVA analysis followed by Uncorrected Fisher's LSD multiple comparisons test, $p < 0.05$). **c**, Sucrose induces KIN10 protein accumulation in *p35S::KIN10-myc* transgenic plants. Seedlings of *p35S::KIN10-myc* transgenic plants were grown in sugar free ½ MS liquid medium for 3 days and then treated with 1% sucrose or 1% mannitol for different time periods.

2) MAJOR CONCERN: There is definitely more KIN10 in the sugar treated seedlings.

However:

➤ That does not mean that there is more activity (quite the contrary, as often KIN10 accumulation is observed under conditions where it is inactive)

➤ This could be due to osmotic stress rather than to the presence of sugar, as pointed out in point 1)

➤ So, in brief, in the absence of a proof that there is indeed enhanced KIN10 activity in these conditions, the authors should limit their comments to their observation that there is more protein accumulation

Response: We agree and therefore we have changed the manuscript as suggested.

3) Why is there a duplet in the p35S:KIN10-Myc line in panels (c) and (d)?

Response: Thank you for pointing this out. When using the *p35S::KIN10-Myc* transgenic plants for immunoblot experiments, there are two bands of KIN10-Myc if the SDS-PAGE gel percentage is lower than 7% and the running time is longer. To test whether these two bands is due to the phosphorylated modification of KIN10, KIN10-Myc protein was immunoprecipitated from *p35S::KIN10-Myc* transgenic plants and subjected to phosphatase calf-intestinal alkaline phosphatase (CIP) treatment. The results showed that CIP treatment had no effects on the shift of KIN10-myc protein, indicating these two bands are not caused by the phosphorylation.

Supplementary Fig. CIP phosphatase had no significant effects on the KIN10-myc protein.

Seedlings of *p35S::KIN10-myc* were growing in ½ MS liquid medium containing 1% sucrose for 3 days under long-day condition. KIN10-myc was immunoprecipitated and treated with protein phosphatase. KIN10 proteins were probed with anti-myc, and actin bands were used as loading control.

4) Sometimes the KIN10 blots are indicated to have been done with IP:ed KIN10 (Figure 4) and sometimes not (Figure 3). Which one is it?

Response: Thank you for pointing this out. For the total KIN10 protein, we used the conventional protocol that total protein were lysed from harvested plant tissues and then directly analyzed by immunoblotting assay. However, the levels of phosphorylated KIN10 in plants are too low to be clearly detected. In order to get clear and reliable results, we firstly immunoprecipitated the KIN10 proteins and then analyzed using western blot with the antibody anti-AMPK T172 that recognizes the phosphorylated KIN10 at T175.

Figure 4 (and associated text)

1) MAJOR CONCERN: The claim that sugar controls stomata development by controlling the levels of phosphorylated KIN10 is a strong overstatement (e.g. lines 245-247). The enhanced T-loop signal is fully explained by the higher KIN10 accumulation. Only in the very high sugar concentrations (3-5%) lower phosphorylation ratios can be observed, but given that the focus are the conditions where higher stomatal index is observed (1% sucrose), the effect of 3-5% sucrose on Kin10 phosphorylation is irrelevant.

Response: Thanks for the suggestion. We have changed it.

2) The quality of the P-AMPK westerns is quite low, and if T-loop phosphorylation is to be made a strong point, they should be improved.

Response: Thank you for pointing this out, we have repeated these experiments and made new graphs.

Fig.4. Accumulated Tre6P under energy-rich conditions caused reduced phosphorylation of KIN10.

a-b, Immunoblot analysis of the total KIN10 proteins and phosphorylated KIN10 proteins in the presence of different concentrations (**a**) or different types (**b**) of sugar. Seedlings of *p35S::KIN10-myc* transgenic plants were grown in sugar free ½ MS liquid medium for 3 days and then treated with different concentrations of sucrose or different types of sugar for 6 hours. Myc-trap was used to enrich KIN10 protein by immunoprecipitation. **c**, MG132 and 3-MA had no significant effects on the levels of the phosphorylated KIN10. Seedlings of *p35S::KIN10-myc* plants were grown in sugar free ½ MS liquid medium for 3 days and then treated with or without 3-MA, MG132, and 1% or 5% sucrose for 6 hours. Myc-trap was used to enrich KIN10 protein by immunoprecipitation. Total KIN10 proteins were probed with anti-myc (**a-c**) antibody, phosphorylated KIN10 proteins were analyzed by anti-AMPKT172 antibody. Actin bands were used as loading control.

3) Panel (e). BiFC controls with empty vectors are missing. Most importantly, without showing by Western that both SnAK and KIN10 accumulate similarly in 1% and 5% sucrose one cannot conclude that the decreased BiFC signal in 5% is due to decreased interaction (as opposed to decreased accumulation of either one or both proteins).

Response: As suggested, we have repeated these experiments and analyzed the protein levels of SnAK2 and KIN10 in different concentrations of sucrose. The results showed that both SnAK2 and KIN10 accumulate similar protein levels in 1% and 5% sucrose (Fig. 4g). KIN10 interacted with SnAK1 in the epidermal cells of tobacco leaves, but this interaction was significantly decreased in 5% sucrose condition, indicating that high-dose sugar inhibited the interaction of KIN10 and SnAK1 (Fig. 4e-g).

4) Panel (h). Is there indeed higher T-loop phosphorylation in the SnAK OX lines? That would support the interpretation of the authors that the effect of sucrose on stomatal development is mediated by an increase in KIN10 phosphorylation.

Response: Thanks for the suggestion. We have analyzed the phosphorylated KIN10 levels in the *SnAK1-Ox* transgenic plants. The results showed that overexpression of *SnAK2* resulted in the increased phosphorylated-KIN10 protein (Supplementary Fig. 11a,b).

Supplementary Fig. 11. Overexpression of *SnAK2* led to the increased levels of phosphorylated KIN10 in plants.

a, Immunoblot analysis of phosphorylated KIN10 proteins in wild type Col-0 and *SnAK2-Ox* plants. Seedlings of Col-0 and *p35S:SnAK2-GFP* (*SnAK2-Ox*) were grown in ½ MS liquid medium under long-day condition for 10 days. The phosphorylated KIN10 proteins were analyzed by anti-AMPKT172 antibody, SnAK2-GFP were probed with anti-GFP antibody. Actin bands were used as loading control. **b**, Quantification analysis of the levels of phosphorylated KIN10 protein in wild type and *SnAK2-Ox* plants. The ratio of phosphorylated KIN10 to actin was quantified by ImageJ software. Error bars mean the S.D. (n = 4). Asterisk above dots indicates significant accumulation of phosphorylated KIN10 protein by *SnAK2* overexpression (Student test, ***p* < 0.01).

Figure 5

1) Panel (a). Missing the control of at least BD-SPCH with empty AD vector.

Response: Thanks for the suggestion. We have repeated Y2H experiments and made new graph.

2) Panel (d). It would be more correct to see whether SPCH interacts with GFP alone, as for the MBP in vitro pull down experiments. What are the plants used for these experiments? They are double overexpressors? They are not described anywhere.

Response: Thanks for the suggestion. The transgenic plants co-expressing *KIN10-YFP* and *SPCH-myc* from 35S promoter or only expressing *SPCH-myc* were used to perform co-immunoprecipitation assay. The results showed that KIN10 interacts with SPCH in plants. To verify this interaction, we have performed CoIP assay in Arabidopsis mesophyll cell protoplast by co-expressing *KIN10-YFP* and *SPCH-myc* or *YFP* and *SPCH-myc*. The results showed that KIN10-YFP, but not YFP, interacted with SPCH in plants. We further analyzed the tissue-expression patterns of KIN10 and SPCH using the *pKIN10::KIN10-YFP* and *pSPCH::SPCH-RFP* transgenic plants, and found that KIN10 and SPCH were simultaneously distributed in the nucleus of

the stomatal lineage cells. Together these results indicated that KIN10 physically interacted with SPCH in the stomatal lineage cells.

3) Panel (g) Wouldn't you expect that the D mutation causes higher stomata index, just as what happens in the KIN10-OE lines? At the very least this should be mentioned and discussed.

Response: Thank you pointing this out. To systematically analyzed the effects of KIN10-mediated phosphorylation of SPCH on stomatal development, we analyzed the stomatal phenotypes of *pSPCH:SPCH-RFP*, *pSPCH::SPCH4A-RFP* and *pSPCH:SPCH4D-RFP* transgenic plants under different growth conditions. The results showed that when plants were grown in the liquid medium, the stomatal index of *pSPCH4D::SPCH-RFP* transgenic plants was similar to that of *pSPCH::SPCH-RFP* plants. However, when plants were grown in solid medium, *pSPCH::SPCH4D-RFP* transgenic plants showed the higher stomatal index than *pSPCH::SPCH-RFP* plants (Supplementary Fig. 18a,b). This maybe because that plants growing in liquid medium led to mild hypoxia for plants, which activates KIN10 to phosphorylate and stabilize SPCH. When plants were grown on solid medium, the activity of KIN10 was lower, and SPCH4D was more stable than SPCH, resulting the increased stomatal index of *pSPCH::SPCH4D-RFP* plants comparing to *pSPCH::SPCH-RFP*.

Supplementary Fig. 18. The KIN10-phosphorylation sites are critical for SPCH to promote stomatal development.

a-b, Quantification of the effects of different mutant versions of SPCH on stomatal development. Seedlings of wild type, *pSPCH:SPCH4A-RFP/spch-4* and *pSPCH:SPCH4D-RFP/spch-4* were grown in ½ MS liquid medium (a) or solid medium (b) for 10 days under 16h photoperiod condition. Error bars means the S.D. (n=10-20). Different letters above the bars indicated statistically significant differences between the samples (ANOVA analysis followed by Uncorrected Fisher's LSD multiple comparisons test, $p < 0.05$).

4) Given all the different crosses and transgenics the authors have, it would be good to see some *in vivo* evidence that SPCH phosphorylation is indeed altered in the KIN10 OE background. They could do a Phostag for this TF.

Response: Thanks for this suggestion. We have tried many times to detect the phosphorylation of SPCH by KIN10 in plants, including using phostag, but unfortunately we failed to get clear results. However, our results showed that SPCH interacted with KIN10 *in vivo* and *in vitro*, and was phosphorylated by KIN10 at Thr49, Thr50, Ser51 and Ser52 *in vitro*. Transformation of SPCH with the replacement of these four residues with alanine (SPCH4A) failed to rescue the stomatal development and defective growth phenotype of *spch-4* mutant, due to the instability of SPCH4A protein. These results indicated KIN10 phosphorylates SPCH, thereby increasing SPCH protein stability and subsequently promoting stomatal development.

Material and methods

1) They refer to the *kin10* mutant as *kin10*, which it is not. I believe this is *kin10-3*; also the *kin11* allele is not *kin11-1* and to my knowledge this has not been previously characterized, so a basic characterization should be performed if this is to be included.

Response: Thank you for pointing this out. We have added the details of *kin10* and *kin11* mutants in the material and method section as followed:

“The T-DNA insertion mutants of *GABI_579_E09* (*kin10/snrka1*) and *WiscDsLox384F5* (*kin11/snrka2*) were ordered from The European Arabidopsis Stock Centre. The *GABI_579_E09* mutant has been used the knock out mutant of *KIN10* to determine the its functions in hypocotyl elongation, root growth and stress response (Mair et al., 2015; Ramon et al., 2019; Simon et al., 2018). The mutant *WiscDsLox384F5* has not been used in previous study. Our RT-PCR analysis showed there are some amount of *KIN10* transcript in *kin10* mutant, but no *KIN11* transcript in *kin11* mutant. Immunoblot further confirmed that no significant amount of KIN10 protein and KIN11 protein could be detected in *kin10* and *kin11* mutants, respectively (Supplementary Fig. 6a-c).”

References:

Mair A, Pedrotti L, Wurzinger B, Anrather D, Simeunovic A, Weiste C, *et al.* SnRK1-triggered switch of bZIP63 dimerization mediates the low-energy response in plants. *eLife* 2015, **4**, e05828.

Ramon M, Dang TVT, Broeckx T, Hulsmans S, Crepin N, Sheen J, *et al.* Default activation and nuclear translocation of the plant cellular energy sensor SnRK1 regulate metabolic stress responses and development. *The Plant cell* 2019, **31**:1614-1632.

Simon NML, Sawkins E, Dodd AN. Involvement of the SnRK1 subunit KIN10 in sucrose-induced hypocotyl elongation. *Plant signaling & behavior* 2018, **13**(6): e1457913.

2) MAJOR CONCERN: There is no explanation on the metabolomics analyses, so it is hard to assess whether the T6P measurements are reliable or not. Often LC-MS is used and this cannot separate T6P from other disaccharide-monophosphate isomers.

Response: Thank you for pointing this out. We have added the explanation on the metabolomics analyses in the materials and methods section.

3) What are the light conditions? The plants in Suppl Fig. 1 look quite atrophied and with so elongated petioles?

Response: Thank you for pointing this out. Arabidopsis plants were grown in a greenhouse with white light at $100 \mu\text{mol}\cdot\text{m}^{-2}\cdot\text{s}^{-1}$ and relative humidity of 50% under a 16h light/8h dark cycle at 22 °C for general growth and seed harvesting. In supplementary fig1, the seedlings of wild type Col-0 were grown in ½ MS liquid medium containing different concentrations of sucrose under long-day condition for 10 days.

Is this effect on stomatal index only observed under very low light?

Response: Thank you for pointing this out. To determine the effects of sugar on the stomatal development under different light intensity conditions, we analyzed the stomatal developmental phenotypes of wild type plants that were grown on ½ MS solid medium with or without 1% sucrose under different light quantity and light irradiance time. The results showed that the stomatal index, the number of stomata relative to total epidermal cells, gradually increased with the increasing light photon irradiance (Fig.1a). Similar to this, the stomatal index of plants grown under the 16h

light/8h dark photoperiod condition was much higher than that of plants grown under 4h light/20h dark photoperiod condition (Fig. 1b). However, the decreased stomatal index of plants resulting from the low light quantity and short-time light irradiance was partially recovered by the exogenous sucrose supply (Fig. 1a, b).

Fig. 1. Sucrose promotes stomatal development and alters cell fate in Arabidopsis epidermis.

a, The stomatal index changes in response to light quantity. Seedlings of wild type plants were grown on $\frac{1}{2}$ MS solid medium with or without 1% sucrose under 16h light/8h dark photoperiod with different light intensity for 10 days. **b**, Quantification of the effects of photoperiod and exogenous sucrose on the stomatal development. Seedlings of wild type plants were grown on $\frac{1}{2}$ MS solid medium with or without 1% sucrose under different photoperiod with $100 \mu\text{Mol}\cdot\text{m}^{-2}\cdot\text{s}^{-1}$ for 10 days. **c-d**, Quantification of the effects of different sugars on stomatal index. Seedlings of wild type Col-0 were grown in $\frac{1}{2}$ MS liquid medium containing 30mM mannitol (Man, 0.55%), sucrose (Suc, 1%), glucose (Glu, 0.54%), fructose (Fru, 0.54%), glucose-6-phosphate (G6P, 0.78%) and fructose-1,6-bisphosphate (FBP, 1%) (**c**), or different concentrations of sucrose or mannitol (**d**) under 16h light/8h dark photoperiod with $100 \mu\text{Mol}\cdot\text{m}^{-2}\cdot\text{s}^{-1}$ continuously for 10 days. Error bars indicate S.D. (n=15-20). Different letters above the bars indicated statistically significant differences between the samples (ANOVA analysis followed by Uncorrected Fisher's LSD multiple comparisons test, $p < 0.05$).

4) There is no information of any of the lines generated in this study. A basic molecular characterization should be provided in the supplementary material and a precise description of the constructs employed for their generation. Also the lines should always be referred to in the same manner throughout the text and in the figures. Indicate which variant of KIN10 is expressed in the KIN10 lines (which gene model).

Response: Thank you for pointing this out. We have added the description of the mutants and transgenic plants in the materials and methods section.

Other minor issues:

- Line 100. I would not call dramatic an increase from 11% to 15%.

Response: Thank you for pointing this out, we have changed.

- Line 288. “Wild type plants grown in medium with different concentrations of sucrose”. Is it grown or treated for 4h, as said in the legend?

Response: Thank you for pointing this out, we have changed it.

- Lines 473-474. “Overexpression of the upstream kinase SnAK2 and KIN10T175D...” Is the latter indeed an overexpressor? In the figure it is marked as ProKIN10...

Response: Thank you for pointing this out, we have changed it.

- What is the KIN10-OX used in Suppl Figure 3?

Response: The *p35S::KIN10-myc-2#* was used in the supplementary figure 3.

- Sugars have been mostly shown to inactivate KIN10, so the issue of sucrose activating KIN10 should be discussed in this context

Response: Thank you for pointing this out, we had add discussion as followed:

“SnRK1 and AMPK are conserved protein kinase and share an $\alpha\beta\gamma$ heterotrimeric structure; however, they show different responses to sugar starvation. Studies on animals have shown that AMPK is not only activated by increasing AMP/ATP and ADP/ATP ratios but also by glucose starvation (Lin et al., 2018). When glucose deprivation, the levels of the glycolytic intermediate fructose-1,6-bisphosphate (FBP) decreased and released aldolase, allowing the formation of an AXIN-based AMPK-activation complex to increase the Thr172 phosphorylation of AMPK (Lin et al., 2018). However, in our study, we showed that sugar starvation inhibits the activity of KIN10 by repressing its translation. Sucrose supply induced the accumulation of KIN10 protein by promoting its translation, and the increased KIN10 interacts and phosphorylates SPCH to promote stomatal development. In addition, under high sucrose concentrations, high levels of Tre6P were produced to inhibit the activity of KIN10 via attenuating its interaction with SnAKs. In animal cells, glucose starvation activates AMPK to repress mRNA translation through at least three different

mechanisms, including phosphorylation of RAPTOR to inhibit the activity of mTOR, phosphorylation of eukaryotic elongation factor-2 (eEF2) kinase that can phosphorylate and inactivate eEF2, and phosphorylation of Unc-51 like autophagy activating kinase 1 (ULK1) to block the leucylation of tRNA(Leu) (Zhang et al., 2020). A recent study showed that SnRK1 also inhibits the mRNA translation by phosphorylating the cap binding proteins eIF4E and eIFiso4E (Bruns et al., 2019). However, whether SnRK1 is involved in sugar starvation-reduced translation remains unclear.”

References:

- Bruns AN, Li S, Mohannath G, Bisaro DM. Phosphorylation of Arabidopsis eIF4E and eIFiso4E by SnRK1 inhibits translation. *The FEBS journal* 2019, **286**(19): 3778-3796.
- Lin SC, Hardie DG. AMPK: Sensing Glucose as well as Cellular Energy Status. *Cell metabolism* 2018, **27**(2): 299-313.
- Zhang CS, Hardie DG, Lin SC. Glucose Starvation Blocks Translation at Multiple Levels. *Cell metabolism* 2020, **31**(2): 217-218.

- The manuscript would benefit from language editing

Response: Thank you for pointing this out, we have extensively edited the revised manuscript.

Reviewer #2 (Remarks to the Author):

The authors report the novel and important finding that the SnRK1 kinase mediates sugar-regulated stomatal development in Arabidopsis cotyledons. The study provides strong genetic and molecular evidence that KIN10 (a SnRK1 catalytic subunit) is a positive regulator of stomatal development by directly interacting with and phosphorylating and stabilizing the SPCH transcription factor. The science is of high quality and the results are carefully presented.

SnRK1 is generally established as a metabolic stress sensor, which is activated by carbon starvation and repressed by increased sugar levels. The latter is also observed

in this study with higher sucrose levels (and associated increased T6P levels) that inhibit stomatal development. The observation that lower levels of sucrose (1%) increase SnRK1 activity by increasing KIN10 translation to promote stomatal development is surprising. The manuscript therefore requires more discussion of these apparent discrepancies with previous findings on SnRK1 regulation.

Response: Thank you for pointing this out, we had add discussion as followed:

“SnRK1 and AMPK are conserved protein kinase and share an $\alpha\beta\gamma$ heterotrimeric structure; however, they show different responses to sugar starvation. Studies on animals have shown that AMPK is not only activated by increasing AMP/ATP and ADP/ATP ratios but also by glucose starvation (Lin et al., 2018). When glucose deprivation, the levels of the glycolytic intermediate fructose-1,6-bisphosphate (FBP) decreased and released aldolase, allowing the formation of an AXIN-based AMPK-activation complex to increase the Thr172 phosphorylation of AMPK (Lin et al., 2018). However, in our study, we showed that sugar starvation inhibits the activity of KIN10 by repressing its translation. Sucrose supply induced the accumulation of KIN10 protein by promoting its translation, and the increased KIN10 interacts and phosphorylates SPCH to promote stomatal development. In addition, under high sucrose concentrations, high levels of Tre6P were produced to inhibit the activity of KIN10 via attenuating its interaction with SnAKs. In animal cells, glucose starvation activates AMPK to repress mRNA translation through at least three different mechanisms, including phosphorylation of RAPTOR to inhibit the activity of mTOR, phosphorylation of eukaryotic elongation factor-2 (eEF2) kinase that can phosphorylate and inactivate eEF2, and phosphorylation of Unc-51 like autophagy activating kinase 1 (ULK1) to block the leucylation of tRNA(Leu) (Zhang et al., 2020). A recent study showed that SnRK1 also inhibits the mRNA translation by phosphorylating the cap binding proteins eIF4E and eIFiso4E (Bruns et al., 2019). However, whether SnRK1 is involved in sugar starvation-reduced translation remains unclear.”

References:

Bruns AN, Li S, Mohannath G, Bisaro DM. Phosphorylation of Arabidopsis eIF4E and eIFiso4E by SnRK1 inhibits translation. *The FEBS journal* 2019, **286**(19): 3778-3796.

Lin SC, Hardie DG. AMPK: Sensing Glucose as well as Cellular Energy Status. *Cell metabolism* 2018, **27**(2): 299-313.

Zhang CS, Hardie DG, Lin SC. Glucose Starvation Blocks Translation at Multiple Levels. *Cell metabolism* 2020, **31**(2): 217-218.

More specific remarks and suggestions:

- Line 7-11: The abstract could be clearer on what was demonstrated before by other groups (T6P inhibition of KIN10 interaction with its upstream kinases) and what is found in the present study.

Response: Thank you for pointing this out, we have changed the sentence as followed:

“Sucrose not only induces the accumulation of KIN10, the catalytic α -subunit of SnRK1, by enhancing KIN10 translation; but also triggers the production of trehalose-6-phosphate (Tre6P) to inhibit the activity of KIN10”

- These studies are done with cotyledons. Why? Would similar results be obtained with true (sink) leaf? Wouldn't this be physiologically more relevant?

Response: Stomatal formation in Arabidopsis follows a more or less tip-to-base gradient in leaves, which means that stomata firstly form at the leaf tip and then gradually appear lower down. This gradient complicates population level studies of the timing of stomatal development. However, this gradient is reduced or absent in cotyledons. For this reason, cotyledons were used to study the stomatal development (Pyke et al., 1991; Donnelly et al., 1999; Geisler and Sack, 2001).

To determine whether the positive effect of KIN10 on the stomatal development is similar in the cotyledon and true leaf, the fifth rosette leaves of wild type, *KIN10-Ox* and *kin10* that were grown in soil for 5 weeks were used to perform the stomatal index analysis. The results showed that the stomatal index of rosette leaves of *KIN10-Ox* was higher, and that of *kin10* mutant was lower than that of wild type, indicating that KIN10 is a positive regulator for stomatal development in cotyledon and leaves.

Supplementary Fig. 8. KIN10 promotes the stomatal development in rosette leaves.

a, Representative plants of wild type Col-0, *p35S::KIN10-myc* and *kin10* grown in soil under 4h light/20h dark photoperiod for 5 weeks. Scale bar represent 1 cm. **b-d**, Quantification of stomatal index (**b**), stomatal density (**c**) and cell density (**d**) in the 5th rosette leaves of wild type Col-0, *p35S::KIN10-myc* and *kin10* that were grown in soil under 4h light/20h dark photoperiod for 5 weeks. Error bars indicate standard deviation (s.d.) (n=20). Different letters above the bars indicated statistically significant differences between the samples (ANOVA analysis followed by Uncorrected Fisher's LSD multiple comparisons test, $p < 0.05$).

References:

- Donnelly PM, Bonetta D, Tsukaya H, Dengler RE, NG. D. Cell cycling and cell enlargement in developing leaves of Arabidopsis. *Development Biology* 1999, **215**: 407-419.
- Geisler MJ, FD. S. Variable timing of developmental progression in the stomatal pathway in Arabidopsis cotyledons. *New Phytologist* 2001, **152**: 469-476.
- Pyke K, Marrison JL, RM L. Temporal and spatial development of the cells of the expanding first leaf of Arabidopsis thaliana. *Journal of experimental botany* 1991, **42**: 1407-1416.

- Line 96: A prolonged darkness of 7 days seems extreme. Are similar effects seen after a few days of darkness, which should already cause a significant starvation condition?

Response: Thanks for the suggestion. We have systematically analyzed the stomatal development of plants grown on the prolonged darkness of different days. The results showed that the prolonged darkness significantly reduced the stomata index, but this

reduction was partially repressed by exogenous sucrose treatment (Supplementary Fig. 1a,b), suggesting that sugar promotes the stomatal development.

Supplementary Fig. 1. Quantification of the effects of prolonged darkness and exogenous sucrose on the stomatal development.

a, The prolonged darkness inhibited the stomatal development on the abaxial cotyledons of 10-day-old wild type plants. Seedlings of wild type Col-0 were grown on ½ MS solid medium without sucrose under long-day condition for 10 days or transferred to darkness different days (n=20). **b**, Sucrose supply partially suppressed the decreased stomatal index by the prolonged darkness. Seedlings of wild type Col-0 were grown on ½ MS solid medium with or without 3% sucrose under long-day condition for 3 days and then transferred to darkness for 7 days, or continuously under long-day condition for 10 days (n=10). Error bars means the S.D.. Different letters above the bars indicated statistically significant differences between the samples (ANOVA analysis followed by Uncorrected Fisher's LSD multiple comparisons test, $p < 0.05$).

- Figure 1: In (a) the effect of 3% sucrose appears much smaller than in (d). What could be the reason for that? Is there a lot of variation between experiments? In (b) the stomatal index does not appear to be affected in the absence of exogenous sucrose, as would be expected with these mutations. Why have G6P and (especially) FBP been tested specifically? Are such sugar-phosphates taken up?

Response: Thank you for pointing this out. The different effects of sucrose on the stomatal development in figure 1a, 1b and figure 1d were because of the different growth conditions in these experiments. The seedlings of Col-0 were grown on ½ MS solid medium or in ½ MS liquid medium for 10 days under long-day condition in figure 1a, 1b and figure 1d, respectively. The plants grown in the liquid medium without sugar displayed more severe sugar starvation than that in solid medium, then subsequently resulting the much significant promotion of sucrose on the stomatal development.

- Figure 2: Why is a different representation of the statistical differences used here (similar multiple comparisons test)? In (c) the stomatal index of the *kin10* mutants appears higher than WT in the absence of sucrose. Is this (statistically) relevant?

Response: Thank you for pointing this out. We have re-statistically analyzed the difference of stomatal index among wild type, *KIN10-Ox* and *kin10* in the presence or absence of sucrose. These results showed there were no significant difference between wild type, *KIN10-Ox* and *kin10* mutant in the absence of sucrose, while in the presence of sucrose, overexpression of *KIN10* led to the increased stomatal index and knock-out of *KIN10* caused the decreased stomatal index comparing to wild-type plants. These maybe because of the low translation efficiency of KIN10 under the carbon starvation, which resulted in the similar low levels of KIN10 protein in wild type, *KIN10-Ox* and *kin10* mutants. While in the presence of sucrose, the protein translation of KIN10 was no longer the rate-limiting factor, so the *KIN10-Ox* transgenic plants contain the high levels of KIN10, and consequently resulting in the increased the stomatal index. These results provide strong genetic evidence that sucrose can regulate the activity of KIN10 to regulate the stomatal development.

Fig. 2. KIN10 plays crucial roles in sugar-induced stomatal development

a-c, Quantification of the stomatal index of wild type plants, *KIN10-Ox* and *kin10* mutants. Seedlings of wild type plants (Ler and Col-0), *p35S::KIN10-HA* (Ler background), *p35S::KIN10-myc* (Col-0 background), *kin10* mutant and *kin11-1* were grown in ½ MS liquid medium with or without 1% sucrose for 10 days under long-day condition. Error bars indicate standard deviation (s.d.) (n=15-20). Different letters above the bars indicated statistically significant differences between the samples (ANOVA analysis followed by Uncorrected Fisher's LSD multiple comparisons test, $p < 0.05$).

- Line 196: Not sure the 'activity' of SnRK1 is tested here.

Response: Thank you for pointing this out. We have changed the sentence as followed:

“To investigate how sugar optimizes the stomatal development through KIN10, we firstly analyzed the effects of sucrose on the protein levels of KIN10 in the *pKIN10::KIN10-myc* transgenic plants that were grown in liquid half-strength MS medium without any sugar for three days and then treated with sucrose for various lengths of time.”

- Figure 3: SnRK1 activity is not really quantified. TPS5 is a known sugar upregulated gene, but some of the (many) established SnRK1 targets should be tested in these conditions. I think this is very important to confirm that SnRK1 activity is indeed up in these conditions! T-loop phosphorylation does not always correlate well with activity.

Response: We agree the opinions of reviewer that T-loop phosphorylation does not always correlate with the activity of KIN10. However, phosphorylation of a highly conserved threonine residue in the T-loop is essential for the catalytic activity of SnRK1 (Crepin et al., 2019; Crozet et al., 2014). Nucleocytoplasmic shuttle is another important regulatory mechanism for KIN10 activity (Ramon et al., 2019). KIN10 with phosphorylation at the T-loop site need to be located in the nucleus to interact and phosphorylate some transcription factors to reprogram gene expression, and then regulate plant growth and stress response. Here, in this study we showed that nuclear-localized KIN10 is highly enriched in the stomatal lineage cells to phosphorylate and stabilize SPCH and promote stomatal development. The protein levels of SPCH could indirectly reflect the activity of KIN10 in stomatal lineage cells. Our genetic and biochemical assays showed that SPCH is highly accumulated in the overexpression of *KIN10* plants or wild type plants that were grown on medium with 1% sucrose, indicating that KIN10 has higher activity in these conditions.

References:

- Crepin N, Rolland F. SnRK1 activation, signaling, and networking for energy homeostasis. *Current opinion in plant biology* 2019, **51**: 29-36.
- Crozet P, Margalha L, Confraria A, Rodrigues A, Martinho C, Adamo M, *et al.* Mechanisms of regulation of SNF1/AMPK/SnRK1 protein kinases. *Frontiers in plant science* 2014, **5**: 190.
- Ramon M, Dang TVT, Broeckx T, Hulsmans S, Crepin N, Sheen J, *et al.* Default activation and nuclear translocation of the plant cellular energy sensor SnRK1 regulate metabolic stress responses and development. *The Plant cell* 2019, **31**:1614-1632.

In (b) expression is tested only after 4h, while (c) shows that protein levels are not upregulated yet at that timepoint.

Response: Thank you for pointing this out. We have performed the qRT-CPR to analyzed the effects of sucrose treatment at different times on gene expression. The results showed that *TPS5*, but not *KIN10*, is significantly induced by sucrose, indicating that sucrose induces the accumulation of KIN10 via post-transcriptional regulation.

- Figure 4: The authors correctly conclude that there is ‘an accumulation of phosphorylated KIN10’ with 6h of low levels of sucrose. However, it is not clear whether the T-loop phosphorylation status (i.e. the ratio of phosphorylated versus non-phosphorylated KIN10) changes. Apparently not, indicating that protein levels go up without changing T-loop phosphorylation status, still resulting in more total active SnRK1. Upon addition of higher inhibitory sucrose levels, however, this ratio clearly decreases as expected. This could maybe be discussed better.

Response: We have repeated these experiments and calculated the ratio of phosphorylated versus total protein of KIN10 under different conditions. The results showed that sucrose induced the accumulation of KIN10 protein in a dose-dependent manner. While sucrose accumulated the phosphorylated KIN10 at low concentrations, but not at high concentration. The ratio of phosphorylated KIN10 proteins to total KIN10 proteins in plants was similar when the concentrations of exogenous sucrose were less than 3%, but this ratio significantly decreased in the presence of 5% sucrose, which is consistent with the reported inhibition of KIN10 phosphorylation by high levels of sugar in plants.

The figure 4 legend title suggests a causal link between T6P and T-loop phosphorylation, which was indeed shown before, but cannot be concluded from these data (correlation). I would change the title.

Response: As suggested, we have changed the subtitle as follows:

“Sucrose dynamically regulated the phosphorylation status of KIN10”

T6P levels also already go up significantly after just 4h of 1% sucrose.

Response: We agree that 1% and 5% sucrose treatment both induced the production of Tre6P. Compared with the control, 1% sucrose treatment increased the content of Tre6P about 17 fold, but 5% sucrose treatment increased the levels of Tre6P about 40 fold. Sucrose with 1% or 5% concentrations both promoted the accumulation of KIN10 protein, but the active phosphorylated-KIN10 was determined by the ratio of sucrose to Tre6P in plants. Under the 1% sucrose growth condition, the ratio of sucrose to Tre6P is higher, resulting the most of KIN10 proteins in plants can be active by the SnAK1-mediated phosphorylation at the conserved T175. Whereas under 5% sucrose growth condition, the ratio of sucrose to Tre6P was decreased due to the large amount of Tre6P induced synthesis, consequently causing the decreased phosphorylated KIN10 protein in plants. These results indicated that the dynamic balance of sucrose and Tre6P optimizes the output specificity of KIN10 to regulate plant development and stress response.

Analysis of stomatal development in a *tps1* mutant (containing a rescue construct to bypass embryo-lethality) could be interesting.

Response: Thank you for this excellent suggestion. We have analyzed the stomatal development of a weak mutant of TPS1, *tps1-11*, which has been used by several groups to study the function of TPS1 in seed germination (Gomez et al., 2010), circadian clock (Frank et al., 2018) and immune response (Lowe et al., 2009). The results showed that the stomatal index of *tps1-11* was significantly higher than that of wild type plants. Further, we showed exogenous Tre6P supply also significantly reduced the stomatal index. These results indicated that TPS1 inhibits the stomatal development by controlling the biosynthesis of Tre6P.

Supplementary Fig. 10. Tre6P negatively regulates stomatal development.

a, Quantification of the effects of Tre6P on stomatal index in wild type abaxial cotyledons. Seedlings of wild type Col-0 were grown in the liquid ½ MS medium containing 100μM Tre6P and 1% sucrose or only 1% sucrose for 10 days under 16h light/8h dark photoperiod (n=14). **b**, Mutation TPS1 resulted in the increased stomatal index. Seedlings of wild type Ler and *tps1-11* were grown in the liquid ½ MS medium containing 1% sucrose for 10 days under 16h light/8h dark photoperiod (n=20). Error bars indicate standard deviation (s.d.). Different letters above the bars indicated statistically significant differences between the samples (ANOVA analysis followed by Uncorrected Fisher's LSD multiple comparisons test, $p < 0.05$).

References:

- Frank A, Matioli CC, Viana AJC, Hearn TJ, Kusakina J, Belbin FE, *et al.* Circadian Entrainment in Arabidopsis by the Sugar-Responsive Transcription Factor bZIP63. *Current biology : CB* 2018, **28**(16): 2597-2606 e2596.
- Gomez LD, Gilday A, Feil R, Lunn JE, Graham IA. AtTPS1-mediated trehalose 6-phosphate synthesis is essential for embryogenic and vegetative growth and responsiveness to ABA in germinating seeds and stomatal guard cells. *The Plant journal : for cell and molecular biology* 2010, **64**(1): 1-13.
- Lowe RG, Lord M, Rybak K, Trengove RD, Oliver RP, Solomon PS. Trehalose biosynthesis is involved in sporulation of Stagonospora nodorum. *Fungal genetics and biology : FG & B* 2009, **46**(5): 381-389.

- No methods information is included for sugar and sugar-P level quantification; especially for T6P, this is critical.

Response: Thank you for pointing this out. We have added the explanation on the metabolomics analyses in the materials and methods section.

- Figure 5 is very convincing (complementary, independent ways to show physical interaction and phosphorylation). Could SPCH stabilization (h) be confirmed using immunoblotting?

Response: As suggested by reviewer, we have analyzed the stabilization of SPCH

and its mutant version SPCH4A with or without sucrose treatment using immunoblotting assays. The results showed that sucrose supply significantly induced the accumulation of SPCH-myc, but had weak effects on the SPCH4A-myc (Supplementary Fig. 19c-e).

Supplementary Fig. 19. Sucrose increases the SPCH protein stability.

c, RT-PCR analysis the expression levels of *SPCH* and mutation form of *SPCH* in wild type and different transgenic plants. *PP2A* was used to verify equal cDNA loading. **d-e**, Immunoblot analysis of the effects of sucrose on the protein levels of SPCH-myc and SPCH4A-myc. Seedlings of *p35S::SPCH-myc* and *p35S::SPCH4A-myc* were grown in sugar free medium for 3 days, and then treated with 1% sucrose for different times (**d**) or for 24 hours (**e**). Total KIN10 proteins were probed with anti-myc antibody, actin bands were used as loading control. An antibody against actin was used to verify equal protein loadings.

- Discussion/Line 404: As mentioned above, I am not sure the prolonged darkness is reflecting a physiologically relevant starvation condition. Just like high sugar supply feedback inhibits stomatal development, I would expect mild starvation stress to stimulate (through SnRK1) stomatal development and C uptake.

Response: Thank you for pointing this out, we have analyzed the stomatal phenotype of plants grown under low intensity light or short-day photoperiod conditions that mimics the mild sugar starvation stress. The result showed that the stomatal index gradually increased with the increasing light photon irradiance (Fig.1a). Similar to this, the stomatal index of plants grown under the 16h light/8h dark photoperiod condition was much higher than that of plants grown under 4h light/20h dark photoperiod condition (Fig. 1b). However, the decreased stomatal index of plants resulting from the low light quantity and short-time light irradiance was partially recovered by the exogenous sucrose supply (Fig. 1a, b). Furthermore, the increased stomatal index of *KIN10-Ox* transgenic plants comparing to wild type was more obvious under 4h light/20h dark photoperiod condition than other photoperiod conditions (Supplementary Fig. 7a-c). These results indicated that sugar promotes stomatal development by regulating the activity of SnRK1.

Fig. 1. Sucrose promotes stomatal development and alters cell fate in Arabidopsis epidermis.

a, The stomatal index changes in response to light quantity. Seedlings of wild type plants were grown on $\frac{1}{2}$ MS solid medium with or without 1% sucrose under 16h light/8h dark photoperiod with different light intensity for 10 days. **b**, Quantification of the effects of photoperiod and exogenous sucrose on the stomatal development. Seedlings of wild type plants were grown on $\frac{1}{2}$ MS solid medium with or without 1% sucrose under different photoperiod with $100 \mu\text{Mol}\cdot\text{m}^{-2}\cdot\text{s}^{-1}$ for 10 days. **c-d**, Quantification of the effects of different sugars on stomatal index. Seedlings of wild type Col-0 were grown in $\frac{1}{2}$ MS liquid medium containing 30mM mannitol (Man, 0.55%), sucrose (Suc, 1%), glucose (Glu, 0.54%), fructose (Fru, 0.54%), glucose-6-phosphate (G6P, 0.78%) and fructose-1,6-bisphosphate (FBP, 1%) (**c**), or different concentrations of sucrose or mannitol (**d**) under 16h light/8h dark photoperiod with $100 \mu\text{Mol}\cdot\text{m}^{-2}\cdot\text{s}^{-1}$ continuously for 10 days. Error bars indicate S.D. ($n=15-20$). Different letters above the bars indicated statistically significant differences between the samples (ANOVA analysis followed by Uncorrected Fisher's LSD multiple comparisons test, $p < 0.05$).

Supplementary Fig. 7. KIN10 is required for sugar-induced stomatal development

a-c, Quantification of the effects of KIN10 on the stomatal development under different photoperiod conditions. Seedlings of Col-0, *p35S::KIN10-myc* and *kin10* were grown on $\frac{1}{2}$ MS solid medium with or without 1% sucrose for 10 days under 16h light/8h dark photoperiod (**a**), 12h light/12h dark photoperiod (**b**), and 4h light/20h dark photoperiod conditions (**c**). Error bars indicate standard deviation (s.d.) ($n=20$). Different letters above the bars indicated statistically significant differences between the samples (ANOVA analysis followed by Uncorrected Fisher's LSD multiple comparisons test, $p < 0.05$).

- Line 468: I would be careful making statements about T-loop phosphorylation status

correlating with stomatal development. An ‘accumulation of phosphorylated KIN10’ with low sucrose concentrations (Line 470) is correct.

Response: As suggested, we have changed it.

Minor remarks:

- Line 36: SnAK1 and SnAK2 are GRIK2 and GRIK1

Response: Thank you for pointing this out, we have changed it.

- Line 43: Delete ‘in plants’

Response: Thank you for pointing this out, we have changed it.

- Line 45: How is this a feedback loop?

Response: Thank you for pointing this out, we have changed it.

- Line 172: ‘containing’

Response: Thank you for pointing this out, we have changed it.

- Some other typo’s

Response: Thank you for pointing this out, we have extensively edited the revised manuscript.

Reviewer #3 (Remarks to the Author):

Review - Han et al. 2019 submitted to Nat Comm

The author present evidence that the sugar-sensitive SnRK1 kinase regulates SPCH stability in a sugar concentration dependent manner to regulate stomatal development. They provide evidence that low excess sugar levels increase stomatal index (SI) while high excess sugar level does not. Overexpressing and mutating the SnRK1 subunit KIN10 enhances and attenuates the sugar effect on SI, respectively. The authors show that sugar accumulates KIN10 through regulating translation and not transcription. Interestingly, the KIN10 protein level increases linearly with sugar concentration but

the active phosphorylated form of KIN10 is sensitive to too high sugar levels likely due to an excess of T6P, which scales linearly with sucrose. Finally, the authors present evidence that KIN10 indeed interacts with and phosphorylates SPCH at specific residues leading to stabilization of SPCH and, consequently, to a higher SI.

While much of the data presented is sound, the story built in a logical and easy-to-read manner, I would like to see the following issues addressed:

Major concerns

1. solid MS plates vs. liquid MS cultures

For their initial findings of how sucrose induces a higher SI, the authors use plate grown plates (eg. 1A, B) but most of their other quantifications stem from liquid MS culture grown plants. While I understand that these are much easier to deal with when performing protein work, I am a bit worried about how submergence of plants affects stomatal development and the sensitivity to sugar b/c gas exchange is obviously highly limited. On plate in Fig. 1A the change of SI from 0% to 3% sugar is from 28% to 32%, whereas in Fig. 2b and c in liquid culture SI changes from less than 25% to more than 40% in 1% sucrose. Therefore, I would like to see how solid vs. liquid culture affects stomatal development (ie. SI) with 1% sucrose and w/o sucrose for wildtype, KIN10-OE and kin10 mutant plants.

Response: Thank you for this excellent suggestion, we have compared the stomatal developmental phenotypes of wild type, *KIN10-Ox* and *kin10* that were grown in ½ MS liquid medium or on ½ MS solid medium with or without 1% sucrose. When plants were grown in ½ MS liquid medium containing 1% sucrose, *KIN10-Ox* transgenic plants showed the significantly increased stomatal index, but the *kin10* mutants had the decreased stomatal index comparing to wild type plants (Fig. 2a-c). To rule out the effects of liquid growth condition on the function of KIN10, seedlings of wild type, *kin10* and *KIN10-Ox* were grown on the ½ MS solid medium under different photoperiod conditions. The results showed that overexpression of *KIN10* significantly induced the stomatal development, especially under the 4h light/20h dark photoperiod condition (Supplementary Fig. 7a-c). To further determine whether the positive effects of KIN10 on the stomatal development is similar in the cotyledon and true leaf, the fifth rosette leaves of wild type, *KIN10-Ox* and *kin10* that were grown

in soil under the 4h light/20h dark photoperiod condition for 5 weeks were used to perform the stomatal index analysis. The results showed that the stomatal index of rosette leaves of *KIN10-Ox* was higher, and that of *kin10* mutant was lower than that of wild type (Supplementary Fig. 8a-d). Together, these genetic data strongly supported that KIN10 is a positive regulator for stomatal development in Arabidopsis.

Figure 2: KIN10 plays crucial roles in sugar-induced stomatal development.

a-c, Quantification of the stomatal index of wild type plants, *KIN10-Ox* and *kin10* mutants. Seedlings of wild type plants (Ler and Col-0), *p35S::KIN10-HA* (Ler background), *p35S::KIN10-myc* (Col-0 background), *kin10* and *kin11* were grown in 1/2 MS liquid medium with or without 1% sucrose for 10 days under long-day condition. Error bars indicate standard deviation (s.d.) (n=15-20). Different letters above the bars indicated statistically significant differences between the samples (ANOVA analysis followed by Uncorrected Fisher's LSD multiple comparisons test, $p < 0.05$).

Supplementary Fig. 7. KIN10 is required for sugar-induced stomatal development

a-c, Quantification of the effects of KIN10 on the stomatal development under different photoperiod conditions. Seedlings of Col-0, *p35S::KIN10-myc* and *kin10* were grown on 1/2 MS solid medium with or without 1% sucrose for 10 days under 16h light/8h dark photoperiod (**a**), 12h light/12h dark photoperiod (**b**), and 4h light/20h dark photoperiod conditions (**c**). Error bars indicate standard deviation (s.d.) (n=20). Different letters above the bars indicated statistically significant differences between the samples (ANOVA analysis followed by Uncorrected Fisher's LSD multiple comparisons test, $p < 0.05$).

Supplementary Fig. 8. KIN10 promote the stomatal development in rosette leaves.

a, Representative plants of wild type Col-0, *p35S::KIN10-myc* and *kin10* grown in soil under 4h light/20h dark photoperiod for 5 weeks. Scale bar represent 1 cm. **b-d**, Quantification of stomatal index (**b**), stomatal density (**c**) and cell density (**d**) in the 5th rosette leaves of wild type Col-0, *p35S::KIN10-myc* and *kin10* that were grown in soil under 4h light/20h dark photoperiod for 5 weeks. Error bars indicate standard deviation (s.d.) (n=20). Different letters above the bars indicated statistically significant differences between the samples (ANOVA analysis followed by Uncorrected Fisher's LSD multiple comparisons test, $p < 0.05$).

2. sugar-dependency

Throughout the manuscript the authors are a bit inaccurate regarding the effects they see, which are mostly sugar dependent (eg. line 168-170, 107-108, 177-179) whereas in other places it is clearly specified (eg. line 172). I would like to ask the authors to carefully go through the text and figure legends again and clarify where appropriate. Along similar lines it is sometimes unclear if the data derives from sugar-treated plants or not. Therefore, please specify this for each subpanel in the figure legends and in addition add the sugar concentration above the graphs that do not compare different sucrose levels throughout the manuscript like in Fig. 4G and H. In addition, the authors keep using the expression "sugar starvation" (eg. in Fig. 6 and the discussion). Except when grown in the dark, this expression is not true. 0% sugar is rather "no sucrose excess. Fig. 6 and throughout the text should rather say no sucrose, low sucrose and high sucrose. For example line 378 claims that washing out sucrose leads to sugar starvation. But the plants still experience light and can make their own

sucrose. Along these lines, I also don't really understand how the prolonged darkness experiment fits in. This is a confounding effect of light regulation on stomatal development together with a potential sugar starvation signal. Please elaborate or remove.

Response: We agree the reviewer's opinion that it is difficult for plant to achieve the sugar starvation because plants produce sugar by photosynthesis. In order to better understand the effects of sugar on stomatal development, the prolonged darkness experiments were used to reduce the endogenous sugar in plants. In the new revised manuscript, we have systematically analyzed the stomatal development of plants grown on the prolonged darkness of different days, or plants grown under different light quantity or different photoperiod conditions. The results showed the stomatal index gradually increased with the increasing light photon irradiance (Fig.1a). Similarly, the stomatal index of plants grown under the 16h light/8h dark photoperiod condition was much higher than that of plants grown under 4h light/20h dark photoperiod condition (Fig. 1b). However, the decreased stomatal index of plants resulting from the low light quantity and short-time light irradiance was partially recovered by the exogenous sucrose supply (Fig. 1a, b). Prolonged darkness, which induces the carbon starvation phenotype in seedlings due to the lack of photosynthesis and the depletion of the reserved starch, inhibits stomatal development (Supplementary Fig. 1a). Under such starvation conditions, exogenous application of sucrose dramatically increased stomatal development, and the stomatal index increased to approximately 15% (Supplementary Fig. 1b). These results indicated that sugar promotes the stomatal development.

Fig. 1. Sucrose promotes stomatal development and alters cell fate in Arabidopsis epidermis.

a, The stomatal index changes in response to light quantity. Seedlings of wild type plants were grown on $\frac{1}{2}$ MS solid medium with or without 1% sucrose under 16h light/8h dark photoperiod with different light intensity for 10 days. **b,** Quantification of the effects of photoperiod and

exogenous sucrose on the stomatal development. Seedlings of wild type plants were grown on $\frac{1}{2}$ MS solid medium with or without 1% sucrose under different photoperiod with $100 \mu\text{Mol}\cdot\text{m}^{-2}\cdot\text{s}^{-1}$ for 10 days. **c-d**, Quantification of the effects of different sugars on stomatal index. Seedlings of wild type Col-0 were grown in $\frac{1}{2}$ MS liquid medium containing 30mM mannitol (Man, 0.55%), sucrose (Suc, 1%), glucose (Glu, 0.54%), fructose (Fru, 0.54%), glucose-6-phosphate (G6P, 0.78%) and fructose-1,6-bisphosphate (FBP, 1%) (c), or different concentrations of sucrose or mannitol (d) under 16h light/8h dark photoperiod with $100 \mu\text{Mol}\cdot\text{m}^{-2}\cdot\text{s}^{-1}$ continuously for 10 days. Error bars indicate S.D. (n=15-20). Different letters above the bars indicated statistically significant differences between the samples (ANOVA analysis followed by Uncorrected Fisher's LSD multiple comparisons test, $p < 0.05$).

Supplementary Fig. 1. Quantification of the effects of prolonged darkness and exogenous sucrose on the stomatal development.

a, The prolonged darkness inhibited the stomatal development on the abaxial cotyledons of 10-day-old wild type plants. Seedlings of wild type Col-0 were grown on $\frac{1}{2}$ MS solid medium without sucrose under long-day condition for 10 days or transferred to darkness different days (n=20). **b**, Sucrose supply partially suppressed the decreased stomatal index by the prolonged darkness. Seedlings of wild type Col-0 were grown on $\frac{1}{2}$ MS solid medium with or without 3% sucrose under long-day condition for 3 days and then transferred to darkness for 7 days, or continuously under long-day condition for 10 days (n=10). Error bars means the S.D.. Different letters above the bars indicated statistically significant differences between the samples (ANOVA analysis followed by Uncorrected Fisher's LSD multiple comparisons test, $p < 0.05$).

3. T6P

The authors show correlative evidence that T6P might inhibit the interaction of SnAK and KIN10 in high excess sugar. Accordingly, growing plants with T6P and 1% sucrose should decrease the SI much like the kin10 mutant. I would like to see this experiment to support their claim.

Response: As suggested by the reviewer, we have analyzed the stomatal development of wild type Col-0 grown in the medium containing $100\mu\text{M}$ Tre6P and 1% sucrose or only 1% sucrose for 10 days under long-day condition. The results showed that the stomatal index of plant grown in the medium only containing 1% sucrose is

approximately 37%, but it significantly reduced to 28% when plants were grown in the medium containing both sucrose and Tre6P (Supplementary Fig. 10a). Further, we showed that mutation in *TPS1* that is the key biosynthesis enzyme of Tre6P resulted in the significantly decreased stomatal index (Supplementary Fig. 10b). These results indicated that Tre6P negatively regulate stomatal development.

Supplementary Fig. 10. Tre6P negatively regulates stomatal development.

a, Quantification of the effects of Tre6P on stomatal index in wild type abaxial cotyledons. Seedlings of wild type Col-0 were grown in the liquid ½ MS medium containing 100μM Tre6P and 1% sucrose or only 1% sucrose for 10 days under 16h light/8h dark photoperiod (n=14). **b**, Mutation *TPS1* resulted in the increased stomatal index. Seedlings of wild type Ler and *tps1-11* were grown in the liquid ½ MS medium containing 1% sucrose for 10 days under 16h light/8h dark photoerpiod (n=20). Error bars indicate standard deviation (s.d.). Different letters above the bars indicated statistically significant differences between the samples (ANOVA analysis followed by Uncorrected Fisher's LSD multiple comparisons test, $p < 0.05$).

4. nuclear localization of KIN10-YFP

One of the key points of this paper is that KIN10-YFP is present in the nucleus in stomatal cells. Since the authors have the KIN10-T175D cloned it would be nice to see if the phosphomimic KIN10 is preferentially targeted to the nucleus compared to KIN10-YFP.

Response: Thanks for the suggestion. We have compared the subcellular localization of KIN10-YFP and KIN10^{T175D}-YFP in the transgenic plants. The results showed that the subcellular localization of KIN10^{T175D}-YFP is similar to that of KIN10-YFP in stomatal lineage cells and pavement cells, but the nuclear-to-cytoplasmic ratio of KIN10^{T175D}-YFP is significant lower than that of KIN10-YFP in mature guard cells (Supplementary Fig. 15a-f), indicating that the Thr175 phosphorylation of KIN10 reduces its nuclear localization in guard cells and has no significant effect on its subcellular localization in stomatal lineage cells and pavement cells.

Supplementary Fig 15. The T-loop phosphorylation had no significant effects on the subcellular location of KIN10-YFP in stomatal cells.

a-b, The subcellular location of pKIN10:KIN10-YFP and pKIN10:KIN10^{T175D}-YFP in Arabidopsis epidermal leaves. Seedlings of pKIN10:KIN10-YFP (**a**) and pKIN10:KIN10^{T175D}-YFP (**b**) were grown in ½ MS liquid medium for 5 days under long-day condition. Scale bars in confocal images represent 20 µm. **c-e**, Quantification of nuclear localization of KIN10-YFP and KIN10^{T175D} in different scale epidermal cells of panel (**a**) and panel (**b**), respectively. Nuclear and cytoplasmic KIN10-YFP or KIN10^{T175D}-YFP signals from more than 200 epidermal cells in 5 cotyledons were analyzed by ImageJ software. Scatter plot (**c** and **d**) and box plot (**e**) showed negative relationship of KIN10-YFP nuclear/cytoplasmic ratio and the size of epidermal cells. Asterisk between bars indicated statistically significant differences between the samples (Student's t-test, **** $p < 0.0001$). Error bar indicates S.D. **f**, Quantification of the stomatal index of wild type, pKIN10:KIN10-YFP and pKIN10:KIN10^{T175D}-YFP. Seedlings were grown in ½ MS liquid medium for 10 days under long-day condition. Error bars indicate standard deviation (s.d.). Different letters above the bars indicated statistically significant differences between the samples (ANOVA analysis followed by Uncorrected Fisher's LSD multiple comparisons test, $p < 0.05$).

5. original blots

Since many of the findings relate on quantifications of Western blots, I would like to see all the original blots as supplementary figure(s).

Response: Thanks for the suggestion. We have provided the original blots in the supplementary figure 20.

Medium concerns

6. Activation of SnRKs

In animal and yeast SnRKs get activated through energy depletion but this seems not to be the case here. I would like an extended discussion on this and also how the authors think that this finding translates in real nature.

Response: Response: Thank you for pointing this out, we have added the discussion as followed:

“SnRK1 and AMPK are conserved protein kinase and share an $\alpha\beta\gamma$ heterotrimeric structure; however, they show different responses to sugar starvation. Studies on animals have shown that AMPK is not only activated by increasing AMP/ATP and ADP/ATP ratios but also by glucose starvation (Lin et al., 2018). When glucose deprivation, the levels of the glycolytic intermediate fructose-1,6-bisphosphate (FBP) decreased and released aldolase, allowing the formation of an AXIN-based AMPK-activation complex to increase the Thr172 phosphorylation of AMPK (Lin et al., 2018). However, in our study, we showed that sugar starvation inhibits the activity of KIN10 by repressing its translation. Sucrose supply induced the accumulation of KIN10 protein by promoting its translation, and the increased KIN10 interacts and phosphorylates SPCH to promote stomatal development. In addition, under high sucrose concentrations, high levels of Tre6P were produced to inhibit the activity of KIN10 via attenuating its interaction with SnAKs. In animal cells, glucose starvation activates AMPK to repress mRNA translation through at least three different mechanisms, including phosphorylation of RAPTOR to inhibit the activity of mTOR, phosphorylation of eukaryotic elongation factor-2 (eEF2) kinase that can phosphorylate and inactivate eEF2, and phosphorylation of Unc-51 like autophagy activating kinase 1 (ULK1) to block the leucylation of tRNA(Leu) (Zhang et al., 2020). A recent study showed that SnRK1 also inhibits the mRNA translation by phosphorylating the cap binding proteins eIF4E and eIFiso4E (Bruns et al., 2019).

However, whether SnRK1 is involved in sugar starvation-reduced translation remains unclear.”

References:

Bruns AN, Li S, Mohannath G, Bisaro DM. Phosphorylation of Arabidopsis eIF4E and eIFiso4E by SnRK1 inhibits translation. *The FEBS journal* 2019, **286**(19): 3778-3796.

Lin SC, Hardie DG. AMPK: Sensing Glucose as well as Cellular Energy Status. *Cell metabolism* 2018, **27**(2): 299-313.

Zhang CS, Hardie DG, Lin SC. Glucose Starvation Blocks Translation at Multiple Levels. *Cell metabolism* 2020, **31**(2): 217-218.

If there is excess sucrose in the plant, why would the plant want to build more stomata? More stomata means more water loss so why making more stomata if there is already photoassimilates? In the manuscript the authors simply use generic terms like "finetuning of sugar and stomata to efficiently regulate photosynthesis" which in my opinion is not enough. In line 435 for example the authors talk about water-use efficiency, which I think is not appropriate. More stomata, more water loss, so sucrose induces stomatal pores to provide more carbon for photosynthesis thereby reducing their water-use efficiency under suboptimal light conditions (i.e. dawn and dusk, clouds etc.)

Response: Response: Thank you for pointing this out, we have added the discussion as followed:

“Previous studies showed that increased light quantity resulted in the significant increase of stomatal index (Casson et al., 2009). Plant grown under high photon irradiance had higher stomatal and epidermal cell densities than those plants grown under low photon irradiance. Plants grown in shade light also led to the reduced stomatal index and density comparing to those grown in normal light condition (Lake et al., 2001), suggesting a positive relationship between the light irradiance and stomatal development. In the present study, the positive relationship between light irradiance time and stomatal density was revealed. We found that the stomatal index of plants grown under the 16h light/8h dark photoperiod condition was significantly higher than that of plants grown under the 4h light/20h dark photoperiod condition. Exogenous sucrose treatment significantly increased the stomatal index of plants grown under 4h light/20h dark condition, indicating that the decreased stomatal index

of plants caused by short-time light irradiance maybe due to the insufficient sugar produced by photosynthesis. Light has been reported to regulate the stomatal development through phyB and PIF4 (Casson et al., 2009; Casson et al., 2014). High temperature has also been reported to activate PIF4 to directly repress the expression of *SPCH*, consequently inhibiting stomatal development (Lau et al., 2018). A recent study showed that KIN10 interacts with and phosphorylates PIF4 to promote its degradation (Hwang et al., 2019). These data indicated that KIN10 not only phosphorylates and destabilizes PIF4 to increase the transcript levels of *SPCH*, but also directly phosphorylates *SPCH* to increase the protein levels of *SPCH*, consequently promoting stomatal development. Taken together, these results revealed that light promote the stomatal development not only through its signal transduction pathway but also via controlling the photosynthesis to produce optimal sugar levels, which in turn finely regulates SnRK1 activity.”

References:

- Casson SA, Franklin KA, Gray JE, Grierson CS, Whitelam GC, Hetherington AM. phytochrome B and PIF4 regulate stomatal development in response to light quantity. *Current biology : CB* 2009, **19**(3): 229-234.
- Casson SA, Hetherington AM. phytochrome B Is required for light-mediated systemic control of stomatal development. *Current biology : CB* 2014, **24**(11): 1216-1221.
- Hwang G, Kim S, Cho JY, Paik I, Kim JI, Oh E. Trehalose-6-phosphate signaling regulates thermoresponsive hypocotyl growth in *Arabidopsis thaliana*. *EMBO reports* 2019, **20**(10): e47828.
- Lake JA, Quick WP, Beerling DJ, Woodward FI. Plant development. Signals from mature to new leaves. *Nature* 2001, **411**(6834): 154.
- Lau OS, Song Z, Zhou Z, Davies KA, Chang J, Yang X, et al. Direct Control of *SPEECHLESS* by PIF4 in the High-Temperature Response of Stomatal Development. *Current biology : CB* 2018, **28**(8): 1273-1280 e1273.

7. Discussion

The last sentence in the discussion is confusing. How would the repression of SnRK1Beta1 (what is this by the way?) reduce the nuclear translocation of KIN10? Please elaborate and clarify

Response: Thank you for pointing this out, we had rewrite the discussion as followed:

“In this study, KIN10-YFP fluorescence signal was present in all epidermal leaf cells. Nevertheless, the nuclear localization signal of KIN10-YFP was significantly enhanced in the smaller cells that might belong to the stomatal lineage cells, leading to phosphorylation of SPCH by KIN10 in the nucleus to increase SPCH protein stability and stomatal development. The published ChIP-Seq data of SPCH showed that SPCH could directly bind to the promoter region of *SnRK1β1* (Lau et al., 2014). Cell specific RNA-Seq data proved that *SnRK1β1* was down regulated in *pSPCH::SPCH-YFP* expressed cells (Adrian et al., 2015), indicating that SPCH represses the expression of *SnRK1β1* in stomatal lineage cells. *SnRK1β1* encodes a regulatory β-subunit of the heterotrimeric complex SnRK1. In Arabidopsis, there are three β-subunits, SnRK1β1, SnRK1β2 and SnRK1β3. The protoplast transient expression analysis has showed that SnRK1β1 and SnRK1β2, but not SnRK1β3, restrict the nuclear localization of KIN10 and reduce the promoting effects of KIN10 on the expression of *DIN6*, which has been used as a physiologically relevant readout of SnRK1 activity (Baena-Gonzalez et al., 2007; Ramon et al., 2019). In addition, myristoylated and membrane-associated regulatory β-subunits restrict the nuclear localization of KIN10 (Ramon et al., 2019). All these data elucidated that, in stomatal lineage cells, SPCH inhibits the expression of *SnRK1β1* to promote the nuclear localization of KIN10, and nuclear-localized KIN10 phosphorylates SPCH to increase the stability and activity of SPCH, thus forming a positive-feedback loop to promote the stomatal development.”

References:

- Adrian J, Chang J, Ballenger CE, Bargmann BO, Alassimone J, Davies KA, *et al.* Transcriptome dynamics of the stomatal lineage: birth, amplification, and termination of a self-renewing population. *Developmental cell* 2015, **33**(1): 107-118.
- Baena-Gonzalez E, Rolland F, Thevelein JM, Sheen J. A central integrator of transcription networks in plant stress and energy signalling. *Nature* 2007, **448**(7156): 938-U910.
- Lau OS, Davies KA, Chang J, Adrian J, Rowe MH, Ballenger CE, *et al.* Direct roles of SPEECHLESS in the specification of stomatal self-renewing cells. *Science* 2014, **345**(6204): 1605-1609.
- Ramon M, Dang TVT, Broeckx T, Hulsmans S, Crepin N, Sheen J, *et al.* Default activation and nuclear translocation of the plant cellular energy sensor SnRK1 regulate metabolic stress responses and development. *The Plant cell* 2019, **31**:1614-1632.

8. Figures:

Fig. 2a-c. I would like to see statistical analysis of the differences in 0% Sucrose conditions. There seems to be a difference between Col-0 and *kin10* and *kin11* also sucrose independently. If this is the case then this observation should be discussed.

Response: Thank you for pointing this out. We have repeated these experiments and re-statistically analyzed the difference of stomatal index among wild type, *KIN10-Ox* and *kin10* in the presence or absence of sucrose. These results showed there were no significant difference between wild type, *KIN10-Ox* and *kin10* mutant in the absence of sucrose, while in the presence of sucrose, overexpression of *KIN10* led to the increased stomatal index and knock-out of *KIN10* caused the decreased stomatal index comparing to wild-type plants. These maybe because of the low translation efficiency of *KIN10* under the carbon starvation, which resulted in the similar low levels of *KIN10* protein in wild type, *KIN10-Ox* and *kin10* mutants. While in the presence of sucrose, the protein translation of *KIN10* was no longer the rate-limiting factor, so the stomatal index increased in the *KIN10* overexpression plants but decreased in the *kin10* mutant. These results provide strong genetic evidence that sucrose can regulate the activity of *KIN10* through controlling the translation of *KIN10* to regulate the stomatal development.

Fig. 2. KIN10 plays crucial roles in sugar-induced stomatal development

a-c, Quantification of the stomatal index of wild type plants, *KIN10-Ox* and *kin10* mutants. Seedlings of wild type plants (Ler and Col-0), *p35S::KIN10-HA* (Ler background), *p35S::KIN10-myc* (Col-0 background), *kin10* mutant and *kin11* were grown in $\frac{1}{2}$ MS liquid medium with or without 1% sucrose for 10 days under long-day condition. Error bars indicate standard deviation (s.d.) (n=15-20). Different letters above the bars indicated statistically significant differences between the samples (ANOVA analysis followed by Uncorrected Fisher's LSD multiple comparisons test, $p < 0.05$).

Fig.4. Accumulated Tre6P under energy-rich conditions caused reduced phosphorylation of KIN10.

a-b, Immunoblot analysis of the total KIN10 proteins and phosphorylated KIN10 proteins in the presence of different concentrations (**a**) or different types (**b**) of sugar. Seedlings of *p35S::KIN10-myc* transgenic plants were grown in sugar free ½ MS liquid medium for 3 days and then treated with different concentrations of sucrose or different types of sugar for 6 hours. Myc-trap was used to enrich KIN10 protein by immunoprecipitation. **c**, MG132 and 3-MA had no significant effects on the levels of the phosphorylated KIN10. Seedlings of *p35S::KIN10-myc* plants were grown in sugar free ½ MS liquid medium for 3 days and then treated with or without 3-MA, MG132, and 1% or 5% sucrose for 6 hours. Myc-trap was used to enrich KIN10 protein by immunoprecipitation. Total KIN10 proteins were probed with anti-myc (**a-c**) antibody, phosphorylated KIN10 proteins were analyzed by anti-AMPKT172 antibody. Actin bands were used as loading control. **d**, Metabolomic analysis of sugar compounds seedlings of wild type that were grown in ½ MS sugar free medium for 3 days, then treated with 1% sucrose and 5% sucrose for 4 hours.

9. Methods:

Since some of the protein level quantifications are done by confocal microscopy i would appreciate if the authors include a "microscopy" section including type of microscope, settings (laser intensity, gain etc.). This paragraph should also contain details regarding the BIFC in Figure 4e.

Response: Thank you for pointing this out, we have added the description about the settings of microscope in the material and methods section.

10. Figure legends:

Throughout the figure legends the authors should indicate the number of replicates (both individuals and fields-of-views that were quantified).

Response: Thank you for pointing this out, we have added the description for the number of replicates in the figure legends.

Minor concerns

114: F6P has a significant effect (Fig. 1C), please correct.

Response: Thank you for pointing this out, we had changed it.

273ff: This is not a complete sentence, please change.

Response: Thank you for pointing this out, we had changed it.

295ff and 319ff. BiFC is introduced in line 319 rather than in line 295. I would suggest to introduce it in line 295ff

Response: Thank you for pointing this out, we had changed the introduction of BiFC to the place where it first used in this study.

325ff: in the text it says KIN10-myc and SPCh-YFP, in Figure 5D it is vice versa. Please correct.

Response: Thank you for pointing this out, we had changed it.

375ff. The authors mention qRT-PCR, but the figure shows quantifications with a transcriptional reporter. Please change.

Response: Thank you for pointing this out, we had changed it.

391ff. The authors claim here that they show dynamic regulation of SnRK1 activity to different environmental conditions. It is, however, only sucrose excess. Please correct.

Response: Thank you for pointing this out, we had changed it.

Figure 6 Figure legend: This legend in particular has quite some grammar mistakes and typos. Please correct.

Response: Thank you for pointing this out, we had changed it.

475; stabilisation rather than activation of SPCH.

Response: Thank you for pointing this out, we had changed it.

508: please indicate the ecotypes for kin10 and kin11

Response: Thank you for pointing this out. The ecotypes of *kin10* and *kin11* mutants are Col-0.

“The T-DNA insertion mutants of *GABI_579_E09* (*kin10/snrka1*) and *WiscDsLox384F5* (*kin11/snrka2*) were in Col-0 background and ordered from The European Arabidopsis Stock Centre. The *GABI_579_E09* mutant has been used the knock out mutant of *KIN10* to determine the its functions in hypocotyl elongation, root growth and stress response in plants. The mutant *WiscDsLox384F5* has not been used in previous study. Our RT-PCR analysis showed there are some amount of *KIN10* transcript in *kin10* mutant, but no *KIN11* transcript in *kin11* mutant. Immunoblot further confirmed that no significant amount of KIN10 protein and KIN11 protein could be detected in *kin10* and *kin11* mutants, respectively (Supplementary Fig. 6a-c).”

Reviewer #4 (Remarks to the Author):

This manuscript investigates for the first time what are the molecular mechanisms underlying stomatal development promoting effect of sucrose. Authors combine genetic and thorough biochemical analyses to tackle this question. Authors show that sucrose promotes KIN10 translation, which in turn phosphorylate SPEECHLESS (SPCH), master regulator of stomatal development. This leads to stabilization of SPCH protein and promotion of stomatal developmental pathway. Authors identify set of KIN10 phosphorylation sites in SPCH and show that phosphorylation of these sites is sufficient to enhance SPCH stability. Authors show that KIN10 and SPCH physically interact, are colocalized in the stomatal lineage, and both stability of SPCH and amount of phosphorylated KIN10 are inhibited in high sucrose conditions. Authors show that sucrose promotes accumulation of trehalose-6-phosphate (tre6P) in dose dependent fashion, which is known to prevent interaction between KIN10 and kinases acting upstream of KIN10 thereby inhibiting KIN10 activity. KIN10 has been shown to localize to nucleus as a response to metabolic stress. Surprisingly, KIN10 is nuclear-localized in stomatal lineage cells. Authors suggest that SPCH might directly repress SnRK1 β 1 in stomatal lineage cells, which inhibit nuclear localization of KIN10, hypothesis supported by previous studies. Authors suggest a model where upon low and intermediate sucrose conditions KIN10 translation is enhanced by sucrose. Kinases upstream of KIN10 phosphorylate KIN10, which in turn phosphorylates SPCH thereby stabilizing it. In high sucrose condition tre6P prevents

interaction between KIN10 and its upstream kinases and thus, further increase in SPCH levels and stomatal numbers.

Proposed mechanism would allow fine-tuning of the SPCH stability and lead to optimization of stomatal numbers depending on sucrose availability during leaf development. Such a mechanism would allow coordination of gas-exchange ability with photosynthesis status, two crucial processes for plant growth and development and for biomass production. The work presented and model constructed are novel and most of the data presented is convincing and impressive. However, there are some concerns related to stomatal developmental phenotypes and KIN10/SPCH coexpression data. These concerns are detailed below.

1. Authors use two sucrose metabolism mutants; *sex1* (high starch/low free sugar content) and *pgm* (low starch content/high free sugar content) to show that sucrose-induced changes in stomatal development are modified in the conditions with altered sucrose/starch levels. Stomatal indexes in these mutants are similar to wild-type if grown without sucrose supplementation. Stomatal development in *sex1* mutant is less sensitive for external sucrose whereas *pgm* mutant shows increased sensitivity to sucrose treatment. It is puzzling that dramatically altered cellular sucrose levels in these mutants do not lead to changes in the stomatal development but external sucrose treatment does. Please clarify this aspect.

Response: Thank you for pointing this out. The similar stomatal indexes of *pgm*, *sex1* and wild type in the absence of sugar maybe due to the low levels of KIN10 proteins. In the presence of exogenous sucrose, *sex1* converts more of the absorbed sugar into starch, thus showing a sugar less sensitive phenotype; while *pgm* mutants can produce more sugar to increase the protein levels of KIN10 and promote stomatal development. Considering the confusing and puzzling results of *sex1* and *pgm* and the suggestions of other reviewers, we removed the data of *pgm* and *sex1* from this revised manuscript.

2. Authors examine how different sugars modify stomatal index by growing seedlings for 10 days in liquid growth media supplemented with various sugars. However,

condition without any added sugars is not included into this experiment. Authors state “Sucrose exerts the most obvious positive effects on the development of stomata”. Data on wild-type grown in liquid growth media without any sugars for 10d should be added to this experiment in order to be able to evaluate effect of different sugars on stomatal index.

Response: Thank you for pointing this out, we have added the stomatal index data of wild type Col-0 grown in ½ liquid medium without any sugar for 10 days under the long-day condition.

Fig. 1. Sucrose promotes stomatal development and alters cell fate in Arabidopsis epidermis.

a, The stomatal index changes in response to light quantity. Seedlings of wild type plants were grown on ½ MS solid medium with or without 1% sucrose under 16h light/8h dark photoperiod with different light intensity for 10 days. **b**, Quantification of the effects of photoperiod and exogenous sucrose on the stomatal development. Seedlings of wild type plants were grown on ½ MS solid medium with or without 1% sucrose under different photoperiod with $100 \mu\text{Mol}\cdot\text{m}^{-2}\cdot\text{s}^{-1}$ for 10 days. **c-d**, Quantification of the effects of different sugars on stomatal index. Seedlings of wild type Col-0 were grown in ½ MS liquid medium containing 30mM mannitol (Man, 0.55%), sucrose (Suc, 1%), glucose (Glu, 0.54%), fructose (Fru, 0.54%), glucose-6-phosphate (G6P, 0.78%) and fructose-1,6-bisphosphate (FBP, 1%) (**c**), or different concentrations of sucrose or mannitol (**d**) under 16h light/8h dark photoperiod with $100 \mu\text{Mol}\cdot\text{m}^{-2}\cdot\text{s}^{-1}$ continuously for 10 days. Error bars indicate S.D. (n=15-20). Different letters above the bars indicated statistically significant differences between the samples (ANOVA analysis followed by Uncorrected Fisher's LSD multiple comparisons test, $p < 0.05$).

3. Authors show that wild-type plants show dramatic reduction in total stomatal numbers when grown 10 days in liquid growth conditions without sucrose compared to sucrose supplemented plants (Figure S1a-f). Seedlings shown in Figure S1a and S1d show smaller cotyledon size and tiny true leaves suggesting that growth has been severely compromised in these growth conditions. Since presence or absence of sucrose alone does not cause such a differences on plant growth (see for example Hanson et al (2001) Plant Molecular Biology 45: 247–262; figure 3) it seems likely

that the observed differences are caused by combination of liquid growth condition related stress and presence/absence of sucrose. Growing plants in liquid growth media may expose them to mild hypoxia and it clearly causes stress by severely reducing seedling growth (Figure S1a-f). Use of liquid growth media might be justified when performing large scale experiments such as biochemical assays, however, analysis of developmental phenotypes should be done as little stress causing conditions as possible. Cho et al. (2016) (J Ex Bot 67, 2745–2760) showed that submergence is sufficient to activate KIN10 and this causes broad changes in the phosphorylation status of the target proteins of KIN10. Authors should discuss previously mentioned findings on submergence related stress and KIN10 activation and evaluate what implications it may have on their data on stomatal development.

Response: Thank you for this excellent suggestion! To determine whether the liquid growth system has effects on the stomatal developments, we have compared the stomatal index of seedlings of wild type Col-0, *kin10* mutant and *KIN10-Ox* transgenic plants grown in ½ MS liquid medium or solid medium with or without 1% sucrose under 16h light/8h dark photoperiod for 10 days. In the absence of exogenous sucrose, the stomatal index of Col-0 grown in liquid medium was about 20%, but that in solid medium was 28%. However, in the presence of 1% sucrose, the stomatal index of Col-0 grown in liquid medium increased to nearly 40%, while the stomatal index of Col-0 grown in solid medium remained unchanged. These results indicated that liquid growth system may lead to carbon starvation for plants and thus inhibit stomatal development, while exogenous sugar treatment restores the carbon demand of plants, thereby promoting stomatal development.

When plants grown in solid medium under 16h photoperiod, we found exogenous sucrose and KIN10 have no promoting effects on stomatal development, which maybe due to the low activity of KIN10 under this growth condition. It has been reported that the prolonged darkness can activate KIN10. To examine the effects of sugar and KIN10 effects on stomatal development of plants grown on solid medium, the seedlings of Col-0, *kin10* and *KIN10-Ox* grown under 12h light/12h dark photoperiod and 4h light/20h dark photoperiod conditions were used to perform stomatal phenotype analysis. The results showed that, as the light irradiance time become shorter, the promoting effects of sugar and KIN10 on the stomatal development become more and more obvious. These results indicated that sugar

promoted the accumulation of KIN10, while liquid culture or prolonged darkness induced the phosphorylation of KIN10. The combination of multiple regulatory mechanisms control the activity of KIN10 and in turn optimize the stomatal development.

Fig. 2. KIN10 plays crucial roles in sugar-induced stomatal development

a-c, Quantification of the stomatal index of wild type plants, *KIN10-Ox* and *kin10* mutants. Seedlings of wild type plants (Ler and Col-0), *p35S::KIN10-HA* (Ler background), *p35S::KIN10-myc* (Col-0 background), *kin10* mutant and *kin11* were grown in ½ MS liquid medium with or without 1% sucrose for 10 days under long-day condition. Error bars indicate standard deviation (s.d.) (n=15-20). Different letters above the bars indicated statistically significant differences between the samples (ANOVA analysis followed by Uncorrected Fisher's LSD multiple comparisons test, $p < 0.05$).

Supplementary Fig. 7. KIN10 is required for sugar-induced stomatal development

a-c, Quantification of the effects of KIN10 on the stomatal development under different photoperiod conditions. Seedlings of Col-0, *p35S::KIN10-myc* and *kin10* were grown on ½ MS solid medium with or without 1% sucrose for 10 days under 16h light/8h dark photoperiod (**a**), 12h light/12h dark photoperiod (**b**), and 4h light/20h dark photoperiod conditions (**c**). Error bars indicate standard deviation (s.d.) (n=20). Different letters above the bars indicated statistically significant differences between the samples (ANOVA analysis followed by Uncorrected Fisher's LSD multiple comparisons test, $p < 0.05$).

In order to get more clearly description on the effects of liquid growth condition on the stomatal development, we have added a paragraph in the discussion section as followed:

“The phosphorylation of KIN10 has been reported to be activated by submergence Cho et al., 2016). However, submergence treatment had no significant effects on the transcriptional level and total protein level of KIN10, although it significantly increased the phosphorylation of KIN10 at the T-loop conserved threonine residue (Cho et al., 2016). Here, in this study, the liquid culture system was used to assess stomatal index and KIN10 activity. The significant promotion of stomatal development in plants grown in liquid medium with sucrose under 16h photoperiod might be due to the activation of KIN10 by mild hypoxia resulting from liquid culture, similar to submergence treatment. Accordingly, exogenous sucrose supply had no obvious promoting effects on stomatal index of plants grown on solid medium under the 16h light photoperiod condition, but significantly increased the stomatal index of plants grown on solid medium under the 4h photoperiod condition. This may be due to the activation of KIN10 under the prolonged darkness activates KIN10 (Baena-Gonzalez et al., 2007). Overall, these results indicated that sugar promotes the accumulation of KIN10, while liquid culture or prolonged darkness induce its phosphorylation. The combination of multiple regulatory mechanisms control the activity of KIN10 and thereby optimize the stomatal development.”

References:

- Cho HY, Wen TN, Wang YT, Shih MC. Quantitative phosphoproteomics of protein kinase SnRK1 regulated protein phosphorylation in Arabidopsis under submergence. *Journal of experimental botany* 2016, **67**(9): 2745-2760.
- Baena-Gonzalez E, Rolland F, Thevelein JM, Sheen J. A central integrator of transcription networks in plant stress and energy signalling. *Nature* 2007, **448**(7156): 938-U910.

4. Authors state (rows 157-161): “To examine how sugar regulates stomatal development, we analyzed the stomatal phenotypes of key components of different sugar signal transduction pathways. We found that several sugar signaling pathways

are involved in sugar-mediated stomatal developmental process. Here, in this study, we will focus on the function of SnRK1 in stomatal development.” Please consider, whether it is necessary to refer to the data not included to this manuscript. If this is necessary, please indicate that data is not shown in this article.

Response: Thank you for pointing this out, we have added the stomatal phenotypes of key components of sugar signal transduction in the supplemental Figure 4.

Supplementary Fig. 4. TOR is required for KIN10-promoted stomatal development in Arabidopsis epidermis.

a-b, Quantification of the effects of sucrose on stomatal index in wild type plants, *gin2-1* (a) and *tor-es* (b). Seedlings of wild type Col-0 and Ler, mutants *gin2-1* and *tor-es* were grown in ½ MS liquid medium containing 1% sucrose or 1% mannitol 9 days under 16h Light/8h Dark photoperiod. 10µM estradiol was added to liquid culture to induce deletion of TOR expression. Error bars indicate standard deviation (s.d.) (n=10-15). Different letters above the bars indicated statistically significant differences between the samples (ANOVA analysis followed by Uncorrected Fisher's LSD multiple comparisons test, $p < 0.05$).

5. Authors claim that MG132 treatment does not further decrease levels of phosphorylated KIN10 in the high sucrose conditions, however, the data presented in Figure 4c does not support this statement. Please quantify the data (as is done in Fig 3e) and revise the text according to results of the quantification.

Response: Thank you for pointing this out, we have repeated these experiments and did the quantitative analysis. The results showed that sucrose induced the accumulation of KIN10 protein at all concentrations, but phosphorylated KIN10 proteins decreased in response to high sucrose treatment. 3-MA treatment had no significant effects on this decrease, while MG132 treatment enhanced this decrease. These results indicated that the reduction in phosphorylated KIN10 caused by high sugar treatment is not due to the degradation of phosphorylated KIN10 protein.

Fig.4. Accumulated Tre6P under energy-rich conditions caused reduced phosphorylation of KIN10.

a-b, Immunoblot analysis of the total KIN10 proteins and phosphorylated KIN10 proteins in the presence of different concentrations (**a**) or different types (**b**) of sugar. Seedlings of *p35S::KIN10-myc* transgenic plants were grown in sugar free ½ MS liquid medium for 3 days and then treated with different concentrations of sucrose or different types of sugar for 6 hours. Myc-trap was used to enrich KIN10 protein by immunoprecipitation. **c,** MG132 and 3-MA had no significant effects on the levels of the phosphorylated KIN10. Seedlings of *p35S::KIN10-myc* plants were grown in sugar free ½ MS liquid medium for 3 days and then treated with or without 3-MA, MG132, and 1% or 5% sucrose for 6 hours. Myc-trap was used to enrich KIN10 protein by immunoprecipitation. Total KIN10 proteins were probed with anti-myc (**a-c**) antibody, phosphorylated KIN10 proteins were analyzed by anti-AMPKT172 antibody. Actin bands were used as loading control. **d,** Metabolomic analysis of sugar compounds seedlings of wild type that were grown in ½ MS sugar free medium for 3 days, then treated with 1% sucrose and 5% sucrose for 4 hours.

6. Authors say that“ KIN10 displayed the ubiquitous expression pattern in all epidermal cells, and mainly localized in the nucleus in guard cells and the smaller cells less than 200 μm^2 , while distributed in the cytoplasm in the pavement cells (Supplementary Fig. 5a-c)”. It is great that quantification has been performed. However, these images do not show convincingly the cytosolic localization or absence of nuclear localization in the pavement cells. Figure S5a shows YFP signal in DAPI stained elongated pavement cell nucleus. Please revise the text or replace with more representative images.

Response: Thank you for pointing this out, we have repeated this experiment under the same conditions and made new graphs (Fig. S14).

7. As authors state, it has been shown that hypoxia causes nuclear localization of KIN10 and authors see nuclear localized KIN10 in stomatal cells. If seedlings have been grown in liquid growth conditions, it seems possible that plants had experienced certain extent of hypoxia related stress and this would explain the nuclear localization. It is not indicated how plants have been grown in this particular experiment (Figure

5e). Please describe in the figure legend or methods systematically used growth conditions. Subcellular localization of KIN10 should be studied also in seedlings grown on solid growth media to rule out that nuclear localization is due to stress caused by liquid growth conditions.

Response: Thank you for this excellent suggestion. We have compared the subcellular location of KIN10-YFP in seedlings grown in liquid medium and solid medium. Quantification of the nuclear and cytoplasmic KIN10-YFP signals revealed a similar ratio for the most of epidermal cells of seedling grown in either liquid medium or solid medium. Although KIN10-YFP nuclear/cytoplasmic signaling were enhanced for the $<200\mu\text{m}^2$ cells in liquid culture compared to that in solid culture, it always performed higher nuclear/cytoplasmic ratio for smaller cells to that in larger cells in both liquid and solid condition, which indicating the liquid growth condition has no significant effects on the subcellular location pattern of KIN10-YFP in epidermal cells.

Supplementary Fig 14. High levels of nuclear-localized KIN10 in guard cells and smaller cells.

a-b, The subcellular location of pKIN10:KIN10-YFP in Arabidopsis epidermal leaves. Seedlings of pKIN10:KIN10-YFP were grown in ½ MS solid medium (**a**) or in ½ MS liquid medium (**b**) for 3 days under long-day condition. Scale bars in confocal images represent 20 µm. **c-e**, Quantification of nuclear localization of KIN10 in different scale epidermal cells of panel (**a**) and panel (**b**), respectively. Nuclear and cytoplasmic KIN10-YFP signal from more than 200 epidermal cells in 5 cotyledons were analyzed by ImageJ software. Scatter plot (**c** and **d**) and box plot (**e**) showed negative relationship of KIN10-YFP nuclear/cytoplasmic ratio and the size of epidermal cells. Error bars indicate standard deviation (s.d.). Different letters above the bars indicated statistically significant differences between the samples (Tukey's multiple comparisons test, $p < 0.05$).

Minor comments:

8. Please indicate number of samples used for each experiment in the figure legend as a numbers ($n=x$). Currently sample numbers are indicated for most experiments as a dots in the graph, however, counting these dots is tedious for the reader. In addition, for some experiments this information is completely lacking, for example Figure 1d.

Response: Thank you for pointing this out, we have added the details in the figure legend.

9. Supplementary files include data on cell-size distribution (data Fig S1g and Fig S3g). Currently nothing is mentioned on this data in the results or methods sections. Please remove or refer to this data in the text and explain how it has been produced in the methods.

Response: As suggested, we have added the description about the results of Fig. S1 and S3g (Fig. S2 and S5, in the revised manuscript) in the Results section, and added the detail to explain how to produce these data in the Material and Methods section.

10. Please consider indicating significant differences between samples using consistent style throughout the figures. Currently, two styles (letters and asterisks) are used (see for example Fig 1b vs Fig 2 a-c or Fig 1d vs Fig 3e). Also, please consider adding control line to consistent position in the panels. It is misleading that in some panels it is shown in the middle of the samples (Figure 2a-c) and sometimes on the left side (Figure 4g-h).

Response: Thank you for pointing this out, we have changed them.

11. Most quantifications of stomatal marker gene expression are done with plants grown in liquid growth conditions (Figure S2, Figure 2e-f). For subset of them, it is not indicated whether solid or liquid conditions has been used (For example Figure 1e-l). Please indicate systematically how plants have been grown.

Response: Thank you for pointing this out, we have added the details in the figure legend.

12. Figure 3e is mislabeled in legend to be Figure 3f. Please revise the lettering to match the figure.

Response: Thank you for pointing this out, we have changed it.

13. Supplementary Figure 4 shows changes in metabolic pathways as a response to sucrose treatment. Currently the title is “Metabolism pathway of Arabidopsis seedling treated with different concentrations of sucrose”. Please revise the title of the figure legend and legend text to indicate what concentration of sucrose has been used. Please also explain how to interpret the data shown in this figure.

Response: Thank you for pointing this out, we have changed it.

14. “Co-localization analysis with pKIN10::KIN10-YFP/pSPCH::SPCH-RFP showed that KIN10 and SPCH were simultaneously distributed in the nucleus in the stomatal lineage cells, indicating that KIN10 physically interacted with SPCH in stomatal lineage cells (Fig 5e).” Co-localization data does not show that two proteins physically interact. Please revise the sentence accordingly or refer to another figure panel.

Response: Thank you for pointing this out, we have changed it.

15. Figure 5e, legend: Legend states that anti-YFP and anti-myc antibodies were used. The Figure panel is labeled with α -myc and α -GFP. Please revise this image or legend.

Response: Thank you for pointing this out, we have changed them.

16. Figure S8a: Please indicate what sucrose concentration has been used for the treatment. Currently only length of the sucrose treatment is indicated in the Figure legend.

Response: Thank you for pointing this out, we have added the details in the figure legend.

17. “RT-qPCR analysis showed that the short time sucrose treatment had marginal effects on the transcription of SPCH, but significantly induced the protein accumulation of SPCH protein (Supplementary Fig 8b).” Fig S8b does not show qRT-PCR data, but Fig S7b does so please refer here also to Fig S7b.

Response: Thank you for pointing this out, we have changed it.

18. “Consisten with these results, the pBASL::BASL-GFP-labeled cells representing the dividing MMCs gradually increase as the increased concentrations of sucrose, while the pMUTE::MUTE-GFP-labeled cells representing the differentiated cells are most abundant at 1% sucrose concentration and are relatively lower at other high or low concentrations of sucrose.” Please correct the description of expression domains. MUTE-GFP labels guard mother cells (cell committed to form stomata) whereas BASL-GFP labels MMCs, meristemoids and the larger SLGCs. Please also revise latter part of the sentence, which is currently hard to interpret.

Response: Thank you for pointing this out, we have changed them.

REVIEWER COMMENTS

Reviewer #1 (Remarks to the Author):

In this revision, authors have included new controls, increased the quality of some figures, provided all the missing experimental details, and added new data, altogether leading to a significant improvement of the manuscript.

The work now clearly demonstrates that KIN10 promotes stomata development under certain conditions, and that this involves SPCH phosphorylation and consequent stabilization. This is shown for plants at different developmental stages and grown under different conditions that are likely associated with some level of stress (very short days or liquid cultures, which are likely hypoxic). Furthermore, a *tps1* mutant where KIN10 is more active due to reduced accumulation of T6P, also shows a similar phenotype of higher stomata index. And the phenotypes of plants expressing phosphomutant versions of SPCH provide further and very clear evidence of the relevance of this mechanism for stomata development. Altogether, this part is sound and convincing. This is on its own really exciting, highly novel and very well demonstrated, so in my opinion would be sufficient to merit publication in this journal.

What I think makes the manuscript still very confusing is the effect of sucrose on KIN10. By trying to "force" the manuscript into a story centered around how KIN10 is regulated rather than around its effects on stomata development, the storyline becomes very confusing and also with conflicts and misleading conclusions that can detrimentally impact this field in the long term, if published like this.

The main problem is the conflict between what is observed at the phenotype level and at the molecular level:

1) AT 5% SUCROSE, there is a clear effect of sucrose/T6P on KIN10 at the molecular level (Fig. 4; high T6P accumulation and dephosphorylation of KIN10 due to reduced SnAK-KIN10 interaction; this likely results into lower KIN10 activity), but no KIN10-dependent plant phenotype is shown

2) AT 1% SUCROSE, the phenotype of plants overexpressing KIN10 is very clear (higher SI), but no impact on KIN10 activity is shown at the molecular level. The authors claim that sucrose promotes KIN10 activity at 1% based on the fact that there is more KIN10 accumulated (Fig. 3a and c). However, in those conditions the expression of TPS5 actually tells the opposite (Fig. 3b). This gene is usually induced by sugar and repressed by KIN10 activity, suggesting that in these conditions KIN10 is actually being inhibited. Most of the phenotypic data in the data is from 1% sucrose (at least the data involving KIN10), so to make a point about sucrose having an effect on KIN10 activity at the molecular level, 1% suc would be the most relevant condition and not 5%

Altogether, I think the main "problem" comes from the fact that the authors are trying to extrapolate the results of short sugar feeding (where KIN10 is accumulated in response to 1% suc, but it is likely less active), to the results of long-term growth, where KIN10 activity promotes stomata development. And they may not be extrapolatable. More correct would be to analyze KIN10 accumulation and its phosphorylation under the conditions where the phenotype is observed (growth over several days on liquid medium with/without 1% sucrose). I don't see this as necessary, because I think there is already a beautiful story on the stomata development part and the effect on SPCH.

What I suggest simply is that the weight of the story is removed from the regulation of KIN10 by sucrose and placed on the effect of KIN10 on SPCH and stomata development.

1) This would start from the title (which I think would be more appropriate to change to something along these lines: KIN10 promotes stomata development through stabilization of the SPCH

transcription factor)

2) Then, the results of 5% sucrose might have to be removed

3) The parts of the Discussion that are misleading should be either removed or substantially rephrased:

- 68-69. Perhaps better to indicate "Under certain conditions, sucrose promotes stomatal formation, but the mechanism remains unclear"
- 69-70. "Here, we showed that sucrose dynamically regulated stomatal development by controlling the activity of KIN10". Given that an effect by sucrose on KIN10 phosphorylation (potentially translating into lower activity) is shown only at 5% sucrose, this is a large generalization
- Lines 439-442. "Our genetic and biochemical analyses demonstrated that, despite its conserved functions, SnRK1 regulation has evolved differently from that of analogous SNF1 and AMPK and that sugar produced by photosynthesis finely regulates the activity of SnRK1 in response to different sucrose conditions". This paragraph has implicit the idea that sucrose induces KIN10 activity, with which I strongly disagree and do not think the authors provide support.
- Lines 471-474. "Increasing the activity of KIN10 by mimicking the phosphorylation of a conserved threonine residue in the activation T-loop or overexpression of SnAK2, an upstream kinase of KIN10, also led to increased stomatal index".
 - The T175D mutation was never shown to result in enhanced KIN10 activity in vivo, so please just say that expression of the T175D variant led to increased SI
 - Also, no evidence is provided for increased KIN10 activity in SNAK OE lines (see minor comments) so just say that overexpression of SnAK2 led to increased SI
- Lines 492-498. "Our genetic and biochemical assays showed that the T-loop phosphorylation is important for the KIN10-regulation of stomatal development in Arabidopsis. Low concentrations of sucrose (0.5% and 1%) induced the accumulation of phosphorylated KIN10 protein and induced the stomatal formation. On the contrary, high concentrations of sucrose (3% and 5%) led to increased levels of Tre6P and decreased levels of phosphorylated KIN10 proteins, subsequently reducing the stomatal formation". By mixing what happens at the level of protein accumulation (1%) with what happens at the level of T-Loop phosphorylation (5%) under the common denominator of "levels of phosphorylated KIN10" this becomes very misleading and gives the idea that T-loop phosphorylation per se is changing in these two conditions, which is not true.
- Lines 498-502. "Increasing the activity of KIN10 by overexpression of its upstream kinase SnAK2 or expression the KIN10T175D, which mimics the phosphorylation of KIN10 at T175, both led to the increased stomatal index. These results indicated that the T-loop phosphorylation of KIN10 plays a critical role for sugar-promoted stomatal development". See comments for lines 471-474
- Lines 559-579

Overall the discussion should significantly reduce all these parts related to KIN10 regulation and how this contrasts with the situation in animals, since there is not sufficient evidence to support these claims. Instead the relevance of the uncovered mechanisms of promoting stomata development though KIN10 could be discussed a bit in more detail. It makes all the sense that KIN10 promotes this process, as formation of stomata is the first step towards carbon assimilation. Also the effect of sucrose can be reasoned and justified without entering into how that translates into KIN10 activity. Under very short photoperiods, soil-grown plants induce KIN10 activation, allowing the use of resources to develop more stomata. In hypoxic liquid cultures of very young seedlings, KIN10 activation is probably not enough to support stomata development, simply

because there are no resources to do so, and hence the effect of KIN10 activation is only seen under 1% sucrose (as compared to no sucrose control). Altogether the KIN10-Stomata connection is a really exciting and unique finding that should be emphasized much more!

Other minor things:

1) There are still no molecular analyses provided of the plants used in Fig. 4h-i. How many lines were obtained for the lines in Fig. 4h and did they all behave similarly? Without seeing the level of KIN10 expression for the line expressing the WT vs. the T175D variant is not possible to conclude whether the phenotype differences are indeed due to differences in phosphorylation or in the level of expression of these variants

2) For the SnAK OE. The authors included now new data in Suppl Fig. 11 that shows enhanced accumulation of phosphorylated KIN10 in the SnAK OE lines. However, without showing the total accumulation of KIN10 in these lines this is not very informative and does not allow to make claims related to KIN10 activity/phosphorylation

3) In the Nature publishing forms please include info on the KIN10 T-loop antibody, and about the ANOVA analyses

Reviewer #2 (Remarks to the Author):

I am impressed by the amount of work that was done in such a short time and the high quality of the new data that further strengthen the major conclusions. Most of the comments have been addressed. My only major concern remaining is that conclusions about the effects of sucrose on SnRK1 T-loop phosphorylation and activity should be phrased more carefully/correctly in some places. Lower sucrose levels lead to accumulation of both total and T-loop phosphorylated SnRK1. It should be mentioned more explicitly that there is no effect on the relative T-loop phosphorylation levels. For example, the (new) title on line 258 and the sentence on line 268-269 are still somewhat misleading. Of course, this increase could still lead to an overall increase in SnRK1 activity. Conversely, the effects of carbon starvation conditions on stomatal development are surprising as they should activate SnRK1. For example, on line 566 in the discussion, it is mentioned that the data show that sugar starvation inhibits the 'activity' of KIN10. Therefore, it would still be nice to have an independent readout of SnRK1 activity (in addition to SPCH phosphorylation and stability) in response to the different levels of sugars in the experimental system used, such as qPCR data from selected target gene expression. Some minor language editing is still needed.

Reviewer #3 (Remarks to the Author):

The authors undertook an impressive effort to address my concerns. I am fully convinced by their data regarding the effect of T6P (Fig. S10) and the nuclear localization (Fig. S15), which was even tested in liquid culture and on plates (Fig. S14). This being said I am still not convinced by the main conclusion that sugar regulates stomatal development.

Main concerns:

1. Under 16h and 12h light conditions there is almost no difference if Col-0, kin10 and Kin10-OE are grown on plates and there is only a difference once the photoperiod is changed to only 4h (Fig. S7). These are extremely unnatural conditions for a plant to grow in and do not occur in nature. The same is true for submerged plants grown in liquid culture. Due to a lack of gas exchange this is essentially a carbon starved situation much like the 4h time period. Therefore I think what the

authors observe here is a sugar-dependent role of Kin10 if a plant is darkness-stressed and/or carbon-starved, a situation that never occurs in real life. In my opinion the title of the manuscript implies that sugar optimizes stomatal development in general but actually the MS shows that sugar optimizes stomatal development in extended darkness and when submerged. I would expect that this fact is at least discussed and acknowledged. You could for example discuss that this sugar-dependent signalling could only have been revealed using these extreme carbon-starved conditions and that this module likely coordinates density to photoassimilates and light regime even under natural conditions. I would also like the fact to be stated that liquid culture enhances the effect due to submergence and the resulting strong reduction of gas exchange and thus assimilation.

Minor concerns:

2. On line 585, the authors say that "in the present study, positive relationship between light and stomatal density was revealed" after the actually introduce two studies that showed that years ago in the sentences above. They rather reveal that the lack of sugar is probably affecting this but only if the plant is basically carbon starved.

Reviewer #4 (Remarks to the Author):

Bai et al., have extensively revised their manuscript and provide data from several new experiments. Current manuscript shows that KIN10 promotes stomatal development through phosphorylation of stomatal master regulator SPEECHLESS (SPCH) and this interaction is dependent on sucrose dosage. Authors also show that sucrose promotes KIN10 translation, however, high sucrose concentration reduces KIN10 activity (by decreasing amount of phosphorylated KIN10). Upstream kinases (SnAKs) of KIN10 are required for KIN10 activity: high concentrations of sucrose (in growth media) cause accumulation of Tre6p, which prevent SnAKs-KIN10 interaction and thus, lead to reduction in amount of active KIN10, reduced SPCH stability as well as decreased stomatal index. Authors show that exposure to Tre6p leads to decreased stomatal index whereas *tps1-11* mutant (key enzyme in Tre6p biosynthesis) display increased stomatal index supporting their model. Further, authors identify 4 residues in SPCH which are targets of KIN10 mediated phosphorylation and show that phosphorylation of these residues is required for SPCH stability.

Major comments

1) Role of KIN10 in stomatal development

Authors have added data on several new experiments where they investigate how different growth conditions such as light intensity and duration modify sucrose dependent stomatal development. This is mostly very nice additional data. They also compare effects of liquid and solid growth media on stomatal development in KIN10-OE/kin10/wild-type in combination with sucrose treatment and different day length.

Row 177-179: "The results showed that overexpression of KIN10 significantly induced stomatal development in the presence of sucrose, particularly under the 4h light/20h dark photoperiod condition (Supplementary Fig. 7a-c)."

Based on their data authors conclude that (Row 185-186): "These results indicated that KIN10 positively regulates stomatal development."

Finally, authors study stomatal development in same plant lines grown in the soil and use 5th rosette leaves for analysis:

Row 183- 186: "The stomatal index of rosette leaves of KIN10-Ox was higher, and that of kin10 mutant was lower than the stomatal index of wild type rosette leaves (Supplementary Fig. 8a-d). These results indicated that KIN10 positively regulates stomatal development."

Authors come up with a model where mild hypoxia (caused by liquid growth media) or darkness cause KIN10 activation whereas sucrose induces KIN10 translation. Therefore, in the absence of KIN10 activation stomatal development is not affected by sucrose dosage (on plate). However, there is some confusing and conflicting results on kin10, KIN10-OE and wild-type grown on plate with and without sucrose (cotyledons, solid vs liquid growth media) and in soil (leaves). Please describe this currently mostly ignored data in the text and hypothesize what could be underlying reason for these observations:

- Why kin10 behaves differently under the 12h light/12h dark and 4h light/20h dark photoperiod conditions on solid vs liquid growth media? On solid growth media kin10 shows elevated stomatal index compared to wild-type and KIN10-OE without sucrose whereas in liquid growth media there is no difference between these lines without sucrose. This is not in line with stomata promoting function of KIN10 authors are suggesting!

- Why kin10 do not respond at all to sucrose when grown on solid growth media (even when it is grown under the 4h light/20h dark photoperiod condition unlike other analyzed lines) whereas in liquid growth media it shows elevated stomatal index as a response to sucrose (although less than other lines)?

- Why soil grown wild-type, KIN10-OE and kin10 plants show different stomatal indexes in rosette leaves without sucrose treatment whereas on plate grown plants do not show difference in cotyledon stomata index (KIN10-OE) or show even difference in opposite direction (kin10) in same light conditions? Could this be related to developmental program of the organ or differences in the growth conditions?

2) Endogenous sucrose vs sucrose treatment

Authors do majority of the experiments with liquid growth media supplied with sucrose. To their credit, authors have investigated how different growth conditions such as light intensity and duration modify sucrose dependent stomatal development. They also compare effects of liquid and solid growth media on stomatal development in combination with sucrose treatment. This data shows that sucrose combined with liquid growth media enhances nuclear localization of KIN10 specifically in small stomatal lineage cells (likely to be stomatal precursor cells) compared to sucrose combined with solid growth media. Authors come up with a model where mild hypoxia (caused by liquid growth media) or darkness cause KIN10 activation whereas sucrose induces KIN10 translation. Therefore, in the abundance of KIN10 but absence of active KIN10 stomatal development is not affected by sucrose dosage. However, plants are capable to produce sugar by themselves and do not rely on external sugar source. Thus, it is unclear how well this data applies to the situation in soil grown plants. Please elaborate these aspect in the discussion.

3) Sugar signalling mutant data

Authors have added new data on few additional lines presumably participating in sugar signalling (tor-es, gin2-1). However, description of these genes and their role in sugar signalling are completely missing from current manuscript. Also these lines are not described in the materials and methods. Data is only briefly described:

Row 157 -159: "We found that several sugar signalling pathways are involved in the sugar-mediated stomatal developmental process (Supplementary Fig. 4a, b)"

Please consider whether this is necessary information for this paper. If it is, please explain what are the lines shown in the Supplementary Fig. 4a & b, and how these results support the statement in the manuscript (Row 157 -159).

Minor comments

(Most are related to writing - rigorous proof-reading would be helpfull)

1. Row 53- 56: "Plant stomata represent an excellent system..., due to their simple, flexible, and developmental trajectory"

Please modify: "Plant stomata represent an excellent system..., due to their simple and flexible developmental trajectory"

2. Row 56-57: "The stomatal lineage in *Arabidopsis thaliana* initiates by asymmetric divisions of undifferentiated meristemoid mother cells (MMC)"

Please modify: "The stomatal lineage in *Arabidopsis thaliana* is initiated by asymmetric division of undifferentiated meristemoid mother cell (MMC)"

3. Row 68: "Sugar has been reported to promote the stomatal formation"

Please use either "to promote stomatal development" or "to promote stomata formation"

4. Row 74: "Tre6P was significantly accumulated"

Please rephrase: "Tre6P levels were significantly increased"

5. Row 83: "Stomata, the pores on plant epidermis for gas exchange with the atmosphere"

Please rephrase, for example: "Stomata, the pores on plant epidermis, which facilitate gas exchange between the plant and atmosphere"

6. Row 162-166: "Because the KIN10-Ox genetic material that we obtained from Arabidopsis Biological Resource Center (ABRC) is in the Landsberg (Ler) background, in which ERECTA (ER), a key component of stomatal development, was mutated leading to more stomata".

Please rephrase: "Because the KIN10-Ox genetic material that we obtained from Arabidopsis Biological Resource Center (ABRC) is in the Landsberg (Ler) background, in which ERECTA (ER), a key component of stomatal development, was mutated, it showed increased number of stomata."

7. Row 167-170: "The results showed that, in the presence of sucrose, overexpression of KIN10 in the Col-0 background also led to the increased stomatal index, more stomata number in a whole cotyledon and higher ratio of clustered stomata comparing to Col-0 plants (Fig. 2b and Supplementary Fig. 5a-i)."

Please rephrase: "The results showed that in the presence of sucrose, overexpression of KIN10 in

the Col-0 background also led to the increased stomatal index, elevated stomatal numbers in whole cotyledons, and higher ratio of clustered stomata compared to Col-0 plants. (Fig. 2b and Supplementary Fig. 5a-i)."

8. Figure S5c is labelled with number of stomata/leaf. However, legend says that cotyledons have been analyzed. Please modify and label with number of stomata/cotyledon

9. Row 208-201: "This might be due to the accumulation of SPCH inducing cell division to form many small cells of which only a few will eventually develop into stomata"

This sentence is hard to understand. Please rephrase.

10. Row 328: "florescent protein"

Please modify: "fluorescent protein"

11. Figure 4e-f: Please explain in the legend shortly how tobacco samples have been treated (for example, how long sucrose treatment was used).

12. Row 335-337: "These results indicated that the decreased phosphorylation of KIN10 by high sucrose treatment is due to the accumulation of Tre6P, which reduced the binding affinity of KIN10 to its upstream kinase SnAKs."

These results do show that sucrose induces Tre6P accumulation (Fig 4d) and reduces KIN10 phosphorylation (Fig 4a) but do not show that there is causality between the two. Please modify description of the findings.

13. Row 345-347: "Mutation of TPS1 resulted in a significantly increased stomatal index comparing to that of wild type plants in the presence of sucrose (Supplementary Fig. 10b)."

Tre6P should accumulate above 1% sucrose conditions and this prevents KIN10 activation leading to destabilization of SPCH according to model. Here effects of 1 % sucrose has been tested. It would be more interesting to see whether tsp1 mutant lacks the reduction of stomatal index in high sucrose conditions and whether it shows elevated SPCH levels in high sucrose conditions.

14. Row 373-376: "KIN10 displayed the ubiquitous expression pattern in all epidermal cells, and it was mainly localized in the nucleus of guard cells and the smaller cells less than 200 μm^2 , while distributed in the cytoplasm in the pavement cells (Supplementary Fig. 14a-e)."

It is interesting that there is significantly increased nuclear localization of KIN10 in smaller cells in liquid growth media with 1 % sucrose. These cells contain stomatal precursor cells and thus are the most relevant cells for stomatal development. This data should be discussed in the text more clearly since it suggests that KIN10 activity in stomatal lineage cells is very sensitive for the use of liquid growth condition.

15. Figure 5g . Legend "Quantification of stomatal index in wild type plants and different SPCH

complemented transgenic plants (n=10)."

Please explain in the legend genotypes and names of the transgenes.

16. Supplementary Fig. 19a. legend

Please explain whether "medium" refers to liquid or solid media (Fig S19a).

17. Row 430-432: "More importantly, SPCH-4A performed weaker protein stability compared to wild type SPCH and SPCH-4D in developing cotyledon epidermis (Fig. 5h, Supplementary Fig. 19c,e)."

Fig19.e It is clear that starting dosage of SPCH variants are very different, however, 24h sucrose treatment seems to increase levels of both SPCH-myc and SPCH4A-myc. Please quantify the increase in order to see whether the increase is similar or different. This would tell whether or not sucrose dependent induction of SPCH stability is dependent on those 4 mutated residues.

18. Row 439-442: "Our genetic and biochemical analyses demonstrated that, despite its conserved functions, SnRK1 regulation has evolved differently from that of analogous SNF1 and AMPK and that sugar produced by photosynthesis finely regulates the activity of SnRK1 in response to different sucrose conditions."

This sentence does not accurately describe the data in this manuscript – first, it suggests that this study has compared regulatory pathways between SNF1, AMPK, and SnRK1, which is not correct. Second, it claims that authors have shown that sucrose produced by photosynthesis regulates the SnRK1 as a response to sucrose conditions (whatever this means), which is not the case. Please rephrase to be more accurate, for example like this: "Our genetic and biochemical analyses demonstrated that activity of SnRK1 is regulated in response to different sucrose conditions. This data suggest that despite its conserved functions, SnRK1 regulation has evolved differently from that of analogous SNF1 and AMPK and further, sugar produced by photosynthesis could finely regulate the activity of SnRK1."

19. Row 452-456: "Under the condition of no sucrose, there is a low level of KIN10 protein in plants, which reduces the stability of SPCH and decreased stomatal index. When sucrose supply reaching to basal conditions, sucrose not only promotes cell division to generate more small epidermal cells, but also induces the accumulation of KIN10 protein by promoting its translation efficiency"

When is plant experiencing condition without sucrose? During night time when there is no light? Please elaborate this, and take consideration also in your model!

20. Row 548-550: "As the concentration of sugar increases, the total number of epidermal cells in the leaves gradually increases, but the number of stomata reaches the peak at 1% sucrose concentration, and then gradually decreases."

I wonder what is the physiologically relevant sucrose concentration? Does cellular sucrose levels ever reach the similar levels as when growing on media containing 5 %? Could plants break down storage carbohydrates when modification of stomatal development if needed?

21. Row 455-457: "Second, nuclear-localized KIN10 is highly enriched in guard cells and the cells less than 200 μm^2 that may belong to the stomatal lineage cells, which was further verified by the colocalization of KIN10 and SPCH in nucleus of these cells."

KIN10 is even more nuclear localized in stomatal cells in liquid conditions – is colocalization of SPCH and KIN10 tested also in solid conditions (where sucrose induced stomatal developmental response is absent?)

22. Row 544-548: "When sugar levels of sugar in plants are low under low light quantity, short time light irradiance and prolonged darkness, the expansion and division of epidermal cells in plant leaves are inhibited, resulting in the defective leaf development and low stomatal index."

It would be destructive for plants to show defective leaf development in low light conditions or night time! Increase in subset of stomatal precursor divisions (amplifying and entry divisions) leads to reduced stomatal index whereas increased number of another type of stomatal precursor division (spacing division) leads to elevated stomatal index. This is because these three types of stomatal divisions produce different ratios of pavement cells and stomata during their division activity. Therefore simply reduction of all epidermal divisions or all stomatal cell divisions do not lead to reduced stomatal index. Please modify in order to take this in consideration.

23. Row 551-522: "pBASL::BASL-GFP-labeled cells that represent MMCs, meristemoids and the larger SLGCs in a concentration-dependent manner"

Please modify: "pBASL::BASL-GFP-labeled cells that represent MMCs, meristemoids, and SLGCs"

Responses to comments by reviewers

We wish to express our deep appreciation for the constructive comments on our manuscript by the reviewers. In response to these comments, we have performed additional experiments and revised the text extensively to improve our manuscript. Specifically, we have made the following changes:

1. We have changed the title to “KIN10 promotes stomatal development through stabilization of the SPEECHLESS transcription factor”.
2. We have removed the results of 5% sucrose (Fig. 1d, Fig. 4a-i and Sup Fig. 9 in the old version of manuscript) according to the suggestions of reviewers.
3. In order to emphasize the functions of KIN10 on stomatal development, we have divided the results of KIN10 and SPCH into two figures. The figure 4 introduced that KIN10 interacts with SPCH in the nucleus of stomatal lineage cells; the figure 5 introduced that KIN10 phosphorylates and stabilizes SPCH to promote stomatal development.
4. We have changed our working model (Fig. 6).
5. We have analyzed the stomatal phenotype of the *pKIN10::KIN10-myc*, *pKIN10::KIN10^{T175D}-myc* and *pKIN10::KIN10^{T175E}-myc* transgenic plants to show mutation of Thr175 to Asp or Glu increases the promoting effects of KIN10 on stomatal development (Supplementary Fig. 9a-b).
6. We have analyzed the protein levels of KIN11 in wild type and *kin10* mutants that were grown in liquid medium or on solid medium (Supplementary Fig. 18).
7. We have analyzed the stomatal phenotypes of wild type and *kin10* mutant on the solid medium to confirm that *kin10* mutant is less sensitive to sucrose for stomatal development of plants grown on solid medium.
8. We have analyzed the stomatal phenotype of *tps1-11* in the presence of different concentrations of sucrose.
9. We have performed the immunoblot to analyze the effects of sucrose on the accumulation of PSCH and SPCH-4A (Fig. 5e-f).
10. We have analyzed the colocalization of *pKIN10::KIN10-YFP* and *pSPCH::SPCH-RFP* in the stomatal lineage cells of plants that were grown in liquid medium or on solid medium.
11. We have added the information about ANOVA analysis in this study (Supplementary Table 3).
12. We have changed the abstract and discussion as suggested by reviewers.

Please find below a detailed responses to the points raised.

Sincerely,

Mingyi

REVIEWER COMMENTS

Reviewer #1 (Remarks to the Author):

In this revision, authors have included new controls, increased the quality of some figures, provided all the missing experimental details, and added new data, altogether leading to a significant improvement of the manuscript.

The work now clearly demonstrates that KIN10 promotes stomata development under certain conditions, and that this involves SPCH phosphorylation and consequent stabilization. This is shown for plants at different developmental stages and grown under different conditions that are likely associated with some level of stress (very short days or liquid cultures, which are likely hypoxic). Furthermore, a *tps1* mutant where KIN10 is more active due to reduced accumulation of T6P, also shows a similar phenotype of higher stomata index. And the phenotypes of plants expressing phosphomutant versions of SPCH provide further and very clear evidence of the relevance of this mechanism for stomata development. Altogether, this part is sound and convincing. This is on its own really exciting, highly novel and very well demonstrated, so in my opinion would be sufficient to merit publication in this journal.

Response: Thank you very much for the positive comments about our work.

What I think makes the manuscript still very confusing is the effect of sucrose on KIN10. By trying to "force" the manuscript into a story centered around how KIN10 is regulated rather than around its effects on stomata development, the storyline becomes very confusing and also with conflicts and misleading conclusions that can detrimentally impact this field in the long term, if published like this.

Response: Sorry for this confusing description. In order to make the manuscript more convincing, we have removed the results and discussion about 5% sucrose.

The main problem is the conflict between what is observed at the phenotype level and at the molecular level:

1) AT 5% SUCROSE, there is a clear effect of sucrose/T6P on KIN10 at the molecular

level (Fig. 4; high T6P accumulation and dephosphorylation of KIN10 due to reduced SnAK-KIN10 interaction; this likely results into lower KIN10 activity), but no KIN10-dependent plant phenotype is shown

Response: Thank you for pointing this out. In our experimental condition, we found that the stomatal index of cotyledon at 5% sucrose was lower than that at 1% sucrose, and the MMC cell type marker line *pSPCH::SPCH-GFP* showed the higher expression levels at 1% sucrose than that at 5% sucrose concentration. These changes in the pattern of stomatal index and SPCH protein are consistent with changes in the trends of phosphorylated KIN10 at the variable concentrations of sucrose, suggesting the phosphorylated KIN10 plays an important role in stomatal development. However, these results are confused and difficult to understand, so we have removed these data in the new version of manuscript.

2) AT 1% SUCROSE, the phenotype of plants overexpressing KIN10 is very clear (higher SI), but no impact on KIN10 activity is shown at the molecular level. The authors claim that sucrose promotes KIN10 activity at 1% based on the fact that there is more KIN10 accumulated (Fig. 3a and c). However, in those conditions the expression of TPS5 actually tells the opposite (Fig. 3b). This gene is usually induced by sugar and repressed by KIN10 activity, suggesting that in these conditions KIN10 is actually being inhibited. Most of the phenotypic data in the data is from 1% sucrose (at least the data involving KIN10), so to make a point about sucrose having an effect on KIN10 activity at the molecular level, 1% suc would be the most relevant condition and not 5%

Response: Thank you for pointing this out. KIN10 is a master energy sensor in plants. Multiple regulator mechanisms are combined to regulate the activity of KIN10. Nucleocytoplasmic shuttle is one of the important regulatory mechanisms for KIN10 activity. KIN10 with phosphorylation at the T-loop site needs to be located in the nucleus to interact and phosphorylate some transcription factors to reprogram gene expression, and then regulate plant growth and stress response. Here, in this study we showed that sucrose treatment significantly increased the protein levels of KIN10 in

all types of cells. We found, in most cells such as pavement cell, the accumulated-KIN10 protein were localized in the cytoplasm and failed to regulate the expression of the downstream genes such as *TPS5*. However, in the stomatal lineage cells, KIN10 proteins were found to localize in nuclear to phosphorylate and stabilize SPCH and promote stomatal development. The protein levels of SPCH could indirectly reflect the activity of KIN10 in stomatal lineage cells. Our genetic and biochemical assays showed that SPCH is highly accumulated in the overexpression of *KIN10* plants or wild type plants that were grown on liquid medium with 1% sucrose, indicating that KIN10 has high activity in the stomatal lineage cells at 1% sucrose condition.

Altogether, I think the main “problem” comes from the fact that the authors are trying to extrapolate the results of short sugar feeding (where KIN10 is accumulated in response to 1% suc, but it is likely less active), to the results of long-term growth, where KIN10 activity promotes stomata development. And they may not be extrapolatable. More correct would be to analyze KIN10 accumulation and its phosphorylation under the conditions where the phenotype is observed (growth over several days on liquid medium with/without 1% sucrose). I don't see this as necessary, because I think there is already a beautiful story on the stomata development part and the effect on SPCH. What I suggest simply is that the weight of the story is removed from the regulation of KIN10 by sucrose and placed on the effect of KIN10 on SPCH and stomata development.

Response: We agree and therefore we have changed the manuscript as suggested.

1) This would start from the title (which I think would be more appropriate to change to something along these lines: KIN10 promotes stomata development through stabilization of the SPCH transcription factor)

Response: Thank you for pointing this out. We have changed the title as followed: “KIN10 promotes stomatal development through stabilization of the SPEECHLESS transcription factor”

2) Then, the results of 5% sucrose might have to be removed

Response: Thank you for pointing this out. We have removed the results of 5% sucrose in the new version of manuscript.

3) The parts of the Discussion that are misleading should be either removed or substantially rephrased:

Response: We have changed the manuscript as suggested.

- 68-69. Perhaps better to indicate “Under certain conditions, sucrose promotes stomatal formation, but the mechanism remains unclear”

Response: Thank you for pointing this out, we have changed as suggested.

- 69-70. "Here, we showed that sucrose dynamically regulated stomatal development by controlling the activity of KIN10". Given that an effect by sucrose on KIN10 phosphorylation (potentially translating into lower activity) is shown only at 5% sucrose, this is a large generalization

Response: Thank you for pointing this out, we have changed as followed:

“Sugar has been reported to promote stomatal formation under in liquid culture condition, but the molecular mechanism remains unclear. Here, we showed that KIN10 is involved in the sugar-promoted stomatal development. When plants were grown in liquid media, sucrose treatment significantly induced the KIN10 protein accumulation by increasing its translation. Overexpression of *KIN10* resulted in the increased stomatal index, and loss of function of *KIN10* led to the reduced stomatal index. KIN10 displayed the cell type specific subcellular localization pattern in the epidermal cells of cotyledon, mainly localized in the nucleus of the stomatal lineage cells and guard cells, but localized in the cytoplasm of the pavement cells. The nuclear-localized KIN10 in stomatal lineage cells phosphorylated and stabilized SPCH to promote stomatal development. These results demonstrated that fine-tuning of KIN10 activity by environmental and developmental signals optimizes stomatal

development in Arabidopsis.”

- Lines 439-442. "Our genetic and biochemical analyses demonstrated that, despite its conserved functions, SnRK1 regulation has evolved differently from that of analogous SNF1 and AMPK and that sugar produced by photosynthesis finely regulates the activity of SnRK1 in response to different sucrose conditions". This paragraph has implicit the idea that sucrose induces KIN10 activity, with which I strongly disagree and do not think the authors provide support.

Response: Thank you for pointing this out, we have changed as followed:

“Here, our genetic and biochemical analyses revealed an important role of SnRK1 in plant stomatal development under conditions that are likely associated with mild energy starvation of plants, such as short day photoperiod or liquid cultures. Sucrose supply induces the accumulation of KIN10 by increasing its translation in the liquid culture condition. KIN10 is expressed in all epidermal cells, but displays the cell type specific subcellular location. The nuclear-localized KIN10 is highly enriched in the stomatal lineage cells. Under certain stress conditions, activated KIN10 phosphorylates SPCH to increase its stability, thereby promoting stomatal development. Thus, our research demonstrates the highly conserved SnRK1 kinase as a positive regulator of SPCH and stomatal development that influences many plant responses to changing environmental conditions.”

- Lines 471-474. "Increasing the activity of KIN10 by mimicking the phosphorylation of a conserved threonine residue in the activation T-loop or overexpression of SnAK2, an upstream kinase of KIN10, also led to increased stomatal index".

- The T175D mutation was never shown to result in enhanced KIN10 activity in vivo, so please just say that expression of the T175D variant led to increased SI

- Also, no evidence is provided for increased KIN10 activity in SNAK OE lines (see minor comments) so just say that overexpression of SnAK2 led to increased SI

Response: Thank you for pointing this out, we have changed as followed:

“the transgenic plants of *pKIN10::KIN10^{T175D}-myc* and *p35S:SnAK2-GFP* both

showed the increased stomatal index.”

- Lines 492-498. Our genetic and biochemical assays showed that the T-loop phosphorylation is important for the KIN10-regulation of stomatal development in Arabidopsis. Low concentrations of sucrose (0.5% and 1%) induced the accumulation of phosphorylated KIN10 protein and induced the stomatal formation. On the contrary, high concentrations of sucrose (3% and 5%) led to increased levels of Tre6P and decreased levels of phosphorylated KIN10 proteins, subsequently reducing the stomatal formation". By mixing what happens at the level of protein accumulation (1%) with what happens at the level of T-Loop phosphorylation (5%) under the common denominator of “levels of phosphorylated KIN10” this becomes very misleading and gives the idea that T-loop phosphorylation per se is changing in these two conditions, which is not true.

Response: Thank you for pointing this out. In order to make the manuscript more convincing, we have removed the results and discussion about 5% sucrose.

- Lines 498-502. "Increasing the activity of KIN10 by overexpression of its upstream kinase SnAK2 or expression the KIN10T175D, which mimics the phosphorylation of KIN10 at T175, both led to the increased stomatal index. These results indicated that the T-loop phosphorylation of KIN10 plays a critical role for sugar-promoted stomatal development". See comments for lines 471-474

Response: Thank you for pointing this out, we have changed as followed:

“the transgenic plants of *pKIN10::KIN10^{T175D}-myc* and *p35S:SnAK2-GFP* both showed the increased stomatal index.”

- Lines 559-579

Overall the discussion should significantly reduce all these parts related to KIN10 regulation and how this contrasts with the situation in animals, since there is not sufficient evidence to support these claims. Instead the relevance of the uncovered

mechanisms of promoting stomata development though KIN10 could be discussed a bit in more detail. It makes all the sense that KIN10 promotes this process, as formation of stomata is the first step towards carbon assimilation. Also the effect of sucrose can be reasoned and justified without entering into how that translates into KIN10 activity. Under very short photoperiods, soil-grown plants induce KIN10 activation, allowing the use of resources to develop more stomata. In hypoxic liquid cultures of very young seedlings, KIN10 activation is probably not enough to support stomata development, simply because there are no resources to do so, and hence the effect of KIN10 activation is only seen under 1% sucrose (as compared to no sucrose control). Altogether the KIN10-Stomata connection is a really exciting and unique finding that should be emphasized much more!

Response: Thank you for pointing this out. We have deleted this paragraph of discussion, and added some discussion about the functions of KIN10 on stomatal development.

Other minor things:

1) There are still no molecular analyses provided of the plants used in Fig. 4h-i. How many lines were obtained for the lines in Fig. 4h and did they all behave similarly? Without seeing the level of KIN10 expression for the line expressing the WT vs. the T175D variant is not possible to conclude whether the phenotype differences are indeed due to differences in phosphorylation or in the level of expression of these variants

Response: Thank you for pointing this out. We have performed the semi-quantitative RT-PCR to analysis the expression levels of *KIN10-myc*, *KIN10^{T175D}-myc* and *KIN10^{T175E}-myc* in different transgenic plants. The results showed the expression levels of *KIN10-myc*, *KIN10^{T175D}-myc* and *KIN10^{T175E}-myc* were similar in the *pKIN10:KIN10-myc*, *pKIN10:KIN10^{T175D}-myc* and *pKIN10::KIN10^{T175E}-myc* transgenic plants. Whereas, the *pKIN10:KIN10^{T175D}-myc* and *pKIN10::KIN10^{T175E}-myc* transgenic plants displayed the higher stomatal index than the *pKIN10:KIN10-myc* transgenic plants, indicating the residue Thr175 plays an

important role for KIN10 on stomatal development.

Supplementary Fig.9. Mutation of Thr175 to Asp enhances the promoting effects of KIN10 on stomatal development

a, Semi-quantitative RT-PCR analysis the expression levels of *KIN10-myc* and *KIN10^{T175D}-myc* in wild type and different transgenic plants. Seedlings of Col-0, *pKIN10::KIN10-myc* and *pKIN10::KIN10^{T175D}-myc* transgenic plants were grown on solid medium with 1% sucrose under 16h light/8h dark for 5 days. *PP2A* was used to as the internal control. **b**, Site directed mutation of Thr175 to aspartic acid enhances the promoting effects of KIN10 on stomatal development. Seedlings of Col-0, *pKIN10::KIN10-myc* and *pKIN10::KIN10^{T175D}-myc* transgenic plants were on solid medium with 1% sucrose under 16h light/8h dark for 10 days. Error bars indicate standard deviation (S.D.) (n=10). Different letters above the bars indicated statistically significant differences between the samples (ANOVA analysis followed by Uncorrected Fisher’s LSD multiple comparisons test, $p < 0.05$).

2) For the SnAK OE. The authors included now new data in Suppl Fig. 11 that shows enhanced accumulation of phosphorylated KIN10 in the SnAK OE lines. However, without showing the total accumulation of KIN10 in these lines this is not very informative and does not allow to make claims related to KIN10 activity/phosphorylation

Response: We agree and thereby we changed the manuscript as followed:

“Furthermore, we found that the transgenic plants of *pKIN10::KIN10^{T175D}-myc* and *p35S::SnAK2-GFP* showed the increased stomatal index comparing to wild type plants.”

3) In the Nature publishing forms please include info on the KIN10 T-loop antibody, and about the ANOVA analyses

Response: Thank you for pointing this out, we have added the information of KIN10 T-loop antibody in the Material and Methods, and added the information about ANOVA analyses in the supplemental table 2.

Reviewer #2 (Remarks to the Author):

I am impressed by the amount of work that was done in such a short time and the high quality of the new data that further strengthen the major conclusions. Most of the comments have been addressed. My only major concern remaining is that conclusions about the effects of sucrose on SnRK1 T-loop phosphorylation and activity should be phrased more carefully/correctly in some places. Lower sucrose levels lead to accumulation of both total and T-loop phosphorylated SnRK1. It should be mentioned more explicitly that there is no effect on the relative T-loop phosphorylation levels. For example, the (new) title on line 258 and the sentence on line 268-269 are still somewhat misleading. Of course, this increase could still lead to an overall increase in SnRK1 activity. Conversely, the effects of carbon starvation conditions on stomatal development are surprising as they should activate SnRK1. For example, on line 566 in the discussion, it is mentioned that the data show that sugar starvation inhibits the ‘activity’ of KIN10. Therefore, it would still be nice to have an independent readout of SnRK1 activity (in addition to SPCH phosphorylation and stability) in response to the different levels of sugars in the experimental system used, such as qPCR data from selected target gene expression. Some minor language editing is still needed.

Response: Thank you very much for the positive comments about our work. According to the suggestions of editors and reviewers, we have removed the results

about the effects of sucrose on the T-loop phosphorylation of KIN10, and reframed our manuscript to focus on how KIN10 regulates the stomatal development under certain conditions.

Reviewer #3 (Remarks to the Author):

The authors undertook an impressive effort to address my concerns. I am fully convinced by their data regarding the effect of T6P (Fig. S10) and the nuclear localization (Fig. S15), which was even tested in liquid culture and on plates (Fig. S14). This being said I am still not convinced by the main conclusion that sugar regulates stomatal development.

Main concerns:

1. Under 16h and 12h light conditions there is almost no difference if Col-0, kin10 and Kin10-OE are grown on plates and there is only a difference once the photoperiod is changed to only 4h (Fig. S7). These are extremely unnatural conditions for a plant to grow in and do not occur in nature. The same is true for submerged plants grown in liquid culture. Due to a lack of gas exchange this is essentially a carbon starved situation much like the 4h time period. Therefore I think what the authors observe here is a sugar-dependent role of Kin10 if a plant is darkness-stressed and/or carbon-starved, a situation that never occurs in real life. In my opinion the title of the manuscript implies that sugar optimizes stomatal development in general but actually the MS shows that sugar optimizes stomatal development in extended darkness and when submerged. I would expect that this fact is at least discussed and acknowledged. You could for example discuss that this sugar-dependent signalling could only have been revealed using these extreme carbon-starved conditions and that this module likely coordinates density to photoassimilates and light regime even under natural conditions. I would also like the fact to be stated that liquid culture enhances the effect due to submergence and the resulting strong reduction of gas exchange and thus assimilation.

Response: Thank you for pointing this out. We have changed the title as followed:
“KIN10 promotes stomatal development through stabilization of the SPEECHLESS transcription factor”

In addition, we have added the discussion as followed:

“In this study, a liquid culture system was used to assess stomatal index and KIN10 activity. The significant promotion of stomatal development in plants grown in liquid medium with sucrose might be due to the activation of KIN10 by mild hypoxia of the liquid culture, similar to submergence treatment. Accordingly, exogenous sucrose supply had no obvious promoting effects on stomatal index of plants grown on solid medium under the 16h light/8h dark photoperiod condition, but it significantly increased the stomatal index of plants grown on solid medium under the 4h light/20h dark photoperiod condition. This may be due to the activation of KIN10 under prolonged darkness. These results indicated that sugar induces the accumulation of KIN10 by enhancing its translation, while liquid culture or prolonged darkness might activate KIN10 through inducing its phosphorylation. Subsequently, the activated and nuclear-localized KIN10 in stomatal lineage cells phosphorylated and stabilized SPCH to promote stomatal development. Overall, the combination of multiple regulatory mechanisms controls the activity of KIN10 and thereby optimizes stomatal development.

The promotion of KIN10 on stomatal development may occur when plants encounter sunny days after consistent cloudy weather for many days, or when plants were grown in flooding stress. The cloudy weather or flooding stresses resulted in the energy starvation of plants. When plants encounter the sunny day again, plants restart photosynthesis and produce sugar to induce the accumulation of KIN10. The activated KIN10 phosphorylates and stabilizes SPCH to promote stomatal formation and then increase the ability of plant photosynthesis and carbon assimilation, thus forming a positive feedback loop to help plants recovering from stress. Taken together, our research not only establish the highly conserved SnRK1 kinase as a positive regulator for stomatal development, but it also provides a tractable system for investigating how

environmental stresses integrate with metabolic signals to modulate stomatal development.”

Minor concerns:

2. On line 585, the authors say that "in the present study, positive relationship between light and stomatal density was revealed" after the actually introduce two studies that showed that years ago in the sentences above. They rather reveal that the lack of sugar is probably affecting this but only if the plant is basically carbon starved.

Response: Thank you for pointing this out. In order to make the manuscript more convincing, we have deleted this paragraph of discussion in the new version of our manuscript.

Reviewer #4 (Remarks to the Author):

Bai et al., have extensively revised their manuscript and provide data from several new experiments. Current manuscript shows that KIN10 promotes stomatal development through phosphorylation of stomatal master regulator SPEECHLESS (SPCH) and this interaction is dependent on sucrose dosage. Authors also show that sucrose promotes KIN10 translation, however, high sucrose concentration reduces KIN10 activity (by decreasing amount of phosphorylated KIN10). Upstream kinases (SnAKs) of KIN10 are required for KIN10 activity: high concentrations of sucrose (in growth media) cause accumulation of Tre6p, which prevent SnAKs-KIN10 interaction and thus, lead to reduction in amount of active KIN10, reduced SPCH stability as well as decreased stomatal index. Authors show that exposure to Tre6p leads to decreased stomatal index whereas *tps1-11* mutant (key enzyme in Tre6p biosynthesis) display increased stomatal index supporting their model. Further, authors identify 4 residues in SPCH which are targets of KIN10 mediated phosphorylation and show that phosphorylation of these residues is required for SPCH stability.

Major comments

1) Role of KIN10 in stomatal development

Authors have added data on several new experiments where they investigate how different growth conditions such as light intensity and duration modify sucrose dependent stomatal development. This is mostly very nice additional data. They also compare effects of liquid and solid growth media on stomatal development in KIN10-OE/kin10/wild-type in combination with sucrose treatment and different day length. Row 177-179: “The results showed that overexpression of KIN10 significantly induced stomatal development in the presence of sucrose, particularly under the 4h light/20h dark photoperiod condition (Supplementary Fig. 7a-c).” Based on their data authors conclude that (Row 185-186): “These results indicated that KIN10 positively regulates stomatal development.” Finally, authors study stomatal development in same plant lines grown in the soil and use 5th rosette leaves for analysis: Row 183- 186: “The stomatal index of rosette leaves of KIN10-Ox was higher, and that of kin10 mutant was lower than the stomatal index of wild type rosette leaves (Supplementary Fig. 8a-d). These results indicated that KIN10 positively regulates stomatal development.” Authors come up with a model where mild hypoxia (caused by liquid growth media) or darkness cause KIN10 activation whereas sucrose induces KIN10 translation. Therefore, in the absence of KIN10 activation stomatal development is not affected by sucrose dosage (on plate). However, there is some confusing and conflicting results on kin10, KIN10-OE and wild-type grown on plate with and without sucrose (cotyledons, solid vs liquid growth media) and in soil (leaves). Please describe this currently mostly ignored data in the text and hypothesize what could be underlying reason for these observations:

- Why kin10 behaves differently under the 12h light/12h dark and 4h light/20h dark photoperiod conditions on solid vs liquid growth media? On solid growth media kin10 shows elevated stomatal index compared to wild-type and KIN10-OE without sucrose

whereas in liquid growth media there is no difference between these lines without sucrose. This is not in line with stomata promoting function of KIN10 authors are suggesting!

Response: Thank you for pointing this out. We think this maybe due to the redundant function of KIN10 and KIN11 in regulating stomatal development. To test this possibility, we analyzed the phosphorylated protein levels of KIN10 and KIN11 in wild type, *kin10* and *kin11* mutant that were grown under 12h light/12 dark for 10 days in liquid sugar free medium or on solid sugar free medium. The results showed that the amounts of phosphorylated KIN11 and KIN10 proteins were lower in the *kin10* mutant and *kin11* mutant, respectively, than in wild type when grown in liquid sugar free medium. However, when plants were grown on solid sugar free medium, *kin10* mutants contained the higher levels of phosphorylated KIN11 protein than wild type plants. The different levels of phosphorylated KIN11 protein in *kin10* mutant under different culture conditions could partially explain the different stomatal phenotypes of *kin10* mutant under different growth conditions, but its molecular mechanism needs to be further investigated.

Supplementary Fig.18. Immunoblot analysis the phosphorylated KIN10 and KIN11 in wild type, *kin10* and *kin11* mutants.

Seedlings of wild type, *kin10* and *kin11* mutant were grown under 12h light/12 dark for 10 days in liquid sugar free medium or on solid sugar free medium. The phosphorylated KIN10 proteins or KIN11 proteins were analyzed by anti-AMPKT172 antibody, actin bands were used as loading control.

- Why *kin10* do not respond at all to sucrose when grown on solid growth media (even when it is grown under the 4h light/20h dark photoperiod condition unlike other

analyzed lines) whereas in liquid growth media it shows elevated stomatal index as a response to sucrose (although less than other lines)?

Response: Thank you for pointing this out. Our results showed that sucrose supply increased the stomatal index of wild type plants from 20% to 36% when plants were grown in liquid medium, but only increased the stomatal index of wild type plants from 10% to 13.5% when plants were grown on solid medium under 4h light/20h dark condition, indicating that sucrose had the weaker promoting effects on stomatal development in solid culture condition than that in liquid culture condition. We think this maybe because that in the solid culture condition, sucrose is firstly absorbed by root from medium and then transported to leaf to regulate stomatal development; while in the liquid culture condition, sucrose is directly sensed by the epidermal cells of leaves to control stomatal development. To further determine how *kin10* mutants respond to sucrose in solid culture condition, we analyzed the stomatal phenotype of wild type plants and *kin10* mutants on solid medium with different concentrations of sucrose. The results showed that 1% sucrose and 3% sucrose both significantly increased the stomatal index of wild type plants, while for the *kin10* mutants, only 3% sucrose significantly increased the stomatal index, indicating mutation of KIN10 resulted in the reduced sensitivity to sucrose on stomatal development.

Supplementary Fig. Quantification of the effects of sucrose on the stomatal development in wild type and *kin10* mutants when grown on solid medium.

Seedlings of Col-0 and *kin10* mutants were grown on ½ MS solid medium with different concentration of sucrose. Error bars indicate standard deviation (S.D.) (n=10). Numbers between bars indicated the relative fold changes of average means in the indicated materials. Different letters above the bars indicated statistically significant differences between the samples (ANOVA

analysis followed by Uncorrected Fisher's LSD multiple comparisons test, $p < 0.05$).

- Why soil grown wild-type, KIN10-OE and kin10 plants show different stomatal indexes in rosette leaves without sucrose treatment whereas on plate grown plants do not show difference in cotyledon stomata index (KIN10-OE) or show even difference in opposite direction (kin10) in same light conditions? Could this be related to developmental program of the organ or differences in the growth conditions?

Response: We agree that the promoting effect of KIN10 on stomatal development is related to the specific developmental program of the organ or the growth conditions. We think that sugar produced by photosynthesis or exogenous supply induces the accumulation of KIN10 by enhancing its translation, while mild energy starvation of plants, such as prolonged darkness or liquid culture might activate KIN10 through inducing its phosphorylation. Subsequently, the activated and nuclear-localized KIN10 in stomatal lineage cells phosphorylated and stabilized SPCH to promote stomatal development. Thus, the combination of multiple regulatory mechanisms controls the activity of KIN10 and thereby optimizes stomatal development.

2) Endogenous sucrose vs sucrose treatment

Authors do majority of the experiments with liquid growth media supplied with sucrose. To their credit, authors have investigated how different growth conditions such as light intensity and duration modify sucrose dependent stomatal development. They also compare effects of liquid and solid growth media on stomatal development in combination with sucrose treatment. This data shows that sucrose combined with liquid growth media enhances nuclear localization of KIN10 specifically in small stomatal lineage cells (likely to be stomatal precursor cells) compared to sucrose combined with solid growth media. Authors come up with a model where mild hypoxia (caused by liquid growth media) or darkness cause KIN10 activation whereas

sucrose induces KIN10 translation. Therefore, in the abundance of KIN10 but absence of active KIN10 stomatal development is not affected by sucrose dosage. However, plants are capable to produce sugar by themselves and do not rely on external sugar source. Thus, it is unclear how well this data applies to the situation in soil grown plants. Please elaborate these aspect in the discussion.

Response: Thank you for pointing this out. We have added the discussion as followed:

“The promotion of KIN10 on stomatal development may occur when plants encounter sunny days after consistent cloudy weather for many days, or when plants were grown in flooding stress. The cloudy weather or flooding stresses resulted in the energy starvation of plants. When plants encounter the sunny day again, plants restart photosynthesis and produce sugar to induce the accumulation of KIN10. The activated KIN10 phosphorylates and stabilizes SPCH to promote stomatal formation and then increase the ability of plant photosynthesis and carbon assimilation, thus forming a positive feedback loop to help plants recover from stress. Taken together, our research not only establishes the highly conserved SnRK1 kinase as a positive regulator for stomatal development, but also provides a tractable system for investigating how environmental stresses integrate with metabolic signals to modulate stomatal development.”

3) Sugar signalling mutant data

Authors have added new data on few additional lines presumably participating in sugar signalling (*tor-es*, *gin2-1*). However, description of these genes and their role in sugar signalling are completely missing from current manuscript. Also these lines are not described in the materials and methods. Data is only briefly described: Row 157-159: “We found that several sugar signalling pathways are involved in the sugar-mediated stomatal developmental process (Supplementary Fig. 4a, b)” Please consider whether this is necessary information for this paper. If it is, please explain what are the lines shown in the Supplementary Fig. 4a & b, and how these results

support the statement in the manuscript (Row 157 -159).

Response: Thank you for pointing this out. We think that the weight of the new version of this manuscript is the functions of KIN10 on stomatal development, the results about other sugar signaling components are not necessary, so we have removed these results from our manuscript.

Minor comments

(Most are related to writing - rigorous proof-reading would be helpful)

1. Row 53- 56: “Plant stomata represent an excellent system..., due to their simple, flexible, and developmental trajectory” Please modify: “Plant stomata represent an excellent system..., due to their simple and flexible developmental trajectory”

Response: Thank you for pointing this out, we have changed as suggested.

2. Row 56-57: “The stomatal lineage in Arabidopsis thaliana initiates by asymmetric divisions of undifferentiated meristemoid mother cells (MMC)” Please modify: “The stomatal lineage in Arabidopsis thaliana is initiated by asymmetric division of undifferentiated meristemoid mother cell (MMC)”

Response: Thank you for pointing this out, we have changed as suggested.

3. Row 68: “Sugar has been reported to promote the stomatal formation” Please use either “to promote stomatal development” or “to promote stomata formation”

Response: Thank you for pointing this out, we have changed as suggested.

4. Row 74: “Tre6P was significantly accumulated” Please rephrase: “Tre6P levels were significantly increased”

Response: Thank you for pointing this out, we have changed as suggested.

5. Row 83: “Stomata, the pores on plant epidermis for gas exchange with the atmosphere” Please rephrase, for example: “Stomata, the pores on plant epidermis,

which facilitate gas exchange between the plant and atmosphere”

Response: Thank you for pointing this out, we have changed as suggested.

6. Row 162-166: “Because the KIN10-Ox genetic material that we obtained from Arabidopsis Biological Resource Center (ABRC) is in the Landsberg (Ler) background, in which ERECTA (ER), a key component of stomatal development, was mutated leading to more stomata”. Please rephrase: “Because the KIN10-Ox genetic material that we obtained from Arabidopsis Biological Resource Center (ABRC) is in the Landsberg (Ler) background, in which ERECTA (ER), a key component of stomatal development, was mutated, it showed increased number of stomata.”

Response: Thank you for pointing this out, we have changed as suggested.

7. Row 167-170: “The results showed that, in the presence of sucrose, overexpression of KIN10 in the Col-0 background also led to the increased stomatal index, more stomata number in a whole cotyledon and higher ratio of clustered stomata comparing to Col-0 plants (Fig. 2b and Supplementary Fig. 5a-i).” Please rephrase: “The results showed that in the presence of sucrose, overexpression of KIN10 in the Col-0 background also led to the increased stomatal index, elevated stomatal numbers in whole cotyledons, and higher ratio of clustered stomata compared to Col-0 plants. (Fig. 2b and Supplementary Fig. 5a-i).”

Response: Thank you for pointing this out, we have changed as suggested.

8. Figure S5c is labelled with number of stomata/leaf. However, legend says that cotyledons have been analyzed. Please modify and label with number of stomata/cotyledon

Response: Thank you for pointing this out, we have changed as suggested.

9. Row 208-201: “This might be due to the accumulation of SPCH inducing cell division to form many small cells of which only a few will eventually develop into stomata” This sentence is hard to understand. Please rephrase.

Response: Thank you for pointing this out, we have changed as followed:

“This might be due to that the accumulated SPCH protein induces the division of stomatal precursor cell to form more small cells, but only a few small cells eventually develop into stomata.”

10. Row 328: “florescent protein” Please modify:” fluorescent protein”

Response: Thank you for pointing this out, we have changed as suggested.

11. Figure 4e-f: Please explain in the legend shortly how tobacco samples have been treated (for example, how long sucrose treatment was used).

Response: Thank you for pointing this out. The Agrobacterium transfected tobacco plants were kept in the greenhouse for overnight at 22°C, and then treated with different concentrations of sucrose for 24 hours. Fluorescent signals from at least 100 cells from 5 leaves of 3 different plants were visualized by using the LSM-700 laser scanning confocal microscope (Zeiss) and the signal intensities of YFP and RFP were determined by ImageJ software.

12. Row 335-337: “These results indicated that the decreased phosphorylation of KIN10 by high sucrose treatment is due to the accumulation of Tre6P, which reduced the binding affinity of KIN10 to its upstream kinase SnAKs.” These results do show that sucrose induces Tre6P accumulation (Fig 4d) and reduces KIN10 phosphorylation (Fig 4a) but do not show that there is causality between the two. Please modify description of the findings.

Response: Thank you for pointing this out. In order to make the manuscript more convincing, we have deleted the results about 5% sucrose in the new version of manuscript according to the suggestions of editors and reviewers.

13. Row 345-347: “Mutation of TPS1 resulted in a significantly increased stomatal index comparing to that of wild type plants in the presence of sucrose (Supplementary Fig. 10b).” Tre6P should accumulate above 1% sucrose conditions and this prevents

KIN10 activation leading to destabilization of SPCH according to model. Here effects of 1 % sucrose has been tested. It would be more interesting to see whether *tps1* mutant lacks the reduction of stomatal index in high sucrose conditions and whether it shows elevated SPCH levels in high sucrose conditions.

Response: Thank you for pointing this out. We have analyzed the stomatal index of *tps1* mutants in the presence of different concentrations of sucrose. The results showed that treatment with 1% sucrose and 3% sucrose in wild type plants both increased the stomatal index, and 1% sucrose displayed the better promotion effect. However, in *tps1-11* mutant, 1% sucrose and 3% sucrose showed the similar promoting effects on stomatal development, which maybe due to the fact that high sucrose can no longer induce the Tre6P accumulation in *tps1-11* mutant.

Supplementary Fig. Quantification of the effects of different concentrations of sucrose on stomatal development in *tps1-11* mutants.

Seedlings of wild type Ler and *tps1-11* were grown in the liquid ½ medium with the different concentrations of sucrose under 16h light/8h dark for 10 days. Error bars indicate standard deviation (S.D.) (n=10). Different letters above the bars indicated statistically significant differences between the samples (ANOVA analysis followed by Uncorrected Fisher's LSD multiple comparisons test, $p < 0.05$).

14. Row 373-376: “KIN10 displayed the ubiquitous expression pattern in all epidermal cells, and it was mainly localized in the nucleus of guard cells and the smaller cells less than 200 μm², while distributed in the cytoplasm in the pavement cells (Supplementary Fig. 14a-e).” It is interesting that there is significantly increased nuclear localization of KIN10 in smaller cells in liquid growth media with 1 % sucrose. These cells contain stomatal precursor cells and thus are the most relevant

cells for stomatal development. This data should be discussed in the text more clearly since it suggests that KIN10 activity in stomatal lineage cells is very sensitive for the use of liquid growth condition.

Response: Thank you for pointing this out, we have added the discussion as followed: “In this study, KIN10-YFP fluorescent signals were present in all epidermal leaf cells. Nevertheless, the nuclear localization signals of KIN10-YFP were significantly enhanced in the small cells that might belong to the stomatal lineage cells, leading to phosphorylation of SPCH by KIN10 in the nucleus to increase SPCH protein stability and promote stomatal development. The nuclear/cytoplasmic ratios of KIN10-YFP in stomatal lineage cells of plants grown in liquid growth media were significantly higher than that grown on solid media in the presence of 1% sucrose, indicating KIN10 activity in stomatal lineage cells is more sensitive for use of liquid growth condition. Consistent with this, plants grown in liquid media with 1% sucrose showed the increased stomatal index comparing to the plants grown on solid media with 1% sucrose, which may be due to the increased KIN10 activity by the mild hypoxia of the liquid culture.”

15. Figure 5g . Legend “Quantification of stomatal index in wild type plants and different SPCH complemented transgenic plants (n=10).” Please explain in the legend genotypes and names of the transgenes.

Response: Thank you for pointing this out, we have added the details in the figure legend.

16. Supplementary Fig. 19a. Legend. Please explain whether “medium” refers to liquid or solid media (Fig S19a).

Response: Thank you for pointing this out, we have added the details in the figure legend.

17. Row 430-432: “More importantly, SPCH-4A performed weaker protein stability compared to wild type SPCH and SPCH-4D in developing cotyledon epidermis (Fig.

5h, Supplementary Fig. 19c,e).“ Fig19.e It is clear that starting dosage of SPCH variants are very different, however, 24h sucrose treatment seems to increase levels of both SPCH-myc and SPCH4A-myc. Please quantify the increase in order to see whether the increase is similar or different. This would tell whether or not sucrose dependent induction of SPCH stability is dependent on those 4 mutated residues.

Response: Thank you for pointing this out, we have analyzed the effects of sucrose on the protein stabilities of SPCH-myc and SPCH4A-myc. The results showed that sucrose treatment significantly induced the accumulation of SPCH-4myc, but had a weak effect on the accumulation of SPCH-4A-myc. These results indicated the 4 residues phosphorylated by KIN10 are important for the sucrose-induced accumulation of SPCH.

Fig.5. KIN10 phosphorylates and stabilizes SPCH to promote stomatal development

e-f, Immunoblot analysis of the effects of sucrose on the protein levels of SPCH-myc and SPCH-4A-myc using anti-myc antibody. Seedlings of *p35S::SPCH-myc* and *p35S::SPCH-4A-myc* were grown in sugar free medium for 3 days, and then treated with 1% sucrose for different times. Actin bands were used as loading control. **g**, Analysis the SPCH, SPCH-4A and SPCH-4D protein intensities on abaxial cotyledons of 3-day-old *pSPCH::SPCH-RFP/pKIN10::KIN10-YFP*, *pSPCH::SPCH-4A-RFP/pKIN10::KIN10-YFP* and *pSPCH::SPCH-4D-RFP/pKIN10::KIN10-YFP* transgenic plant by ImageJ software. The red fluorescent signals of SPCH-RFP (n=114), SPCH-4A-RFP (n=114) or SPCH-4D-RFP (n=144) were quantified from more than 100 cells of 5 cotyledons. Error bars indicate S.D. Different letters above the bars indicated statistically significant differences between the samples (ANOVA analysis followed by Uncorrected Fisher's LSD multiple comparisons test, $p < 0.05$).

18. Row 439-442: “Our genetic and biochemical analyses demonstrated that, despite its conserved functions, SnRK1 regulation has evolved differently from that of

analogous SNF1 and AMPK and that sugar produced by photosynthesis finely regulates the activity of SnRK1 in response to different sucrose conditions.”

This sentence does not accurately describe the data in this manuscript – first, it suggests that this study has compared regulatory pathways between SNF1, AMPK, and SnRK1, which is not correct. Second, it claims that authors have shown that sucrose produced by photosynthesis regulates the SnRK1 as a response to sucrose conditions (whatever this means), which is not the case. Please rephrase to be more accurate, for example like this: ”Our genetic and biochemical analyses demonstrated that activity of SnRK1 is regulated in response to different sucrose conditions. This data suggest that despite its conserved functions, SnRK1 regulation has evolved differently from that of analogous SNF1 and AMPK and further, sugar produced by photosynthesis could finely regulate the activity of SnRK1.”

Response: Thank you for pointing this out. According to the suggestions of editors and reviewers, we focused on the regulation of KIN10 on stomatal development in the new version of manuscript. So we have changed this sentence as followed:

“SnRK1 is a central metabolic regulator of energy homeostasis in plants that is functionally and evolutionarily related to SNF1 in yeast and AMPK in mammals. Here, our genetic and biochemical analyses revealed an important role of SnRK1 in plant stomatal development under conditions that are likely associated with mild energy starvation of plants, such as short day photoperiod or liquid cultures. Sucrose supply induces the accumulation of KIN10 by increasing its translation in the liquid culture condition. KIN10 is expressed in all epidermal cells, but displays the cell type specific subcellular location. The nuclear-localized KIN10 is highly enriched in the stomatal lineage cells. Under certain stress conditions, activated KIN10 phosphorylates SPCH to increase its stability, thereby promoting stomatal development. Thus, our research demonstrates the highly conserved SnRK1 kinase as a positive regulator of SPCH and stomatal development that influences many plant responses to changing environmental conditions (Fig. 6).”

19. Row 452-456: “Under the condition of no sucrose, there is a low level of KIN10 protein in plants, which reduces the stability of SPCH and decreased stomatal index. When sucrose supply reaching to basal conditions, sucrose not only promotes cell division to generate more small epidermal cells, but also induces the accumulation of KIN10 protein by promoting its translation efficiency” When is plant experiencing condition without sucrose? During night time when there is no light? Please elaborate this, and take consideration also in your model!

Response: Thank you for pointing this out. We think the promotion of KIN10 on stomatal development may occur when plants encounter sunny days after consistent cloudy weather for many days, or when plants were grown in flooding stress. The cloudy weather or flooding stresses resulted in the energy starvation of plants. When plants encounter the sunny day again, plants restart photosynthesis and produce sugar to induce the accumulation of KIN10. The activated KIN10 phosphorylates and stabilizes SPCH to promote stomatal formation and then increase the ability of plant photosynthesis and carbon assimilation, thus forming a positive feedback loop to help plants recover from stress. Taken together, our research not only establishes the highly conserved SnRK1 kinase as a positive regulator for stomatal development, but also provides a tractable system for investigating how environmental stresses integrate with metabolic signals to modulate stomatal development.

20. Row 548-550: “As the concentration of sugar increases, the total number of epidermal cells in the leaves gradually increases, but the number of stomata reaches the peak at 1% sucrose concentration, and then gradually decreases.” I wonder what is the physiologically relevant sucrose concentration? Does cellular sucrose levels ever reach the similar levels as when growing on media containing 5 %? Could plants break down storage carbohydrates when modification of stomatal development if needed?

Response: Thank you for pointing this out. In order to make the manuscript more convincing, we have deleted the results about 5% sucrose in the new version of manuscript according to the suggestions of editors and reviewers.

21. Row 455-457: “Second, nuclear-localized KIN10 is highly enriched in guard cells and the cells less than 200 μm^2 that may belong to the stomatal lineage cells, which was further verified by the colocalization of KIN10 and SPCH in nucleus of these cells.” KIN10 is even more nuclear localized in stomatal cells in liquid conditions – is colocalization of SPCH and KIN10 tested also in solid conditions (where sucrose induced stomatal developmental response is absent?)

Response: Thank you for pointing this out. We have analyzed the colocalization of KIN10 and SPCH in liquid culture growth and solid growth condition. The results showed that KIN10 colocalized with SPCH in the stomatal lineage cells of plant grown in both liquid media or solid media.

pKIN10::KIN10-YFP/pSPCH::SPCH-RFP

$\frac{1}{2}$ MS solid culture 1% Suc

$\frac{1}{2}$ MS liquid culture 1% Suc

YFP

RFP

Merge

Supplementary Fig. KIN10 colocalized with SPCH in the stomatal lineage cells of plant grown in both liquid media or solid media.

a-b, Seedlings of *pKIN10::KIN10-YFP/pSPCH::SPCH-RFP* were grown in $\frac{1}{2}$ MS liquid medium or on $\frac{1}{2}$ MS solid medium under 16h light/8h dark photoperiod for 3 days. Scale bars in confocal images represent 20 μm .

22. Row 544-548: “When sugar levels of sugar in plants are low under low light quantity, short time light irradiance and prolonged darkness, the expansion and division of epidermal cells in plant leaves are inhibited, resulting in the defective leaf development and low stomatal index.” It would be destructive for plants to show defective leaf development in low light conditions or night time! Increase in subset of stomatal precursor divisions (amplifying and entry divisions) leads to reduced stomatal index whereas increased number of another type of stomatal precursor division (spacing division) leads to elevated stomatal index. This is because these three types of stomatal divisions produce different ratios of pavement cells and stomata during their division activity. Therefore simply reduction of all epidermal divisions or all stomatal cell divisions do not lead to reduced stomatal index. Please modify in order to take this in consideration.

Response: Thank you for pointing this out, we have changed.

23. Row 551-522: “pBASL::BASL-GFP-labeled cells that represent MMCs, meristemoids and the larger SLGCs in a concentration-dependent manner” Please modify: “pBASL::BASL-GFP-labeled cells that represent MMCs, meristemoids, and SLGCs”

Response: Thank you for pointing this out, we have changed as suggested.

REVIEWERS' COMMENTS:

Reviewer #1 (Remarks to the Author):

I think the manuscript has improved enormously and I congratulate the authors for this impressive work. The authors may just consider proofreading by an English native speaker for some minor language corrections. Also, please add reference to Supplementary fig. 5 on p. 17, line 417. Beyond this I don't have any more comments and I am happy with this version.

Reviewer #4 (Remarks to the Author):

The authors have greatly improved the manuscript. Quality and amount of the data is impressive, text have been thoroughly revised, and finally, authors come up with a convincing model well supported by the data. Shift in the focus of the manuscript makes it more approachable and new title fits well to the currently presented data. All my comments have been addressed.

minor comment:

Row 288-290: "Whereas, the Thr175 phosphorylation of KIN10 had no significant effect on its subcellular localization in stomatal lineage cells and pavement cells (Supplementary Figure 13a-f).

Supplementary Figure 13e shows that in nuclear/cytoplasmic ratio is significantly decreased in guard cells of the pKIN10-KIN10 T175D-YFP line compared to pKIN10-KIN10-YFP. Since guard cells are stomatal lineage cells, this sentence is not accurate. Please revise.

Responses to comments of reviews

Reviewer #1 (Remarks to the Author):

I think the manuscript has improved enormously and I congratulate the authors for this impressive work. The authors may just consider proofreading by an English native speaker for some minor language corrections.

Response: Thank you very much for the positive comments about our work. We have extensively edited the revised manuscript.

Also, please add reference to Supplementary fig. 5 on p. 17, line 417. Beyond this I don't have any more comments and I am happy with this version.

Response: Thank you for pointing this out, we have added the reference to Supplementary Fig. 5 on the revised manuscript.

Reviewer #4 (Remarks to the Author):

The authors have greatly improved the manuscript. Quality and amount of the data is impressive, text have been thoroughly revised, and finally, authors come up with a convincing model well supported by the data. Shift in the focus of the manuscript makes it more approachable and new title fits well to the currently presented data. All my comments have been addressed.

Response: Thank you very much for the positive comments about our work.

minor comment:

Row 288-290: "Whereas, the Thr175 phosphorylation of KIN10 had no significant effect on its subcellular localization in stomatal lineage cells and pavement cells (Supplementary Figure 13a-f).

Supplementary Figure 13e shows that in nuclear/cytoplasmic ratio is significantly decreased in guard cells of the pKIN10-KIN10 T175D-YFP line compared to pKIN10-KIN10-YFP. Since guard cells are stomatal lineage cells, this sentence is not accurate. Please revise.

Response: Thanks for point this out. We have changed this sentence as followed:

“Whereas, the Thr175 phosphorylation of KIN10 had no significant effect on its subcellular localization in small scale dividing cells and pavement cells”.